# Reconstructing volcanic radiative forcing since 1990, using a comprehensive emission inventory and spatially resolved sulfur injections from satellite data in a chemistry-climate model

Jennifer Schallock[1], Christoph Brühl[1], Christine Bingen[2], Michael Höpfner[3], Landon Rieger[4], and Jos Lelieveld[1]

[1]Atmospheric Chemistry Department, Max Planck Institute for Chemistry, Mainz, Germany
[2]Royal Belgian Institute for Space Aeronomy, Brussels, Belgium
[3]Institute of Meteorology and Climate Research, Karlsruhe Institute of Technology, Karlsruhe, Germany
[4]Institute of Space and Atmospheric Studies, University of Saskatchewan, Saskatoon, Canada

**Correspondence:** Jennifer Schallock (jennifer.schallock@mpic.de)

**Abstract.** This paper presents model simulations of stratospheric aerosols with a focus on explosive volcanic eruptions. Using various (occultation and limb-based) satellite instruments, providing vertical profiles of sulfur dioxide ($SO_2$) and aerosol extinction, we characterized the chemical and radiative influence of volcanic aerosols for the period between 1990 and 2019.

We established an improved and extended volcanic $SO_2$ emission inventory that includes more than 500 explosive volcanic eruptions reaching the upper troposphere and the stratosphere. Each perturbation identified was derived from the satellite data and incorporated as a three-dimensional $SO_2$ plume into a chemistry-climate model without the need for additional assumptions about altitude distribution and eruption duration as needed for a "point source" approach.

The simultaneous measurements of $SO_2$ and aerosol extinction by up to four satellite instruments enabled a reliable conversion of extinction measurements into injected $SO_2$. In the chemistry-climate model, the $SO_2$ from each individual plume was converted into aerosol particles and their optical properties were determined. Furthermore, the Aerosol Optical Depth (AOD) and the instantaneous radiative forcing on climate were calculated online. Combined with model improvements, the results of the simulations are consistent with the observations of the various satellites.

Slight deviations between the observations and model simulations were found for the large volcanic eruption of Pinatubo in 1991 and cases where simultaneous satellite observations were not unique or too sparse. Weak- and medium-strength volcanic eruptions captured in satellite data and the Smithsonian database typically inject about 10 kt to 50 kt $SO_2$ directly into the upper troposphere/lower stratosphere (UTLS) region or the sulfur species are transported via convection and advection. Our results confirm that these relatively minor eruptions, which occur quite frequently, can nevertheless contribute to the stratospheric aerosol layer and are relevant for the Earth's radiation budget. These minor eruptions cause a total global instantaneous radiative forcing of the order of $-0.1\,\mathrm{Wm}^{-2}$ at the top of the atmosphere (TOA) compared to a background stratospheric aerosol forcing of about $-0.04\,\mathrm{Wm}^{-2}$. Medium strength eruptions injecting about 400 kt $SO_2$ into the stratosphere or accumulation of consecutive smaller eruptions can lead to a total instantaneous forcing of about $-0.3\,\mathrm{Wm}^{-2}$. We show that it is critical to include the contribution of the extratropical lowermost stratospheric aerosol in the forcing calculations.

# 1 Introduction

Next to recent historical events in which large fires have become a major source of aerosols up to the tropopause and above (Kloss et al., 2019), stratospheric aerosol particles are mostly of volcanic origin and consist of an internal liquid mixture of sulfuric acid ($H_2SO_4$) and water ($H_2O$) (Vernier et al., 2011). The typical median diameter of these aerosol particles ranges between 200 nm for the background aerosol and 600 nm for Pinatubo conditions (Wilson et al., 2008). In this study, we incorporated stratospheric aerosol, the sulfur chemistry and the radiative transfer into a comprehensive Chemistry Climate Model (CCM), which we have used to gain a better understanding of the interaction of aerosols with the global climate system, including chemical effects. Particular emphasis is being placed on adequately modelling and understanding the impact of volcanic eruptions and other aerosol sources on the evolution of the stratospheric aerosol burden.

Sulfate and ashes from explosive volcanic eruptions can account for the majority of the aerosol burden in the stratosphere during volcanically active periods and cause strong temporal and spatial variations in the concentration and the size distribution of the particles (Vernier et al., 2011). These changes influence in turn the radiative forcing at the top of the atmosphere (or at tropopause altitude) for several years after strong eruptions (Timmreck, 2012) and can even have a more prolonged impact on the global climate (McGregor et al., 2015). After such a volcanic eruption, the enhanced radiative heating in the stratosphere exerts an effect on dynamics, influences the global spread of the volcanic cloud and leads to an upward transport of the aerosol itself as well as other chemical tracers including ozone (Timmreck et al., 1999). The aerosol radiative heating resulting from large volcanic eruptions like Pinatubo triggers enhanced tropical upwelling, which causes a lofting of the injected $SO_2$ (sulfur dioxide) and the aerosol as well as other compounds. The radiative feedback on dynamics is required to model aerosol extinction in the upper part of the volcanic aerosol plume that corresponds with observations (e.g. Aquila et al. (2013); Toohey et al. (2011)).

Due to the large variability in volcanic emissions, it is challenging to estimate future trends for stratospheric optical depth and forcing (Swindles et al., 2018; Fasullo et al., 2017; Aubry et al., 2021). Therefore, the influence of volcanic eruptions is not included in predictive simulations for future climate scenarios in the IPCC report of 2013 (IPCC, 2013).

Previous studies show that model simulations often cannot fully reproduce the AOD of satellite observations or the derived global forcing of the stratospheric aerosol layer (Solomon et al., 2011), because the number of volcanic eruptions reaching the stratosphere and treated explicitly appears to be underestimated in most analyses (Mills et al., 2016; Brühl et al., 2015; Schmidt et al., 2018; Andersson et al., 2015). Conversely, the intensity of single eruptions can sometimes be overestimated because of incorrect vertical distribution of the injection patterns in the models (e.g. Kasatochi compared to Glantz et al. (2014)). In previous studies smaller volcanic eruptions have often been included in the background atmosphere, even though they can be responsible for a radiative forcing that is twice as strong as the nonvolcanic background conditions in volcanically quiescent periods such as from 1999 to 2002 (IPCC, 2013; Solomon et al., 2011; Vernier et al., 2011). Friberg et al. (2018) included the entire time series of CALIOP (Cloud-Aerosol Lidar with Orthogonal Polarization) data from 2006 to 2015 and derived stratospheric AOD using reanalysis data for the tropopause, but only mention medium size eruptions explicitly. Radiative

forcing is estimated in this case by multiplying AOD with a factor of -25 (Hansen et al., 2005) rather than using a radiative transfer model, an approximation valid in the absence of major forest fires (see e.g. Sellitto et al. (2022)).

The model simulations in this study are compared to GloSSAC V2 (Thomason et al., 2018; Kovilakam et al., 2020), a time dependent multi-satellite zonal average aerosol climatology which provides extinction data (Figure C1 in Appendix C1). In our approach we consider as much as possible small eruptions reaching the stratosphere explicitly. For this purpose we calculate $SO_2$ injections into the stratosphere based on the Smithsonian volcano database and the most recent releases of satellite data sets, in particularly those gathered using limb sounding instruments to derive information on the vertical distribution.

For the ENVISAT (European Environmental Satellite) period 2002-2012 a first version of a new volcanic $SO_2$ inventory with improved temporal and spatial resolution was developed within the framework of ISAMIP (https://isamip.eu) (Timmreck et al., 2018; Brühl et al., 2018). The corresponding data base (https://doi.org/10.1594/WDCC/SSIRC_1) contains 3D-$SO_2$-perturbations derived from satellite data as well as integrated injected $SO_2$ masses. In this work the data base is expanded to the period 1990-2019 and considerably improved for the period 1998-2001. The simultaneous measurements from up to four instruments from 2002 to 2012 enabled us to develop a novel procedure for conversion of aerosol extinction to $SO_2$ needed for the period before and after ENVISAT.

Our method circumvents problems and uncertainties related to the classical point source approach like dependence on the model grid box size and exact location as well as the assumed vertical distribution, the assumed time interval during which the mass is injected, and effects of microphysical and chemical interactions of $SO_2$ and sulfate with injected volcanic ash and water in the early phase (Zhu et al., 2020). Since simulations of point source emissions are very sensitive to the emission conditions, in some cases it may be more appropriate to implement the main plume of volcanic emissions in the model not directly at the volcano location, but instead apply other coordinates according to satellite observations. A case study for point source emissions is shown in Appendix C3.

Non-eruptive permanently degassing volcanoes represent another natural source of aerosols, which are treated separately from active explosive volcanic eruptions. For the stratosphere these contribute in most cases only to the background since most, but not all of the released $SO_2$ is removed by oxidation and rainout in the troposphere and only a small fraction can reach the stratosphere by convection and large scale transport. This holds also for the medium size eruption of Eyjafjallajökull in 2010 from which almost no $SO_2$ reached the stratosphere as shown by MIPAS observations. Stratospheric $H_2SO_4$ is also produced from non-volcanic sulfur precursor gases, like carbonyl sulfide (OCS) (Crutzen, 1976), dimethyl sulfide (DMS) (Kettle and Andreae, 2000), and tropospheric $SO_2$ from pollution, which constitute a source of background concentration of stratospheric aerosol.

This paper is structured as follows: section 2 presents satellite data used for entering the volcanic perturbations of aerosols and $SO_2$ into the model, and for model evaluation. In section 3, the setup used for the climate model simulations is described. Section 4 contains a volcanic sulfur emission inventory with all relevant explosive volcanic eruptions detected between 1990 and 2019, which are included in the model simulations in section 5. The influence of these volcanic eruptions on the stratosphere and climate is analysed in section 6. At the end of section 6 as well as in the final discussion (section 7), the results are discussed in a wider context.

## 2 Satellite observations

To generate the input data from volcanic eruptions for our simulations, we analysed satellite data sets from two instruments on the European Environmental Satellite (ENVISAT) that was launched on 1 March 2002 and lost signal on 8 April 2012, namely MIPAS ($SO_2$ data) and GOMOS (aerosol extinction data). Furthermore, the OSIRIS instrument on board the Odin satellite was used to provide additional aerosol extinction data for the period up to 2019. For the period before 2002, we used the SAGE II instrument for aerosol extinction data. The data processing is described in section 4. Some examples for eruptions where simultaneous observations from all these instruments or at least 3 were available for cross validation are presented in the Appendix B.

### 2.1 Michelson Interferometer for Passive Atmospheric Sounding (MIPAS)

MIPAS was a mid-infrared emission spectrometer on board the ENVISAT satellite. MIPAS scanned the limb, thereby analysing the infrared radiation emitted by the Earth's atmosphere at different tangent altitudes (Fischer et al., 2008).

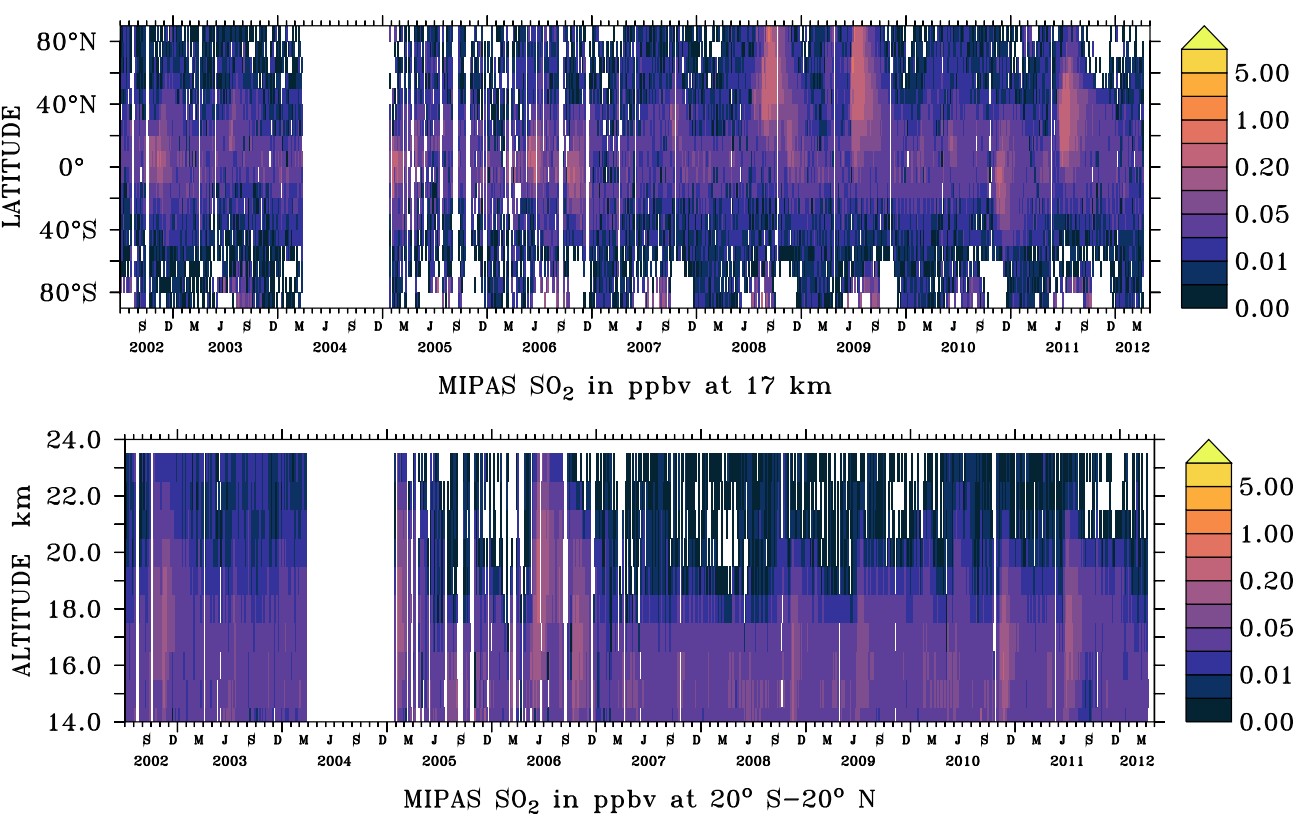

**Figure 1.** The volume mixing ratios of $SO_2$ in ppbv as derived from the MIPAS instrument (Höpfner et al., 2015). The dataset spans the period 1 July 2002–8 April 2012 (5-day averages). Horizontal distribution at 17 km altitude (top) and vertical distribution for tropical regions $20°$ S–$20°$ N (bottom). White: no data.

The atmospheric spectra ranging from 4.15 $\mu$m to 14.6 $\mu$m are inverted to provide vertical profiles of temperature and volume mixing ratios of more than 25 different trace species, like the sulfate aerosol precursor gases $SO_2$ and OCS (Glatthor et al., 2015, 2017; Höpfner et al., 2013, 2015), as well as $H_2O$, ozone ($O_3$), methane ($CH_4$), nitrous oxide ($N_2O$), nitrogen dioxide ($NO_2$) and nitric acid ($HNO_3$), among others.

Vertical $SO_2$ profiles (Figure 1) from the MIPAS $SO_2$ single profile retrieval (Höpfner et al., 2015) were used to identify plumes of volcanic eruptions. We utilised a gridded dataset from these retrievals with a three-dimensional sampling of $60°$ longitude, $10°$ latitude, 1 km altitude with a vertical coverage of 10 km to 23 km and a temporal averaging of five days. The lower altitude limit varies with the tops of clouds in the troposphere, especially in tropical regions.

The typical estimated random uncertainty for a single measurement of a volume mixing ratio profile is estimated to be 70–100 pptv. For the gridded dataset used here, systematic uncertainties are more important. These were estimated to be 10–75 pptv (10–180 %) under background conditions of the UTLS and 10–110 pptv (10–75 %) under volcanic influence (Höpfner et al., 2015).

## 2.2 Global Ozone Monitoring by Occultation of Stars (GOMOS)

The Global Ozone Monitoring by Occultation of Stars (GOMOS) instrument on ENVISAT operates based on the principle of stellar occultation. GOMOS provides data on stratospheric aerosol extinction as well as $O_3$, $NO_2$, nitrogen trioxide ($NO_3$) and air density (Kyrölä et al., 2010). The principle of stellar occultation is described in detail in Bertaux et al. (2010). In short, this self-calibrating sounding method scans the atmosphere by pointing to a star during its set or rise. The measured spectra vary with the tangent altitude due to the absorption and scattering of light by the different atmospheric species along the line of sight.

In a first step, the GOMOS inversion algorithm determines the slant column density of gaseous species and the slant aerosol optical depth along the optical path (Vanhellemont et al., 2004). This process makes use of reference absorption spectra of the main absorbing species (such as the ones provided by the MPI-Mainz UV/VIS Spectral Atlas of Gaseous Molecules of Atmospheric Interest (http://satellite.mpic.de/spectral_atlas) and extinction cross-section values representative for aerosols. Also, it requires the removal of the contribution of Rayleigh scattering by neutral air, which has to be carefully estimated, because satellite measurements cannot discriminate the contributions of neutral air density and very small particles compared to the wavelength (e.g., new particles arising from the conversion of $SO_2$ to fresh aerosol particles), respectively. In a second step, vertical density profiles of the target gas species and vertical profiles of the aerosol extinction coefficient are obtained from the slant quantities (Bertaux et al., 2010).

GOMOS uses four spectrometers providing measurements at wavelengths from the UV-visible to the near-IR range in four spectral regions: 248 nm–371 nm, 387 nm–693 nm, 750 nm–776 nm, and 915 nm–956 nm (Robert et al., 2016). As the original inversion algorithm (the operational algorithm IPF) was poorly effective for the retrieval of the aerosol extinction coefficient and only one extinction channel was obtained at the reference wavelength of 550 nm (Vanhellemont et al., 2010), a new retrieval algorithm called AerGOM was designed (Vanhellemont et al., 2016; Robert et al., 2016) in order to improve the spectral inversion. The main changes brought to AerGOM concern a change in the retrieval strategy where all atmospheric contributions

are retrieved all together instead of one by one, a revision of the parameterisation of the aerosol spectral dependence, and a more accurate estimate of the scattering cross-section by air. Also, the cross-section spectra for the gaseous species were revised using up-to-date reference spectra (Bingen et al., 2019). AerGOM provides the spectral dependence of the aerosol extinction coefficient between about 350 nm and 750 nm.

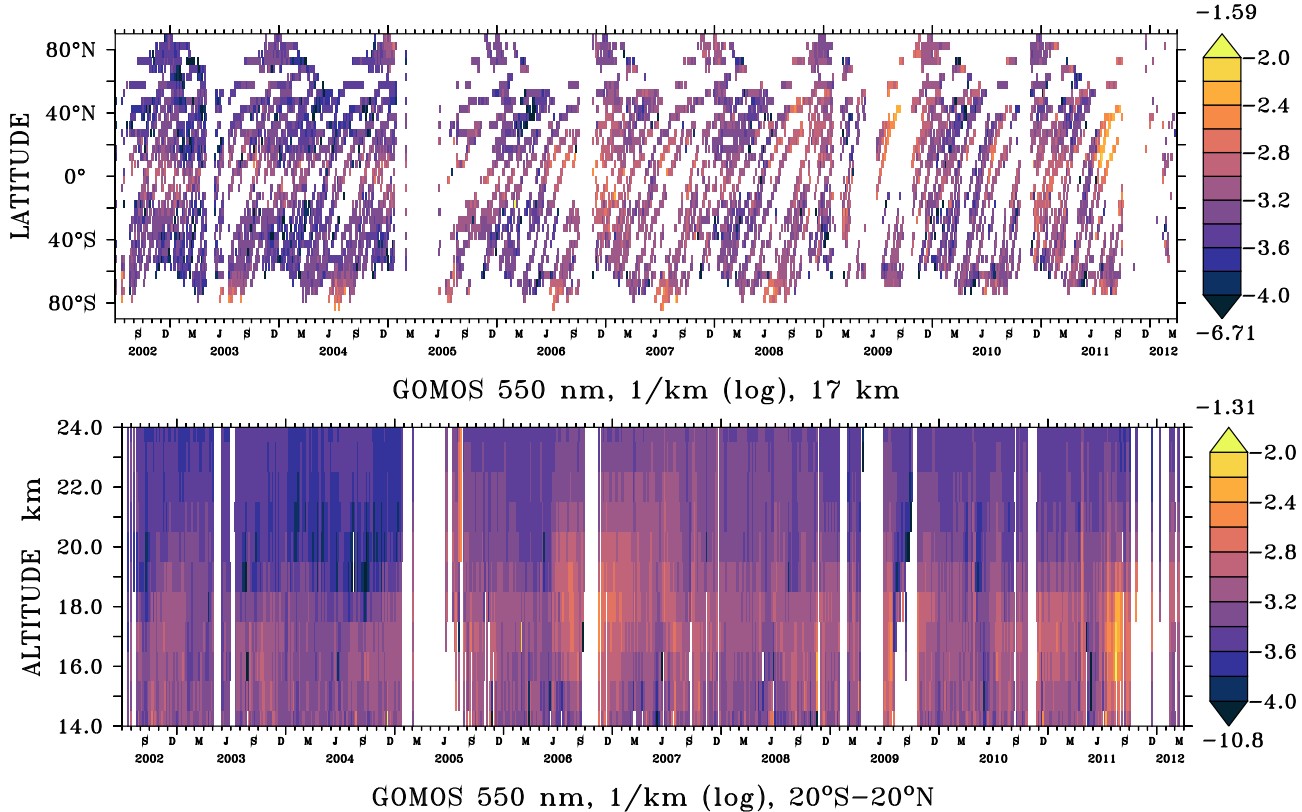

**Figure 2.** The decadal logarithm (log) of the aerosol extinction coefficient (1/km) as derived from the GOMOS instrument data v.3.00 at **550 nm wavelength** from Bingen et al. (2017). The dataset spans the period 15 April 2002–8 April 2012 (5-day averages). Horizontal distribution at 17 km altitude (top) and vertical distribution for tropical regions 20° S–20° N (bottom). Maximum and minimum values appear above (yellow) and below (dark blue) the colour keys, respectively. White: no data.

The typical extinction uncertainty exhibits large variability in function of the star parameters (from about 5–15 % in the most favourable cases of bright, hot stars, to about 40–70 % in the less favourable cases of dim, cold stars) (Bingen et al., 2017). A full validation of AerGOM, version 1.0, is presented by Vanhellemont et al. (2016). A main factor influencing the uncertainty is the weakness of the star signal, which is alleviated by the high measurement rate made possible by the abundance of stars. The large variability in magnitude and temperature of the occultated stars also significantly influences the measurement uncertainty
(Robert et al., 2016).

From AerGOM, climate data records were processed for use in chemistry-climate models (Bingen et al., 2017), and these are the data sets used in the present study. Figure 2 and Figure 3 show the aerosol extinction from the GOMOS instrument at wavelengths of 550 nm (Figure 2) and 750 nm (Figure 3), respectively. In both cases, a gridded aerosol extinction dataset is used (CCI-GOMOS dataset in version 3.00, see Bingen et al. (2017)).

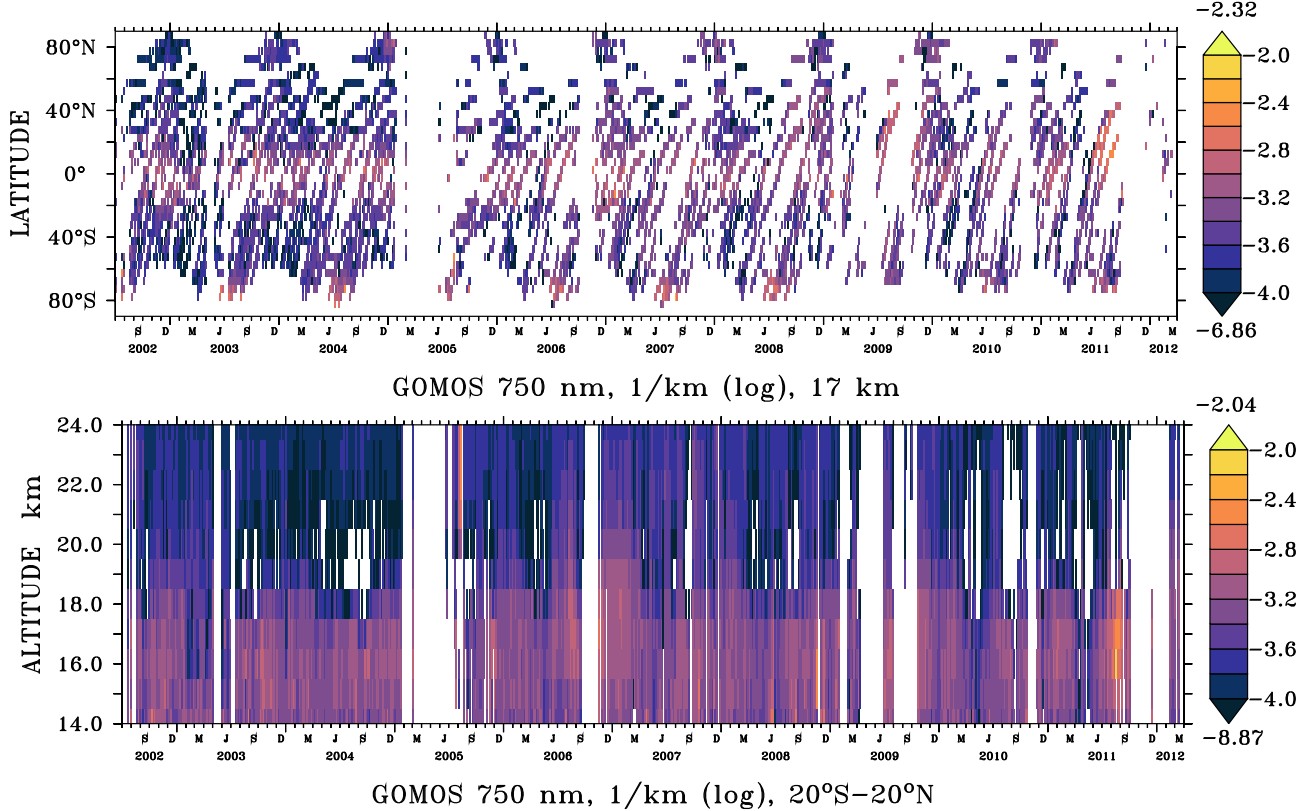

**Figure 3.** As Figure 2 but for **750 nm wavelength**.

The resolution of the CCI-GOMOS dataset was optimized to a grid of 5° latitude by 60° longitude and a time resolution of five days. This choice is made possible by the high measurement rate and is more suitable for describing the aerosol distribution than zonal monthly means, because it allows detection of the signature of aerosol patterns with a lifetime of as short as a week (e.g., medium-sized volcanic eruptions).

### 2.3   Optical Spectrograph and InfraRed Imaging System (OSIRIS)

The dataset from the Optical Spectrograph and InfraRed Imaging System (OSIRIS) allowed us to extend the time series beyond April 2012, after which the signal of the ENVISAT satellite was lost. OSIRIS is a limb scatter instrument, which was launched on board the Odin satellite on 20 February 2001 and is still operating today.

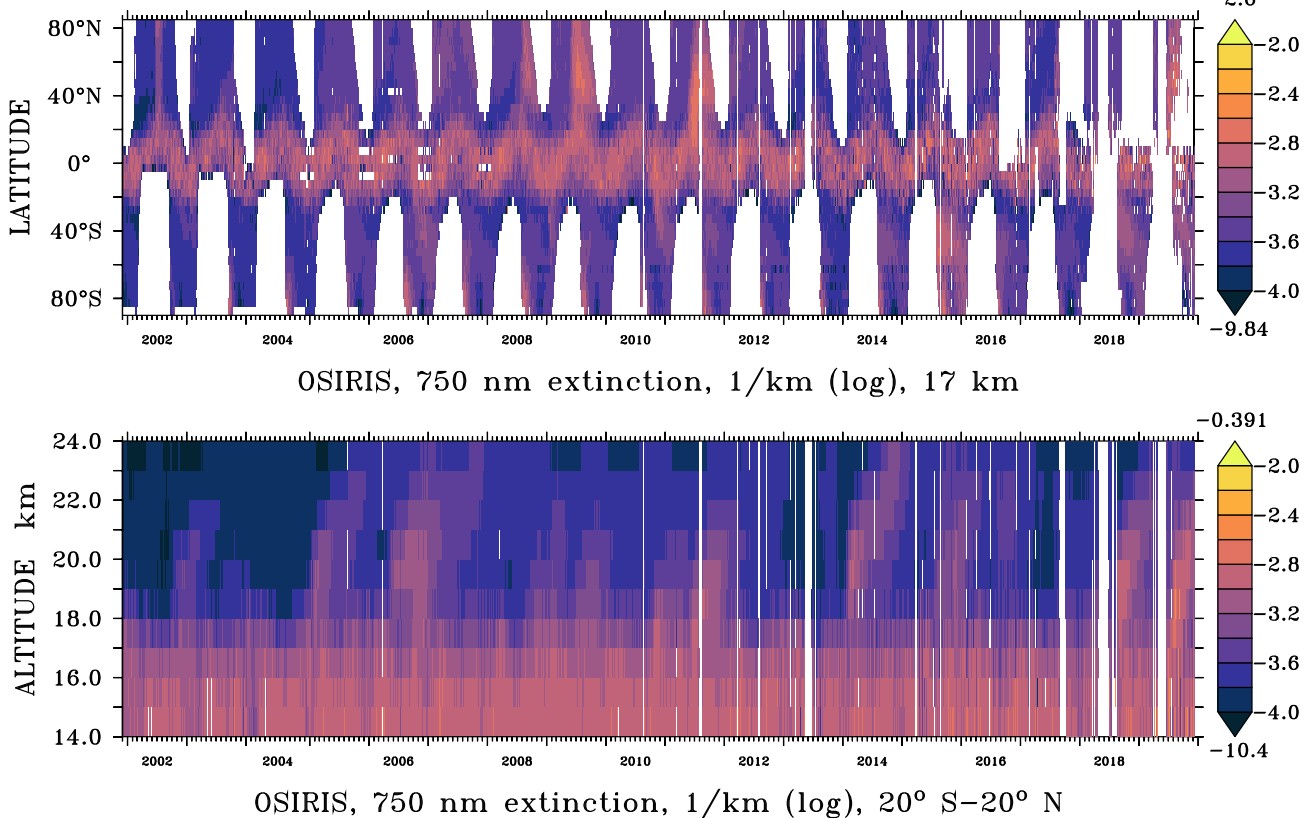

**Figure 4.** The logarithm log(1/km) of the aerosol extinction as derived from the OSIRIS instrument at **750 nm wavelength** by Bourassa et al. (2012a) and Rieger et al. (2019). The dataset spans the period 1 December 2001–December 2019 (5-day averages). Horizontal distribution at 16.5 km altitude (top) and vertical distribution for tropical regions 20° S–20° N (bottom). Maximum and minimum values appear above (yellow) and below (dark blue) the colour keys, respectively. White: no data.

OSIRIS performs limb scans of atmospheric radiance spectra at wavelengths from the UV to the near-IR ranges (274 nm–810 nm) (Bourassa et al., 2012a). To obtain the vertical profiles of aerosol extinction at altitudes from 10 km to 35 km (Rieger

et al., 2015), the aerosol scattering properties are calculated with a refractive index of $1.427+i7.167\times10^{-8}$ using Mie theory at 750 nm wavelength and a sulfate concentration of 75 % $H_2SO_4$ and 25 % $H_2O$ (Rieger et al., 2018).

For this study, the OSIRIS version 5.10 aerosol retrieval was used until October 2017 and the version 7.1a afterwards (for details see Rieger et al. (2019); Bourassa et al. (2012a)). OSIRIS provides coverage from 82° S–82° N over the course of the year. Extinction is retrieved where the tangent point is illuminated, which is primarily in the summer hemisphere (see Figure 4).

The grid resolution is 1 km altitude, 5° latitude and 30° longitude with 5-day-averaged time intervals.

The total uncertainty is about 10–15 % in the aerosol layer between 15–30 km, where the sensitivity of the measurements decreases with increasing optical depth. Due to measurement noise, the uncertainty dominates the signal above 30 km and in

the troposphere (Rieger et al., 2015). At altitudes near and below the tropopause, the OSIRIS measurements are sensitive to clouds that may be interpreted as elevated aerosols. This is likely contributing to larger background extinction values measured below approximately 17 km in the tropics, as can be seen in Figure 4 (bottom), and the uncertainty is higher.

## 2.4 Stratospheric Aerosol and Gas Experiment II (SAGE II)

SAGE II was a solar occultation instrument that performed measurements during sunrise and sunset. The SAGE II aerosol extinction measurements on board the Earth Radiation Budget Satellite (ERBS) started in October 1984 and ended in August 2005. This data set is important for the model setup before the ENVISAT period starting in 2002. The gridded aerosol extinction is derived from the V7.00 Level2 profiles provided by the Earth Observing System Data and Information System of NASA (EOSDIS) database (Figure 5).

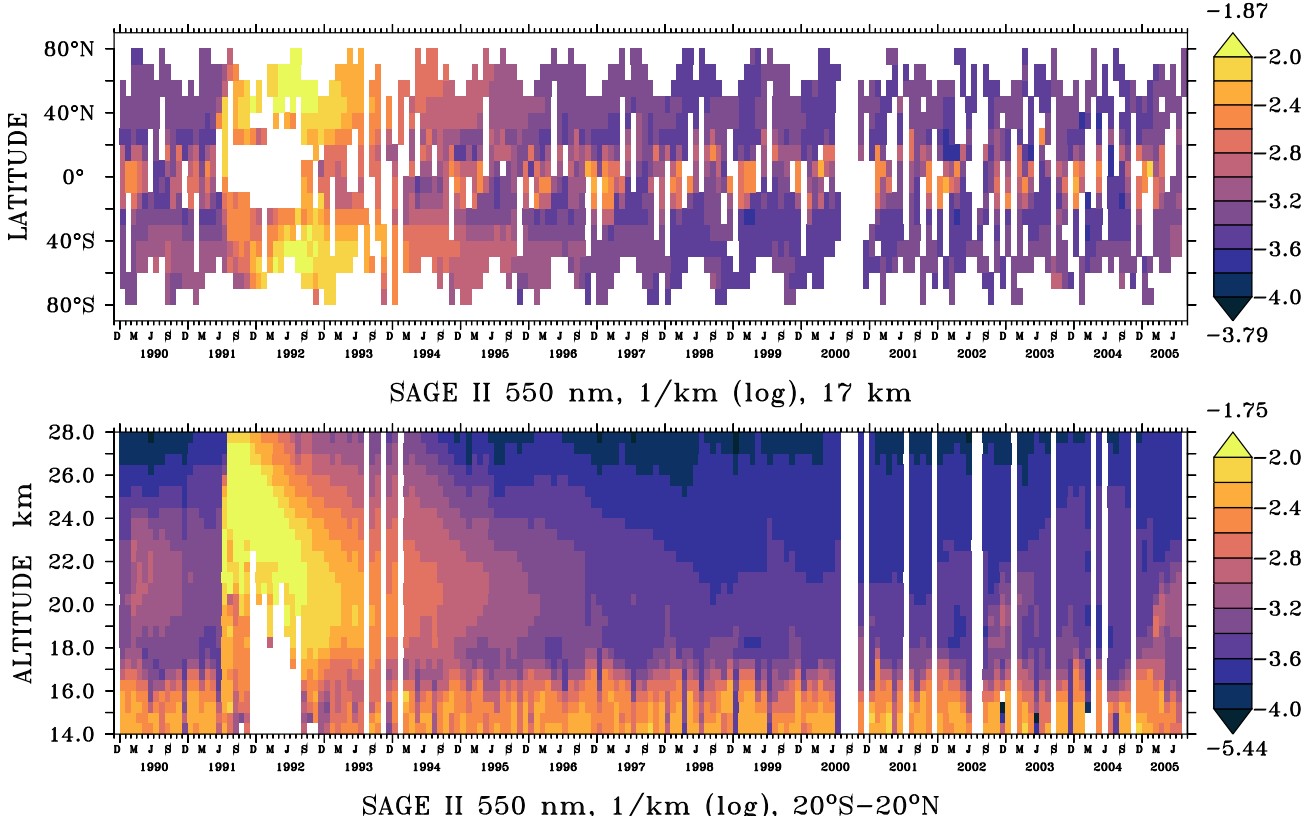

**Figure 5.** The logarithm of the extinction coefficient (1/km) of the SAGE II instrument from Thomason et al. (2008). The dataset spans the period January 1990–August 2005 (Monthly). Horizontal distribution at 16.75 km altitude (top) and vertical distribution for tropical regions 20° S–20° N (bottom). Maximum and minimum values appear above (yellow) and below (dark blue) the colour keys, respectively. White: no data.

The gridded SAGE II dataset used in the present study provides a near-global coverage with latitudes between $80°$ N to $80°$ S, with a horizontal grid resolution of $60°$ in longitude, $10°$ in latitude and a vertical resolution of $0.5\,km$ between $13\,km$ and $30\,km$ altitude. SAGE II measured in occultation, thus, its measuring principle is similar to that of GOMOS. The two main differences between GOMOS and SAGE II, are that the latter used the sun as light source, which results in a much better signal-to-noise ratio. On the other hand, its measurement rate is much lower than that of GOMOS, since only two measurements (one sunrise and one sunset) are possible per orbit, so that a near-global coverage is achieved in about one month.

Measurements occur at seven wavelengths between $386\,nm$ and $1020\,nm$. The vertical profiles of $O_3$, $NO_2$ and water vapor are provided as well as aerosol extinction coefficients at four wavelengths (386, 452, 525 and $1020\,nm$) from the middle troposphere to the upper stratosphere.

After the large eruption of Mount Pinatubo in 1991, "saturation" effects at lower altitudes were observed in the profiles for more than one year, meaning that the aerosol load was so high that the light signal received by the instrument was below the limit of detection. This effect corresponds to the large white areas for 1991 and 1992 in Figure 5. Yellow pixels around $14$–$16\,km$ correspond to measurements contaminated by clouds, increasing the optical depth in the upper troposphere/lower stratosphere (UTLS) region on the lower panel of Figure 5. The perturbations by convective clouds occur mostly over the West Pacific and were excluded in the procedure for estimating the $SO_2$ injections. The data gaps in the year 2000 were caused by an instrument failure causing SAGE II to be switched off for several months.

The uncertainty of the operational SAD (surface area density) product during background periods is affected by several parameters, including the lack of sensitivity to particles with radii smaller than $100\,nm$, the number of degrees of freedom indicated by the averaging kernels of the aerosol extinction at different wavelength channels and the temperature profile used in the data processing (Thomason et al., 2008).

## 3   Description of the setup for the EMAC Model

The simulations in this study were performed with EMAC (ECHAM5/MESSy Atmospheric Chemistry), a coupled atmospheric circulation model consisting of the $5^{th}$ generation of European Centre Hamburg general circulation model (ECHAM5) (Roeckner et al., 2003, 2006) and the Modular Earth Submodel System (MESSy) (Jöckel et al., 2005, 2006, 2010).

The model simulations performed by Bingen et al. (2017) and Brühl et al. (2018) in the period from 2002 to 2012 were extended in this study to 1990 to 2019. For these model simulations, a higher horizontal resolution T63 ($1.87° \times 1.87°$), instead of T42 ($2.81° \times 2.81°$) in Bingen et al. (2017), was chosen. As we focus on the stratosphere, the middle atmosphere version L90 with 90 layers up to $0.01\,hPa$ ($\sim 80\,km$) and high vertical resolution in the lower stratosphere was needed (Giorgetta et al., 2006). The temperature and the dynamics above the boundary layer are nudged to the meteorological ERA-Interim reanalysis data of the European Centre for Medium-Range Weather Forecasts (ECMWF) up to about $100\,hPa$, while the sea surface temperatures (SST) and sea ice are prescribed using ECMWF data (more details in Jöckel et al. (2006)).

As EMAC is a very complex chemistry climate model it contains many submodels and functions which are essential for running the simulations but are not directly related to the sulfur cycle, these are mentioned in Appendix A. In this section we focus on the sulfur cycle and aerosol.

The plumes of outgassing volcanic $SO_2$ emissions (Diehl et al., 2012) are imported via the OFFEMIS submodel as 3-D field volume emission fluxes (Kerkweg et al., 2006b). The exchange of dimethyl sulfide (DMS) between the air-sea interface of the ocean and the atmosphere is simulated by the AIRSEA submodel (Pozzer et al., 2006).

The gas- and aqueous-phase chemistry in the troposphere and stratosphere is calculated interactively with the CAABA/-MECCA (Chemistry As A Boxmodel Application/Module Efficiently Calculating the Chemistry of the Atmosphere) submodel (Sander et al., 2011). Specifically, the chemically generated $SO_2$ is calculated from fluxes of sulfate precursor gases and further transformed to $H_2SO_4$ (Brühl et al., 2018), together with the emitted $SO_2$. OH and ozone are fully interactive. CAABA/MECCA is also coupled to the Multiphase Stratospheric Box Model (MSBM) for heterogeneous reactions on sulfate aerosols and Polar Stratospheric Clouds (PSCs) (Jöckel et al., 2010) to allow for feedback on ozone. The uptake and oxidation of tracers is considered by the SCAVenging submodel for both liquid and mixed phase clouds (Tost et al., 2006a), also including the aqueous sulfur oxidation of $SO_2$ to $SO_4^{2-}$.

For parametrisation of aerosol microphysical processes, we used the Global Modal-aerosol eXtension (GMXe) aerosol module (Kerkweg, 2005; Stier et al., 2005; Vignati et al., 2004) and we described aerosol species using four soluble and three insoluble interacting lognormal aerosol modes. The original mode boundaries of the aerosol size distribution from Pringle et al. (2010) were adapted for this setup to volcanic aerosol conditions in the stratosphere as shown in Table 1 to avoid overly rapid sedimentation of coarse aerosol particles after big volcanic eruptions. The nucleation of new particles consists only of completely soluble sulfate aerosols and is calculated by the parametrisation used by Vehkamäki et al. (2002). Further, the evaporation of liquid sulfate particles back to the gas phase in the middle stratosphere is possible in the model.

| Mode boundaries | Pringle et al. (2010) | $\sigma$ | Brühl et al. (2018) | $\sigma$ |
|---|---|---|---|---|
| Nucleation mode: soluble | $<10\,nm$ | 1.69 | $1\,nm–12\,nm$ | 1.59 |
| Aitken modes: soluble & insoluble | $10\,nm–100\,nm$ | 1.69 | $12\,nm–140\,nm$ | 1.59 |
| Accumulation modes: soluble & insoluble | $100\,nm–1\,\mu m$ | 1.69 | $140\,nm–3.2\,\mu m$ | 1.49 |
| Coarse modes: soluble & insoluble | $>1\,\mu m$ | 2.2 | $>3.2\,\mu m$ | 1.7 |

**Table 1.** Diameters of aerosol mode boundaries in the GMXe submodel for tropospheric (Pringle et al., 2010) and volcanic stratosphere conditions (Brühl et al., 2018), including the corresponding mode distribution width ($\sigma$).

The AERosol OPTical properties in the model are calculated online with the AEROPT submodel (Dietmüller et al., 2016) and are coupled to the GMXe submodel. The resulting extinction coefficient is given at wavelengths of 350, 550, 750 and 1025 nm for comparison with GOMOS, OSIRIS and SAGE. Finally, the aerosol optical properties like wavelength- dependent particle extinction cross section, single scattering albedo, and asymmetry parameter for each aerosol mode from AEROPT (Dietmüller et al., 2016) are used in the radiation scheme as input for the radiative transfer calculations and to calculate the AOD. The

influence of stratospheric aerosol on instantaneous radiative forcing and heating is calculated online for diagnostic purposes
(for details see Dietmüller et al. (2016)). Via multiple calls of the RAD submodel in one simulation, these quantities are calculated online from the difference of fluxes or heating rates for the cases with stratospheric aerosol above the calculated tropopause (in earlier studies only above 100 hPa) and without any aerosol (Brühl et al., 2015), additionally to the call with full aerosol used for the interaction with dynamics.

It is not possible to separate volcanic aerosol from the background in these model simulations.

## 4  Volcanic Sulfur Emission Inventory

In a previous inventory of volcanic eruptions based on $SO_2$ vertical profiles from MIPAS, the aerosol instanteneous radiative forcing from 2002 to 2011 was estimated by simulating the evolution of $SO_2$ in the atmosphere reported by Brühl et al. (2015) and by improving the resulting time series using aerosol measurements from Bingen et al. (2017). The results of these simulations showed that significant discrepancies remained with respect to radiative forcing estimated from measurements
(Brühl et al., 2018).

In this work, we further improve the volcanic sulfur emission inventory by analysing additional satellite data sets (section 2) and by including all identified relevant eruptions between 1990 and 2019. To derive the volcanic 3D $SO_2$ perturbation from MIPAS we normally select the five day interval at the time of the eruption and the following one. For medium size eruptions up to 6 consecutive intervals are used to correct for saturation effects or artifacts from the applied cloud clearing scheme in case
of ash (Höpfner et al., 2015). A background of about 10 pptv, the typical value originating from OCS oxidation, is subtracted. In some cases this value can be larger because of remnants of a previous volcanic event. For MIPAS sometimes corrections in the order of up to 30% were necessary because of gaps (containing zero values) to be consistent with the total injected $SO_2$ mass derived from MLS (Microwave Limb Sounder) or OMI (Ozone Monitoring Instrument). Here the corrected values serve as reference for $SO_2$ derived from the instruments measuring extinction.
The GOMOS dataset is very important to compensate for data missing from the MIPAS instrument (subsection 2.1), where several important eruptions in 2004 and 2006 could not be identified (Bingen et al., 2017). An appropriate comparison of $SO_2$ mixing ratio measurements from MIPAS and of the aerosol extinction from GOMOS requires consideration of a time shift of one to two weeks as a result of the particle formation from the gas phase. We typically select a 10-day period beginning about a week after an eruption in the tropics. For higher latitudes the selected period is later and longer, taking into account the longer
conversion time (due to less OH). The $SO_2$ mixing ratio perturbation $\Delta VMR$ is derived from the extinction perturbation $\Delta\beta_{ext}$ (750nm) as in Equation 1 using a constant ratio between model calculated sulfate concentration and its share on extinction in the lower stratosphere of low latitudes:

$$\Delta VMR = 1.2 \times 10^{12} \frac{\Delta\beta_{ext}}{\rho} f \tag{1}$$

$\rho$ is the altitude dependent air density in molecules/$cm^3$, $f$ an empirical factor which equals 1 for sufficient data coverage (examples and more details see Appendix B). We assume that the spatial patterns of the perturbation of extinction and sulfate are the same as for $SO_2$. A similar technique is used for OSIRIS and can be used for SAGE II data.

If data gaps cause a shift of the time period away from the maximum perturbation or a low bias in the zonal average due to the zero values in the gaps at some longitude bin or in a period, a correction factor $f > 1$ based on comparison of total injected $SO_2$ mass with the one taken from other satellite data is applied in Equation 1. Correction factors up to 2 have to be applied in some cases because of data gaps, incomplete profiles (both containing zero values) or for high latitudes (examples see Appendix B). An extreme case is the eruption of Calbuco, with a correction factor of 3 for removal processes, because of a shift of three months due to a big data gap. To estimate the factor in this "worst" case, we iterated calculated extinctions to agree with OSIRIS and also used observations and assumptions by Vernier et al. (2016) like the decay of extinction by sulfate with time over 4 months.

On the other hand, the factor can be as small as 0.5 to account manually for sulfate remnants of eruptions occurring 2-4 weeks before the date of the eruption to be analysed or for cloud perturbations. The factors $f$, together with the used time periods, are provided in the electronic supplement (Table S1 for OSIRIS and Table S2 for GOMOS).

$SO_2$ column data from the OMI, Total Ozone Mapping Spectrometer (TOMS), Ozone Mapping and Profiler Suite (OMPS) and other nadir instruments were used to verify the consistency of the data and to fill in data gaps, marked as white areas in all satellite images. Especially in 2018 and 2019 the OSIRIS data are so sparse that constraints from instruments like OMPS-LP (Zawada et al., 2018) or analogues events of previous years have to be superimposed for some eruptions. These eruptions are marked with 'T' in Table 2. For the marked events in 2006 and 2007 data from OMI helped to identify them because sparse MIPAS and GOMOS data.

For SAGE II in most cases the $SO_2$ mixing ratio is derived using the parameterisation of Grainger et al. (1995) which converts SAD to volume density as first step. We use pressure and temperature provided to convert from mass density to a volume mixing ratio, assuming that observed sulfate is produced from injected $SO_2$ some weeks ago. With this method it is easier to correct for cloud contamination than by using the extinction directly as above for the other instruments.

Case studies for 3 events, comparing $SO_2$ results from the different satellites and the different conversion methods are presented in Appendix B.

The amount of sulfur emitted by each eruption is calculated by integration over the three-dimensional $SO_2$ perturbation plumes, excluding tropospheric emissions below 12 km at high latitudes, 13 km at mid-latitudes, and 14 km at low latitudes. The latter is selected to include possible convective transport from the upper troposphere into the stratosphere in the tropics. The limits in mid and high latitudes above the mean tropopause were selected to exclude cloud perturbations by frontal systems or uncertain satellite data. This can lead to an underestimate of injected mass in some cases. The plumes do not cover the whole globe, they are always in a latitude range derived manually from the satellite data. In case of multiple events the integration over the perturbed area is split considering the mean wind and consistency with nadir observations for fine tuning of the latitude and longitude boundaries. An example is shown in Appendix B.

Geological information and additional observations on plume heights were received from the Global Volcanism Program, Smithsonian Institution (https://volcano.si.edu/). Their reports several times indicate that even VEI 2 eruptions (volcanic explosivity index) can reach the upper troposphere (or lower stratosphere which confirms the satellite observations).

The resulting volcanic emission inventory is presented in Table 2 and provides the injection time into the model, the coordinates of the ejected plume and the amount of emitted $SO_2$. Each volcano is identified by its name if available, or by the concerned region if the name is unknown. The altitudes and latitudes indicated in the table correspond to the locations of the maximum $SO_2$ mixing ratios of the volcanic plumes. The longitudes refer to the locations of the volcanoes, because the plumes have been moved by the zonal winds during the time lag between eruption and observation. In the cases of OSIRIS, SAGE and GOMOS this shift can easily be 100 degrees. The entries in the table might be used for a point source approach.

It should be noted that the date of the volcanic eruption can differ by a few days from the date of injection in the model simulation, because the temporal resolution of the data sets is five days (or weeks in the SAGE period). In a lot of cases, more than one eruption is found in the same five day interval. In such a case, all eruptions are listed in the same line. Several time the used time period for extraction has to be extended because of data gaps, which increases the uncertainty and complicates the identification of the right volcano. In such a case, the name of the most probable volcano is tagged with a "?". If the $SO_2$ emissions of two volcanoes cannot be separated with certainty, both are indicated with a "+" in the same line. This uncertainty is frequent in the Republic of Vanuatu, an island country located in one of the most volcanically active regions in the South Pacific, referred to as "Vanuatu" in Table 2.

When comparing the $SO_2$ emissions reported here with those of Carn et al. (2016), it should be noted that Carn et al. (2016) makes use of total $SO_2$ emissions, including rapidly removed tropospheric $SO_2$, while the present study only takes into account the long-lived, climate-relevant stratospheric fraction of the emitted $SO_2$. A comparison to Carn et al. (2016) and Mills et al. (2016) of the injected volcanic $SO_2$ masses per year is presented in Appendix C2.

| Volcano or region | Time | Latitude (°) | Longitude (°) | Altitude (km) | SO$_2$ (kt) | Instrument |
|---|---|---|---|---|---|---|
| Kelut | 11 Feb 1990 | -8 | 112 | 16, 22 | 410 | S |
| Gamalama | 25 Apr 1990 | 0 | 127 | 16 | 96 | S |
| Raung (?) | 25 Jul 1990 | -8 | 115 | 16 | 63 | S |
| Pacaya + | 16 Sep 1990 | 15 | -90 | 16 | 10 | S |
| Sabancaya | 6 Oct 1990 | -16 | -72 | 17 | 75 | S |
| Papua + | 12 Jan 1991 | -4 | 145 | 17 | 88 | S |
| Fernandina | 20 Apr 1991 | 0 | -92 | 16 | 118 | S |
| Pinatubo | 16 Jun 1991 | 15 | 120 | 23 | 16942 | S |
| Hudson | 10 Aug 1991 | -46 | -73 | 18 | 1276 | S |
| Cerro Negro | 10 Apr 1992 | 12 | -87 | 22 | 18 | S |
| Spurr | 28 Jun 1992 | 61 | -152 | 17 | 291 | S |
| Spurr | 19 Aug 1992 | 61 | -152 | 16, 18 | 298 | S |
| Spurr | 18 Sep 1992 | 61 | -152 | 17 | 187 | S |
| Lascar | 18 Apr 1993 | -23 | -68 | 22 | 376 | S |
| Langila, Galeras (?) | 30 Oct 1993 | -5, 1 | 145, -70 | 17 | 50 | S |
| Yasur? | 17 Mar 1994 | -16 | 165 | 16 | 80 | S |
| Rinjani, Nyamuragira, Central America | 6 Jul 1994 | -8, -1, 12 | 117, 30, -90 | 16 | 63 | S |
| Rabaul | 20 Sep 1994 | -4 | 150 | 18, 22 | 89 | S |
| Merapi, Ecuador | 23 Nov 1994 | -7, 1 | 110, -70 | 17, 17 | 48, 57 | S |
| Peru, Africa, Vanuatu | 15 Feb 1995 | -15, -1, -15 | -78, 30, 168 | 17, 16, 16 | 7, 43, 25 | S |
| Mexico + Soufriere Hills | 10 Aug 1995 | 16 | -98, -62 | 16 | 81 | S |
| Peru + Colombia, Rabaul | 10 Feb 1996 | -15, 5, -4 | -80, -80, 150 | 17, 16, 16 | 65, 96 | S |
| Soufriere Hills | 26 May 1996 | 16 | -62 | 16 | 53 | S |
| Soufriere Hills + Mexico, Rabaul | 18 Sep 1996 | 16, -4 | -62, -98, 150 | 16, 16 | 59, 28 | S |
| Nyamuragira, Manam | 3 Dec 1996 | -1, -5 | 30, 145 | 17, 17 | 45, 90 | S |
| Manam + Langila | 11 Feb 1997 | -5 | 145 | 17 | 107 | S |
| Popocatepetl | 1 Jul 1997 | 19 | -98 | 16 | 32 | S |
| Soufriere Hills, Philippines | 20 Oct 1997 | 16, 16 | -62, 121 | 15, 16 | 36, 20 | S |
| Soufriere Hills, Papua | 26 Dec 1997 | 16, -8 | -62, 150 | 16, 16 | 37, 22 | S |
| Tungurahua (?), Vanuatu | 2 Feb 1998 | -1, -16 | -78, 168 | 17, 16 | 98, 15 | S |
| Soufriere Hills | 4 Jul 1998 | 16 | -62 | 16 | 56 | S |
| Manam, Cerro Azul, Nyamuragira | 7 Oct 1998 | -5, 0, -1 | 144, -90, 30 | 17, 17, 16 | 28, 39, 19 | S |
| Guagua Pinch + Tungurahua, Vanuatu | 23 Jan 1999 | -1, -16 | -78, 165 | 17, 16 | 75, 49 | S |
| Cameroon | 31 Mar 1999 | 4 | 10 | 16 | 63 | S |
| Mayon, Colombia | 22 Jun 1999 | 13, 2 | 124, -80 | 17, 16 | 41, 46 | S |
| Soufriere Hills + | 24 Jul 1999 | 16 | -62 | 17 | 42 | S |

| Volcano or region | Time | Latitude (°) | Longitude (°) | Altitude (km) | SO$_2$ (kt) | Instrument |
|---|---|---|---|---|---|---|
| Ulawun, Tungurahua + Guagua Pichincha | 16 Nov 1999 | -5, -1 | 150, -78 | 17 | 31, 51 | S |
| Vanuatu, Nyamuragira, Tungurahua | 4 Feb 2000 | -16, -1, 0 | 165, 30, -78 | 17, 16, 16 | 33, 41, 12 | S |
| Mayon + Vanuatu, Tungurahua | 29 Feb 2000 | 13, -16, -1 | 124, 168, -78 | 16, 16, 16 | 25, 32 | S |
| Ulawun (+ Miyakejima) | 26 Sep 2000 | -5 | 150 | 17 | 42 | S |
| Nyamuragira, Mayon(?) | 13 Feb 2001 | -1, 13 | 30, 124 | 16, 18 | 47, 88 | S |
| Ulawun | 29 Apr 2001 | -5 | 150 | 16 | 41 | S |
| Mayon, Lopevi | 23 Jun 2001 | 13, -16 | 124, 168 | 16, 16 | 49, 22 | S |
| Tungurahua, Soufriere Hills | 7 Aug 2001 | 0, 16 | -78, -62 | 16, 16 | 29, 46 | S |
| Africa, Tungurahua + | 25 Sep 2001 | -1, 0 | 30, -78 | 16, 16 | 31, 47 | S |
| Tungurahua (+ Manam), Nyiragongo | 14 Jan 2002 | -1(-4), -1 | -78(144), 30 | 17, 15 | 83, 19 | S |
| Tungurahua (+ Africa) | 20 Mar 2002 | -1 | -78 (30) | 17 | 77 | S |
| *Nyamuragira* | 23 Jul 2002 | -1 | 30 | 15 | 23 | M |
| *Witori* | 2 Aug 2002 | -6 | 150 | 14 | 18 | M |
| *Ruang* | 26 Sep 2002 | 2 | 125 | 18 | 71 | M, G |
| *El Reventador* | 5 Nov 2002 | 0 | -78 | 17 | 77 | M, G |
| *Nyiragongo, Lokon* | 9 Jan 2003 | -1, 1 | 30, 125 | 15, 16 | 12, 10 | M, G |
| *Nyiargongo, Lokon (Rabaul?)* | 5 Mar 2003 | -5, 1 | 30, 125 | 17, 15 | 12, 13 | M, G |
| *Anatahan, Nyiaragongo, Ulawun* | 14 May 2003 | 16, -1, -5 | 143, 30, 150 | 16, 16, 17 | 9, 15, 6 | M |
| *Lewotobi, Kanlaon* | 13 Jun 2003 | -8, 10 | 123, 123 | 15, 15 | 9, 15 | M, G |
| *Soufriere Hills* | 13 Jul 2003 | 16 | -62 | 17 | 41 | M, G |
| *Gamalama, Japan* | 17 Aug 2003 | 1, 33 | 128, 131 | 16, 16 | 8, 7 | M, G |
| *Bezymianny or Klyuchevskoy* | 6 Sep 2003 | 56 | 160 | 14 | 8 | G |
| *Lokon, Soufriere Hills + Masaya* | 26 Sep 2003 | 2, 15 | 125, -62 | 16, 16 | 7, 5 | M, G |
| *Rabaul* | 10 Nov 2003 | -5 | 150 | 16 | 17 | M, G |
| *Rabaul* | 5 Dec 2003 | -5 | 150 | 16 | 13 | M, G |
| *Rabaul, Nyiaragongo?* | 9 Jan 2004 | -5, -1 | 150, 30 | 17, 15 | 11, 9 | M, G |
| *Langila, Nyiaragongo?* | 3 Feb 2004 | -5, -1 | 150, 30 | 17, 17 | 11, 3 | M, G |
| *Soufriere Hills* | 4 Mar 2004 | 10 | -62 | 17 | 22 | M, G |
| *Nyamuragira, Awu + Tengger* | 12 Jun 2004 | -1, 4, -8 | 30, 125, 112 | 17, 15 | 20, 18 | G |
| *Pacaya, Galeras* | 17 Jul 2004 | 15, 1 | -91, -77 | 17, 17 | 11, 11 | G |
| *Galeras* | 11 Aug 2004 | 1 | -77 | 16 | 15 | G |
| *Vanuatu, Rinjani + Kerinci* | 30 Sep 2004 | -16, -8, -2 | 168, 116, 101 | 15, 15, 17 | 7, 15 | G |
| *Manam, Soputan* | 30 Oct 2004 | -4, 1 | 144, 125 | 16, 16 | 8, 11 | G |
| *Manam, Nyiragongo* | 24 Nov 2004 | -4, -1 | 144, 30 | 17, 15 | 18, 11 | G |
| *Nyiaragongo, Reventador* | 4 Dec 2004 | 0, 0 | 30, -77 | 16, 16 | 19, 5 | G |

| Volcano or region | Time | Latitude (°) | Longitude (°) | Altitude (km) | SO$_2$ (kt) | Instrument |
|---|---|---|---|---|---|---|
| *Vanuatu, Soputan* | 24 Dec 2004 | -16, 1 | 168, 125 | 17, 15 | 15, 16 | G |
| *Manam* | 28 Jan 2005 | -4 | 144 | 18 | 130 | M, G |
| *Anatahan, (+)* | 3 Apr 2005 | 16 | 143 | 15 | 15 | M |
| *Anatahan, Soufriere Hills* | 23 Apr 2005 | 16, 16 | 143, -62 | 16, 16 | 21, 21 | M |
| *Anatahan, Fernadina, Vanuatu* | 18 May 2005 | 16, 0, -16 | 143, -91, 168 | 15, 15, 15 | 8, 11, 6 | M |
| *Anatahan, Santa Ana* | 12 Jun 2005 | 16, 14 | 143, -90 | 15, 15 | 12, 9 | M |
| *Anatahan, Soufriere Hills* | 12 Jul 2005 | 16, 16 | 143, -62 | 15, 15 | 14, 10 | M |
| *Anatahan, Raung* | 6 Aug 2005 | 16, -8 | 143, 113 | 15, 15 | 13, 20 | M |
| *Anatahan, Raung* | 16 Aug 2005 | 16, -8 | 143, 113 | 15, 15 | 14, 17 | M, G |
| *Santa Ana* | 5 Oct 2005 | 14 | -90 | 17 | 32 | M |
| *Sierra Negra, Dabbahu* | 25 Oct 2005 | -1, -13 | -91, 40 | 15, 15 | 16, 22 | G |
| *Karthala, Galeras* | 24 Nov 2005 | -10, -2 | 43, -80 | 16, 16 | 13, 11 | M, G |
| *Soputan, Lopevi* | 24 Dec 2005 | 1, -16 | 125, 168 | 16, 16 | 23, 13 | M, G |
| *Rabaul +* | 23 Jan 2006 | -5 | 152 | 16 | 25 | G |
| *Manam, Chile* | 4 Mar 2006 | -5, -40 | 144, -70 | 17, 16 | 58, 6 | G, M, T |
| *Cleveland* | 14 Mar 2006 | 53 | -170 | 13 | 8 | G, M |
| *Ecuador, Tinakula, Lascar* | 18 Apr 2006 | -5, -10, -23 | -78, 166, -68 | 17, 17, 17 | 13, 17, 3 | M |
| *Soufriere Hills* | 23 May 2006 | 16 | -62 | 19 | 125 | M, G, T |
| *Kanlaon* | 2 Jul 2006 | 10 | 123 | 20 | 42 | M |
| *Tungurahua, Rabaul* | 16 Aug 2006 | -2, -4 | -78, 150 | 19, 17 | 40, 20 | M, G, T |
| *Rabaul* | 10 Oct 2006 | -4 | 150 | 17 | 131 | M, T |
| *Ubinas, Vanuatu* | 25 Oct 2006 | -20, -20 | -70, 168 | 17, 15 | 8, 25 | M |
| *Ambrym* | 9 Nov 2006 | -10 | 160 | 17 | 27 | M, T |
| *Nyamuragira, Mexico* | 29 Nov 2006 | 5, 5 | 30, -90 | 17, 15 | 28, 21 | M, G, T |
| *Bulusan, Soputan, Vanuatu* | 24 Dec 2006 | 13, 1, -16 | 125, 125, 168 | 18, 16, 15 | 8, 8, 14 | M, G |
| *Karthala, Bulusan, Lascar, Shiveluch, Vanuatu* | 23 Jan 2007 | -10, 13, -23, 57, -16 | 43, 125, -68, 160, 168 | 17, 17, 15, 15, 15 | 5, 5, 6, 7, 5 | M, G, T |
| *Nevado del Huila, Karthala, Vanuatu* | 22 Feb 2007 | 0, -10, -16 | -70, 43, 168 | 16, 15, 16 | 8, 10, 8 | M, G, T |
| *Etna, Reventador, Ambrym* | 24 Mar 2007 | 38, 0, -16 | 15, -78, 160 | 15, 16, 17 | 8, 17, 14 | M, G, T |
| *Piton de la Fournaise, Reventador +* | 8 Apr 2007 | -20, 0 | 57, -80 | 16, 16 | 22, 11 | M, G, T |
| *Ulawun, Vanuatu, Nevado del Huila* | 3 May 2007 | -5, -25, 3 | 150, 160, -70 | 15, 15, 15 | 11, 5, 6 | M, G, T |
| *Papua, Kamchatka, Nyamuragira, Ubinas + Lascar* | 13 May 2007 | -10, 50, 0, -20 | 150, 150, 30, -75 | 16, 16, 16, 16 | 6, 1, 10, 6 | M, G |
| *Llaima, Vanuatu, Bulusan* | 23 May 2007 | -30, -15, 13 | -70, 160, 125 | 18, 15, 17 | 10, 6, 7 | M, G |
| *Soputan, Bezymianny, Telica* | 12 Jun 2007 | 1, 56, 13 | 125, 160, -87 | 16, 14, 15 | 13, 7, 9 | M, G |
| *Lengai, Mexico* | 2 Jul 2007 | 2, 20 | 29, -90 | 16, 15 | 14, 9 | M |

| Volcano or region | Time | Latitude (°) | Longitude (°) | Altitude (km) | SO$_2$ (kt) | Instrument |
|---|---|---|---|---|---|---|
| *Raung, Japan (+)* | 27 Jul 2007 | -5, 35 | 110, 130 | 15, 15 | 10, 10 | M |
| *Manda Hararo, Java* | 11 Aug 2007 | 12, -5 | 40, 115 | 17, 15 | 13, 14 | M, T |
| *Vanuatu, Mexico* | 20 Sep 2007 | -5, 20 | 180, -90 | 16, 16 | 8, 13 | M |
| *Jebel al Tair, Galeras* | 5 Oct 2007 | 16, 1 | 42, -80 | 16, 16 | 41, 8 | M, T |
| *Galeras, Jebel al Tair, Soputan* | 4 Nov 2007 | -2, 15, -5 | -80, 42, 110 | 16, 16, 16 | 7, 5, 8 | M, G |
| *Soputan or Krakatau,Galeras,Chikurachki* | 14 Nov 2007 | -5, -1, 50 | 110, -75, 155 | 16, 16, 15 | 9, 8, 10 | M |
| *Talang, Galeras* | 9 Dec 2007 | 0, 0 | 100, -75 | 16, 16 | 10, 12 | M |
| *Ulawun?* | 19 Dec 2007 | 1 | 150 | 17 | 17 | M, G |
| *Nevado del Huila, Llaima* | 3 Jan 2008 | 1, -35 | -75, -71 | 17, 15 | 26, 4 | M |
| *Galeras, Anatahan* | 23 Jan 2008 | -3, 15 | -80, 145 | 16, 16 | 14, 7 | M |
| *Tungurahua, Papua* | 12 Feb 2008 | -5, -5 | -80, 155 | 16, 17 | 13, 10 | M |
| *Batu Tara (+)* | 13 Mar 2008 | -5 | 125 | 16 | 26 | M, G |
| *Lengai, Andes, Kerinic* | 28 Mar 2008 | -5, 5, -2 | 36, -80, 101 | 16, 16, 16 | 6, 4, 7 | M |
| *Egon, Nevado del Huila* | 12 Apr 2008 | -5, 5 | 122, -76 | 15, 17 | 14, 9 | M |
| *Mexico, Ibu, Chaiten* | 27 Apr 2008 | 15, 1, -35 | -90, 125, -70 | 16, 16, 16 | 9, 11, 3 | M |
| *Mexico, Barren Island, Chaiten* | 12 May 2008 | 10, 10, -35 | -90, 90, -70 | 14, 16, 14 | 10, 14, 5 | M |
| *Soputan, Nicaragua/Costa Rica* | 16 Jun 2008 | 1, 1 | 125, -85 | 16, 16 | 26, 8 | M |
| *Okmok, Soputan* | 21 Jul 2008 | 53, 1 | -168, 125 | 16, 16 | 51, 27 | M |
| *Kasatochi* | 15 Aug 2008 | 52 | -175 | 17 | 273 | M, G |
| *Dallafilla, Nevado del Huila, Reventador* | 13 Nov 2008 | 14, 3 | 40, -78 | 17, 17 | 39, 28 | M |
| *Karangetang, Galeras, Japan* | 18 Dec 2008 | 3, 0, 30 | 125, -80, 130 | 17, 17, 15 | 15, 10, 9 | M, G |
| *Barren Island, Galeras* | 2 Jan 2009 | 10, 3 | 90, -80 | 17, 15 | 10, 10 | M |
| *Indonesia?, Galeras* | 27 Jan 2009 | -5, 0 | 100, -80 | 16, 16 | 12, 10 | M |
| *Galeras,Villarrica, Karangetang,Vanuatu* | 16 Feb 2009 | -2, -35, 3,-16 | -78, -75,100,168 | 16, 15, 16, 17 | 11, 6, 6, 7 | M |
| *Redoubt, Galeras* | 28 Mar 2009 | 60, 0 | -155, -75 | 13, 15 | 61, 43 | M |
| *Fernandina, Nyamuragira* | 12 Apr 2009 | 0, 0 | -90, 30 | 16, 16 | 12, 16 | M |
| *Galeras + Reventador* | 7 May 2009 | 0 | -75 | 15 | 25 | M |
| *Rinjani, Vanuatu, Reventador* | 22 May 2009 | -5, -15, 3 | 116, 165, -80 | 16, 16, 16 | 4, 4, 13 | M |
| *Sarychev, Manda Hararo* | 21 Jun 2009 | 48, 12 | 153, 40 | 16, 16 | 446, 82 | M, G |
| *Vanuatu, Mayon, Galeras* | 4 Oct 2009 | -15, 13, 2 | 165, 120, -80 | 17, 17, 17 | 4, 6, 10 | M |
| *Tungurahua, Hawaii, Vanuatu* | 19 Oct 2009 | 5, 20, - 16 | -76, -155, 165 | 16, 16, 16 | 7, 5, 5 | M, G |
| *Galeras, Karkar, Vanuatu* | 3 Dec 2009 | 0, -5, -16 | -78, 146, 165 | 17, 17, 17 | 12, 10, 4 | M |
| *Mayon, Nyamuragira, Vanuatu* | 2 Jan 2010 | 13, 0, -15 | 120, 30, 168 | 16, 16, 16 | 8, 8, 9 | M |
| *Turrialba, Vanuatu* | 17 Jan 2010 | 5, -15 | -82, 168 | 16, 16 | 9, 9 | M |
| *Soufriere Hills* | 16 Feb 2010 | 16 | -62 | 17 | 36 | M |
| *Arenal, Indonesia, Vanuatu* | 2 Apr 2010 | 9, 0, -16 | -84, 120, 168 | 15, 15, 15 | 14, 12, 5 | M |

| Volcano or region | Time | Latitude (°) | Longitude (°) | Altitude (km) | SO$_2$ (kt) | Instrument |
|---|---|---|---|---|---|---|
| *Tungurahua, Dukono, Vanuatu* | 2 May 2010 | -5, 2, -16 | -78, 128, 168 | 16, 16, 16 | 14, 10, 7 | M |
| *Pacaya, Ulawun, Sarigan* | 6 Jun 2010 | 15, -5, 16 | -91, 150, 145 | 17, 16, 15 | 27, 6, 4 | M |
| *Ulawun, Costa Rica, Miyakejima* | 16 Jul 2010 | -5, 15, 35 | 150, -87, 140 | 16, 16, 16 | 8, 13, 6 | M, G |
| *Karangetang, Nicaragua, Vanuatu* | 15 Aug 2010 | 3, 15, -16 | 125, -85, 168 | 16, 16, 16 | 12, 12, 6 | M |
| *Galeras, Sinabung* | 30 Aug 2010 | 5, 5 | -77, 100 | 16, 16 | 10, 12 | M |
| *Karangetang, Barren Island* | 4 Oct 2010 | 3, 12 | 125, 94 | 16, 16 | 20, 13 | M |
| *Merapi* | 8 Nov 2010 | -7 | 110 | 17 | 97 | M |
| *Tengger, Tungurahua, Chile* | 23 Dec 2010 | -8, -3, -40 | 110, -78, -75 | 17, 17, 17 | 16, 13, 8 | M |
| *Tengger* | 7 Jan 2011 | -8 | 110 | 16 | 24 | M |
| *Lokon, Planchon, Bulusan* | 26 Feb 2011 | 1, -35, 13 | 125, -75, 125 | 16, 15, 16 | 13, 4, 12 | M |
| *Karangrtang, Sangay, Planchon* | 23 Mar 2011 | 2, -2, -35 | 125, -78, -75 | 15, 15, 15 | 10, 10, 5 | M |
| *Galeras?, Karangetang* | 12 Apr 2011 | 5, 5 | -77, 128 | 16, 16 | 10, 9 | M |
| *Tungurahua, Dukono, Vanuatu* | 2 May 2011 | 2, 2, -16 | -78, 128, 160 | 16, 16, 15 | 13, 9, 5 | M |
| *Grimsvötn, Lokon* | 27 May 2011 | 65, 1 | -20, 125 | 14, 16 | 18, 27 | M |
| *Puyehue* | 11 Jun 2011 | -41 | -71 | 13 | 23 | G, M |
| *Nabro* | 21 Jun 2011 | 13 | 41 | 18 | 406 | M, G |
| *Soputan, Marapi* | 20 Aug 2011 | 1, 0 | 125, 100 | 18, 16 | 9, 3 | M, G |
| *Manam, Tungurahua* | 19 Oct 2011 | -4, -3 | 144, -78 | 16, 16 | 8, 8 | M |
| *Nyamuragira* | 18 Nov 2011 | -2 | 29 | 16 | 31 | M |
| *Gamalama, Nyamuragira* | 18 Dec 2011 | 1, -1 | 128, 29 | 16, 15 | 19, 13 | M |
| *Vanuatu, Nyamuragira* | 12 Jan 2012 | -16, -1 | 168, 29 | 16, 14 | 14, 12 | M |
| *Vanuatu, Nyamuragira* | 11 Feb 2012 | -16, -1 | 168, 29 | 17, 17 | 16, 15 | M |
| *Nevado del Ruiz, Marapi* | 12 Mar 2012 | -3, 0 | -76, 100 | 16, 17 | 12, 15 | M |
| Nyamuragira, Mexico | 7 Jun 2012 | -1, 20 | 29, -95 | 16, 15 | 30, 4 | O |
| Soputan, Nevado del Ruiz, Mexico | 27 Aug 2012 | 1, 5, 20 | 124,-76,-95 | 16, 16, 15 | 30, 15, 5 | O |
| Nyamuragira, Mexico, Peru | 14 Oct 2012 | -1, 20, -20 | 29, -95, -70 | 16, 16, 15 | 40, 15, 10 | O |
| Nyamuragira, Paluweh, Nevado del Ruizz | 7 Nov 2012 | -1, -8, 5 | 29, 122, -76 | 15, 16, 17 | 20, 30, 17 | O |
| Copahue, Lokon + | 22 Dec 2012 | -38, 1 | -71, 125 | 15, 17 | 10, 45 | O |
| Paluweh, Karkar | 3 Feb 2013 | -8, -5 | 122, 145 | 16, 17 | 25, 22 | O |
| Karkar, Vanuatu (+?) | 10 Mar 2013 | -5, -16 | 145, 168 | 17, 16 | 24, 20 | O |
| Rabaul, Nevado del Ruiz, Nyamuragira | 18 Apr 2013 | -3, 5, -1 | 150,-76, 29 | 17, 17, 16 | 40, 9, 20 | O |
| Mayon, Turrialba, Pavlof | 8 May 2013 | 13, 10, 55 | 124, -84, -162 | 17, 16, 14 | 35, 24, 6 | O |
| Rabaul, Mexico | 10 Jul 2013 | -3, 20 | 150, -95 | 16, 15 | 30, 15 | O |
| Pacaya | 15 Aug 2013 | 15 | -91 | 16 | 43 | O |
| Sinabung, Ubinas | 15 Sep 2013 | 3, -16 | 98, -71 | 17, 15 | 35, 8 | O |
| Merapi, Nyamuragira, Pacaya | 18 Nov 2013 | -7, -1, 15 | 110, 29, -91 | 17, 17, 15 | 30, 13, 8 | O |

| Volcano or region | Time | Latitude (°) | Longitude (°) | Altitude (km) | SO₂ (kt) | Instrument |
|---|---|---|---|---|---|---|
| Sinabung, Nyamuragira | 9 Dec 2013 | 3, -1 | 98, 29 | 17, 16 | 26, 15 | O |
| Sinabung + | 11 Jan 2014 | 3 | 98 | 16 | 29 | O |
| Kelut | 15 Feb 2014 | -8 | 112 | 20 | 170 | O |
| Merapi, Tungurahua | 27 Mar 2014 | -7, -1 | 110, -78 | 16, 16 | 31, 33 | O |
| Santa Maria, Semeru | 9 May 2014 | 15, -8 | -91, 113 | 16, 16 | 25, 39 | O |
| Sangeaang-Api | 31 May 2014 | -8 | 119 | 17 | 60 | O |
| Nyamuragira, Pavlof, Fuego, Dukono (Tungurahua) | 9 Jul 2014 | -1, 55, 14, 2 | 29, -162, -91, 128 | 16, 15, 15, 16 | 20, 10, 12, 20 | O |
| Rabaul, Fuego | 29 Aug 2014 | -3, 14 | 150, -91 | 16, 16 | 36, 20 | O |
| Nyamuragira | 11 Sep 2014 | -1 | 29 | 15 | 30 | O |
| Ontakesan | 27 Sep 2014 | 36 | 137 | 17 | 34 | O |
| Sinabung, Turrialba | 23 Oct 2014 | 3, 10 | 98, -84 | 17, 16 | 34, 17 | O |
| Fogo, Semeru, Ubinas | 24 Nov 2014 | 15, -8, -16 | -24, 113, -71 | 17, 17, 16 | 11, 33, 11 | O |
| Nevado del Ruiz, Nyamuragira, Vanuatu | 16 Dec 2014 | 5, -1, -16 | -76, 29, 168 | 15, 17, 16 | 8, 12, 21 | O |
| Nyamuragira, Vanuatu, Honga Tonga | 14 Jan 2015 | -1, -16, -21 | 29, 168, -175 | 16, 16, 15 | 21, 17, 13 | O |
| Vanuatu, Nyamuragira, Soputan | 16 Feb 2015 | -16, -1, 1 | 168, 29, 124 | 17, 16, 16 | 13, 13, 13 | O |
| Soputan, Nevado del Ruiz, Santa Maria, Villarrica | 8 Mar 2015 | 1, 5, 15, -39 | 125,-76,-91,-72 | 17, 16, 15, 15 | 14, 14, 8, 5 | O |
| Tungurahua?, Batu Tara? | 5 Apr 2015 | -1, -8 | -78, 124 | 17, 17 | 17, 22 | O |
| Calbuco | 25 Apr 2015 | -41 | -73 | 18 | 292 | O |
| Manam, Tungurahua? | 8 May 2015 | -4, -1 | 144, -78 | 17, 17 | 24, 25 | O |
| Wolf, Aira + Kuchinoerabujima | 26 May 2015 | 0, 32, 30 | -91, 131, 130 | 16, 15 | 63, 20 | O |
| Raung | 4 Jul 2015 | -5 | 110 | 17 | 27 | O |
| Cotopaxi, Raung, Suwanosjima, Manam | 14 Aug 2015 | 0, -5, 30, -4 | -80,110,130,144 | 16, 16, 16, 20 | 24,18,10,16 | O |
| Nev. Ruiz + Reventador, Fuego, Sumatra | 21 Sep 2015 | 5, 14, 3 | -76, -91, 98 | 16, 17, 16 | 13, 8, 19 | O |
| Sinabung, Fuego, Cotopaxi, Copahue | 15 Oct 2015 | 3, 14, 0, -38 | 98, -91, -80, -71 | 16, 17, 15, 15 | 30, 15, 6, 13 | O |
| Lascar, Sinabung, Nyamuragira, Fuego | 30 Oct 2015 | -23, 3, -1, 14 | -70, 98, 29, -91 | 17, 17, 16, 16 | 13,17,12,17 | O |
| Vanuatu, Tungurahua, Telica, Rinjani | 17 Nov 2015 | -16, -1,13, -5 | 168,-78,-87,116 | 18, 17, 17, 16 | 18,20,10,18 | O |
| Vanuatu, Reventador, Tengger | 5 Dec 2015 | -16, 0, 2 | 168, -78, 120 | 17, 16, 16 | 16, 15, 12 | O |
| Reventador, Sinabung | 18 Dec 2015 | 0, 3 | -78, 100 | 17, 16 | 16, 16 | O |
| Soputan +, Reventador, Fuego | 8 Jan 2016 | 1, 0, 14 | 125, -78, -91 | 16, 17, 14 | 25, 19, 5 | O |
| Semeru, Fuego | 10 Feb 2016 | -8, 14 | 113, -91 | 17, 16 | 34, 25 | O |
| Vanuatu +, Tungurahua | 27 Feb 2016 | -16, -1 | 168, -78 | 16, 16 | 24, 16 | O |
| Tungurahua, Sinabung +, Pavlof | 15 Mar 2016 | -1, 3, 55 | -78, 98, -162 | 16, 17, 15 | 23, 26, 7 | O |
| Reventador, Sinabung +, Fuego, Aira | 13 Apr 2016 | 0, 3, 14, 32 | -78, 98, -91, 131 | 17, 16, 15, 15 | 18, 30, 17, 6 | O |

| Volcano or region | Time | Latitude (°) | Longitude (°) | Altitude (km) | SO$_2$ (kt) | Instrument |
|---|---|---|---|---|---|---|
| Fuego, Nyamuragira + Ecuador, Langila, Sinabung | 7 May 2016 | 14, -1, -5, 3 | -91, 29, 150, 98 | 16, 17, 16, 17 | 16, 18, 16, 26 | O |
| Bulusan, Sinabung, Semeru, Mexico | 10 Jun 2016 | 13, 3, 8, 15 | 125,98,113,-100 | 17, 16, 17, 16 | 16,14,16,10 | O |
| Rinjani, Sinabung, Santa Maria | 1 Aug 2016 | -5, 3, 15 | 116, 98, -91 | 16, 16, 16 | 10, 30, 24 | O |
| Sinabung + Vanuatu, Fuego | 28 Aug 2016 | -16, 14 | 168, -91 | 16, 16 | 42, 23 | O |
| Ubinas, Sinabung | 3 Oct 2016 | -16, 3 | -71, 98 | 15, 16 | 16, 26 | O |
| Sabancaya, Sinabung + Bulusan | 5 Nov 2016 | -16, 3 | -72, 98 | 16, 16 | 38, 46 | O |
| Dukono, Vanuatu, Sabancaya | 12 Dec 2016 | 2, -16, -16 | 128, 168, -72 | 17, 18, 15 | 30, 28, 28 | O |
| Sabancaya, Reventador, Sinabung + Vanuatu | 10 Jan 2017 | -16, 0, 3 | -72, -78, 98 | 16, 17, 17 | 20, 30, 23 | O |
| Sabancaya, Colima, Sinabung | 4 Feb 2017 | -16, 19, 3 | -72, -104, 98 | 15, 16, 16 | 17, 15, 25 | O |
| Sabancaya, Dukono, Fuego, Manam + Vanuatu, Bogoslof, Nevados de Chillán | 5 Mar 2017 | -16, 2, 14, -16, 53, -37 | -72, 128, -91, 168, -170, -71 | 16, 17, 17, 17, 15, 15 | 10, 18, 8, 28, 4, 5 | O |
| Sabancaya, Nevado del Ruiz, Sinabung, Vanuatu, Klyuchevskoy | 10 Apr 2017 | -16, 5, 3, -16, 56 | -72, -75, 98, 168, 160 | 16, 16, 16, 16, 15 | 8, 15, 19, 17, 2 | O |
| Sinabung, Manam, Fuego | 5 May 2017 | 3, -4, 14 | 98, 145, -91 | 16, 17, 17 | 26, 10, 19 | O |
| Sheveluch + Bogoslof | 19 May 2017 | 57 | 161 | 15 | 20 | O |
| Santa Maria, Sheveluch +, Manam | 16 Jun 2017 | 15, 57, -4 | -91, 161, 145 | 16, 16, 15 | 11, 33, 6 | O |
| Fuego, Sinabung +, Sheveluch + | 5 Jul 2017 | 14, 3, 57 | -91, 98, 161 | 15, 16, 15 | 22, 21, 4 | O |
| Sinabung, Cristobal + Fuego, Sheveluch + Bogoslof | 8 Aug 2017 | 3, 13, 54 | 98, -87, -168 | 16, 17, 16 (26?) | 31, 27, 5 | O |
| Tinakula + Ambae | 21 Oct 2017 | -10, -15 | 166, 168 | 15, 15 | 60 | O |
| Agung, Ambae, Sabancaya | 27 Nov 2017 | -8, -15, -5 | 116, 168, -80 | 15, 16, 15 | 22, 7, 12 | O |
| Mayon, Vanuatu, Sabancaya | 22 Jan 2018 | 13, -15, -5 | 124, 168, -80 | 15, 17, 16 | 7, 20, 16 | O |
| Fuego, Vanuatu | 1 Feb 2018 | 14, -15 | -91, 168 | 16, 16 | 20, 17 | O |
| Sinabung, Vanuatu | 19 Feb 2018 | 3, -15 | 98, 168 | 16, 16 | 14, 21 | O |
| Ambae, Vanuatu | 26 Mar 2018 | -15 | 168 | 16 | 60 | O |
| Ambae | 6 Apr 2018 | -15 | 168 | 17 | 91 | O |
| Sabancaya | 15 May 2018 | -16 | -72 | 16 | 16 | O, T |
| Fuego | 3 Jun 2018 | 14 | -91 | 16 | 15 | O, T |
| Fernandina | 17 Jun 2018 | 0 | -92 | 15 | 8 | T |
| Agung, Sabancaya | 28 Jun 2018 | -8, -16 | 115, -72 | 17, 16 | 33, 23 | O, T |
| Sierra Negra | 8 Jul 2018 | -1 | -92 | 15 | 25 | T |
| Ambae | 20 Jul 2018 | -15 | 168 | 17 | 228 | O, T |
| Manam, Sabancaya | 25 Aug 2018 | -3, -16 | 144, -72 | 17, 16 | 25, 12 | O, T |
| Krakatau, Sabancaya | 23 Sep 2018 | -6, -16 | 105, -72 | 16, 16 | 5, 11 | O |
| Manam, Soputan, Reventador + Sangay | 4 Oct 2018 | -3, 1, 0 | 144, 125, -78 | 16, 16, 16 | 7, 4, 22 | O |

| Volcano or region | Time | Latitude (°) | Longitude (°) | Altitude (km) | SO₂ (kt) | Instrument |
|---|---|---|---|---|---|---|
| Nev.Ruiz, Sabancaya | 24 Oct 2018 | 5, -16 | -75, -72 | 16, 16 | 22, 11 | O |
| Fuego, Sabancaya, Krakatau | 6 Nov 2018 | 14, -16, -6 | -91, -72, 105 | 16, 16, 15 | 10, 16, 19 | O |
| Fuego, Sabancaya, Bagana | 26 Nov 2018 | 14, -16, -6 | -91, -72, 155 | 16, 16, 16 | 8, 9, 12 | O |
| Sabancaya, Manam, Soputan, Vanuatu | 8 Dec 2018 | -16, -3, 1,-16 | -72,144,125,168 | 16, 17, 16, 15 | 24, 8, 4, 6 | O |
| Krakatau, Vanuatu, Sabancaya | 23 Dec 2018 | -6, -16, -16 | 105, 168, -72 | 16, 15, 16 | 7, 6, 20 | O |
| Krakatau, Sabancaya, Manam | 4 Jan 2019 | -6, -16, -3 | 105, -72, 144 | 17, 17, 16 | 5, 20, 9 | O |
| Manam, Sabancaya | 24 Jan 2019 | -3, -16 | 144, -72 | 17, 16 | 23, 14 | O |
| Manam, Sabancaya | 14 Feb 2019 | -3, -16 | 144, -72 | 16, 16 | 12, 13 | O |
| Manam, Sabancaya, Mexico, Chile | 19 Mar 2019 | -3,-16,18,-24 | 144,-72,-98,-68 | 17, 16, 18, 15 | 9, 12, 6, 7 | O |
| Sabancaya, Manam, Nev.Ruiz, Gamalama | 20 Apr 2019 | -16, -3, 5, 1 | -72, 144,-75,128 | 17, 16, 16, 16 | 31, 12, 15, 7 | O, T |
| Sinabung, Manam, Sabancaya | 25 May 2019 | 3, -3, -16 | 98, 144, -72 | 17, 16, 16 | 11, 20, 21 | O, T |
| Raikoke | 22 Jun 2019 | 48 | 153 | 17 | 196 | O |
| Raikoke, Ulawun | 29 Jun 2019 | 48, -5 | 153, 151 | 15, 19 | 221, 107 | O, T |
| Ubinas, Raikoke, Manam | 19 Jul 2019 | -16, 48, -3 | -71, 153, 144 | 15, 16, 17 | 72, 141, 15 | O, T |
| Ulawun, Mexico | 3 Aug 2019 | -5, 20 | 151, -100 | 19, 17 | 111, 12 | O |
| Ubinas | 16 Aug 2019 | -16 | -71 | 16 | 27 | O, T |

Table 2: Inventory of volcanic SO₂ emissions into the stratosphere integrated over latitude belts above 14 km in low latitudes, 13 km in mid-latitudes and 12 km in high latitudes from the 3D mixing ratio perturbations. Listed altitudes and latitudes represent the region of maximum mixing ratio perturbation, the altitudes are close to the top injection height. For some eruptions two plumes in different altitudes were identified, the listed mass is the sum. Derived from satellite data (2002–2012) by MIPAS (M) and GOMOS (G). Based on a previous study from Brühl et al. (2018) with scaling factors for T63 and already published in an earlier version in Bingen et al. (2017) *(volcano names in italic)*. Extended with satellite data from SAGE II(V7.00) (S) back to 1990–2002, and from 2012–2019 by OSIRIS (O). Sometimes also TOMS/OMI/OMPS (T) are used for handling data gaps. For detailed description see the text. Data available online as Fortran formatted ASCII table and the 3D-data as netcdf (https://doi.org/10.26050/WDCC/SSIRC_3).

## 5 Implementation of SO₂ emissions into the EMAC Model

In the new approach in this study the SO₂ plumes are incorporated into the model simulations by adding the satellite-derived 3-dimensional perturbations of SO₂ mixing ratios to the simulated SO₂ at the time of the eruptions. In order to get the correct altitude distribution and to reduce additional errors caused by the low temporal resolution of the satellite data and possible numerical problems due to huge gradients or values out the range of used procedures in the model, we did not implement the volcanic SO₂ emissions as point sources. A comparison with point source injections in two case studies is provided in

Appendix C3.

Effusive eruptions and quiescent degassing volcanoes from the time-dependent monthly 3D climatology of Diehl et al. (2012) were added to the tropospheric $SO_2$ background emissions in the model simulations and truncated at an altitude of 200 hPa to avoid double counting in the stratosphere and uppermost troposphere since the original climatology also contains contributions of explosive volcanoes listed in Table 2 (only 1990 to 2009) (Brühl et al., 2018). In some cases, especially in the tropics, some $SO_2$ from degassing is transported by convection to the lowermost stratosphere (see e.g. 1998 in Figure 6).

The $SO_2$ emissions of our inventory are used in the EMAC model simulations, resulting in the time series shown in Figure 6, with mixing ratios between background conditions of a minimum of 0.001 ppbv (parts per billion by volume ($10^{-9}$)) in volcanically quiescent periods, and highly active volcanic conditions with a maximum of 114 ppbv (as indicated at the top of the colour key, 5-day average) after the Pinatubo eruption. Figure 6 shows the modeled vertical distribution of stratospheric $SO_2$ in the Junge-aerosol layer with the local maximum of $SO_2$ around 25 to 30 km altitude (Höpfner et al., 2013), typical mixing ratios of $SO_2$ are about 0.03 ppbv.

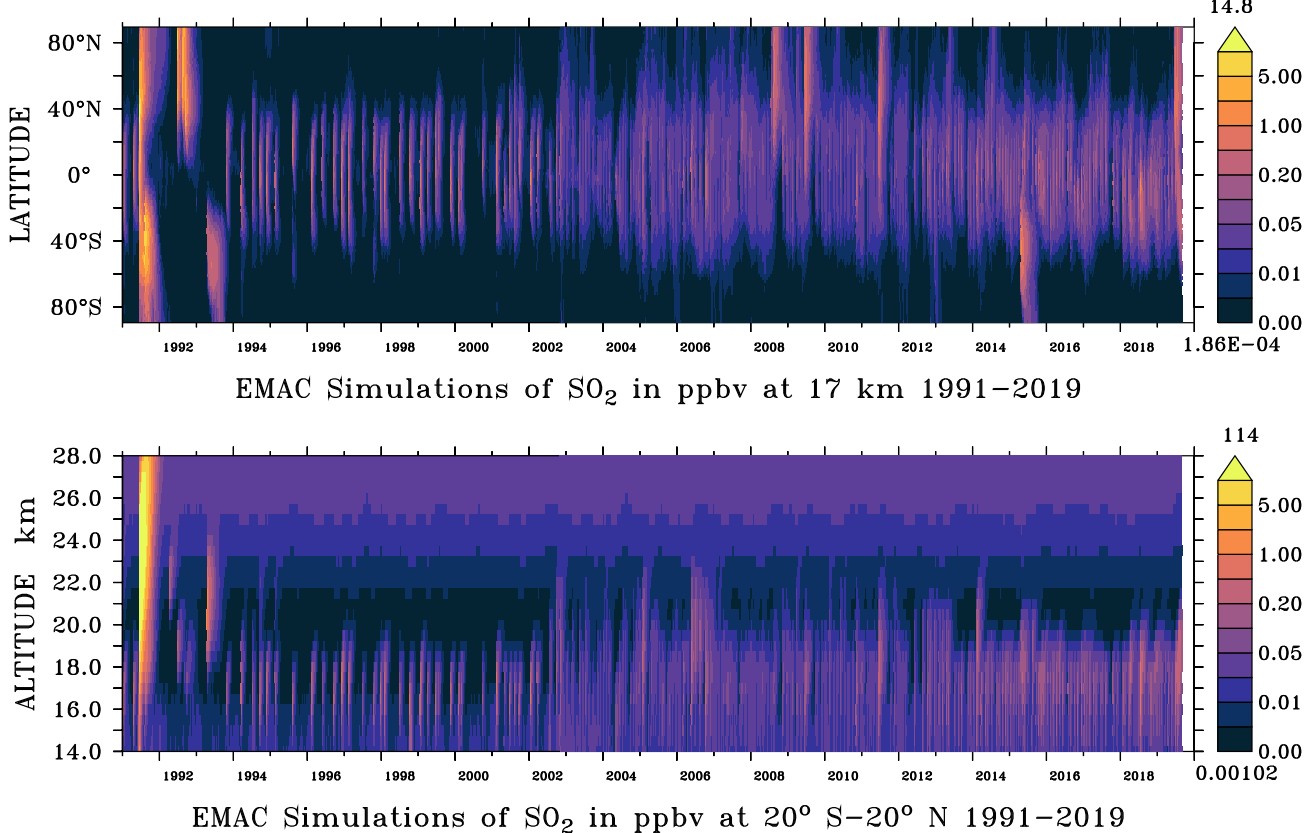

**Figure 6.** EMAC simulation of the stratospheric $SO_2$ mixing ratios (ppbv) (January 1991–August 2019, 5-day averages) using the volcanic sulfur emission inventory (Table 2), at 17 km altitude (top) and in vertical distribution for tropical regions 20° S–20° N (bottom). Maximum and minimum values are indicated above (yellow) and below (dark blue) the colour keys, respectively.

The volcanic eruptions in 1990 are included into the model during the spin-up phase of the model simulations (not shown), with the emissions of the first entry in Table 2 set to the upper limit consistent to the Smithsonian reports, SAGE and TOMS. The low number of volcanic eruptions in 1991 and the following years might be due to the low coverage of satellite data and "saturation" effects of the satellite instrument (see subsection 2.4 about SAGE). The signatures of medium and small volcanic eruptions are too weak to be seen during the high concentrations in the first years after the Pinatubo eruption. There are also less entries in the Smithsonian database. From 2002 onwards, a higher number of small volcanic eruptions is captured in the volcanic sulfur emission inventory. This might be rather due to the improved data coverage enabled by a larger number of satellite instruments, than to higher volcanic activity.

In most cases, the lower stratospheric $SO_2$ mixing ratios are highest at tropical latitudes. For this reason, tropical regions ($20°$ S–$20°$ N) are chosen for the vertical distribution in the lower illustration of Figure 6 and subsequent figures. Exceptions to this typical $SO_2$ pattern are single medium strong volcanic eruptions at high latitudes like Kasatochi (2008), Sarychev (2009) and Raikoke (2019) in the northern hemisphere or Calbuco (2015) in the southern hemisphere. Another noteworthy case is the Nabro (2011) eruption, where the volcanic emissions were transported from the tropics to northern latitudes by the Asian monsoon circulation (Bourassa et al., 2012b; Clarisse et al., 2014; Fairlie et al., 2014).

MIPAS typically captures background $SO_2$ mixing ratios in the lowermost tropical stratosphere at 16 km to 17 km of around 0.02 ppbv to 0.05 ppbv (Figure 1), which can be reproduced by our model by considering many more volcanoes than listed in the online National Aeronautics and Space Administration (NASA) $SO_2$ database (https://so2.gsfc.nasa.gov) or most other data bases in ISAMIP, e.g. Mills et al. (2016). Some time periods with low volcanic activity resulting in almost stratospheric background conditions can be identified between 1996 and 2004. To reach realistic $SO_2$ mixing ratios in the lower tropical stratosphere during these years, the contribution from oxidation of DMS and other sulfur species is important. The lower panel of Figure 6 shows increasing $SO_2$ with altitudes of above 23 km due to additional production from OCS photolysis.

The comparison of the simulated and observed $SO_2$ values shows that the volcanic $SO_2$ emissions from the volcanic sulfur emission inventory in Table 2 correlate well with the peaks of the mixing ratios in Figure 6, as they dominate the stratospheric sulfur burden. In the stratosphere, $SO_2$ is converted to sulfate aerosol, which explains most of the interannual variability of the stratospheric aerosol burden as well as its influence on the stratospheric radiation. Generally, the conversion of $SO_2$ to sulfate aerosol particles depends on several factors, such as the altitude, latitude, or season of the eruption and takes according to Höpfner et al. (2015) about 13, 23 and 32 days in 10–14, 14–18 and 18–22 km altitude, respectively, in midlatitudes. Carn et al. (2016) report an e-folding time varying between 2–40 days. The range agrees with our simulations (and assumptions in section 4). Enhanced $SO_2$ concentrations from Pinatubo via photolysis of gaseous $H_2SO_4$ remained in the mesosphere for several years (Brühl et al., 2015; Rinsland et al., 1995).

## 6 Climate impact of stratospheric aerosol in EMAC simulations

We compared the global influence of sulfur emissions on different atmospheric optical parameters. Based on Mie-theory-lookup tables, optical properties such as optical depth, single scattering albedo and asymmetry factor, which are used in radiative transfer simulations, were calculated online for different aerosol types: inorganics including sulfate, dust, organic carbon and black carbon, sea salt, and aerosol water (Dietmüller et al., 2016). Via multiple calls of the radiation module RAD with and without (stratospheric) aerosol the influence of stratospheric aerosol on instantaneous radiative forcing and heating is computed online (see section 3). Also, the feedback to atmospheric dynamics is included.

### 6.1 EMAC simulations of the stratospheric aerosol extinction

Figure 7 and Figure 8 show the global stratospheric aerosol extinction coefficients (in decadal logarithm) for the period 1991—2019 at 750 nm and 550 nm wavelengths of the EMAC model simulations at 17 km altitude and the vertical profile in tropical regions for 20° S–20° N. For medium eruptions, the maximum of the aerosol extinction lies at an altitude between 16 km and 18 km. For this reason, an altitude of 17 km is chosen in the following analyses.

Figure 7 also shows the extinctions observed by OSIRIS and SAGE (interpolated from the observations at 525 and 1020 nm). The strongest event in these model simulations is the Pinatubo eruption in 1991 (see Table 2), which dominates the stratospheric aerosol extinction coefficient for more than three years after the eruption with a global distribution from the equator to the poles in both hemispheres and a maximum altitude of more than 26 km. All other eruptions are significantly smaller, and for this reason a logarithmic scale is chosen. For further comparisons at 750 nm with GOMOS and OSIRIS we refer to Figure 3, Figure 4 and Brühl et al. (2018).

The EMAC model simulations of the aerosol extinction coefficients at 550 nm (Figure 8) agree well with the satellite measurements of GOMOS (Figure 2), SAGE II (Figure 5) and GloSSAC (Appendix C) for the aerosol layer at an altitude of 16–22 km where measured extinction values exceed $\approx 2 \times 10^{-4}\,\mathrm{km}^{-1}$. Above about 24 km EMAC is lower than the observations likely because in the model meteoric dust particles were not considered.

Figure 7 and Figure 8 show a similar distribution of the aerosol extinction at wavelengths of 550 nm and 750 nm. Due to the typical size and composition of stratospheric aerosol particles, the aerosol extinction is higher at 550 nm than at 750 nm. The peaks caused by mineral dust particles during summer in the Northern subtropics are more pronounced at 750 nm than at 550 nm.

Despite the presence of volcanoes in the Antarctic (like Mount Erebus), the seasonal change of extinction coefficient around 80° S is not due to volcanic eruptions, but to the presence of Polar Stratospheric Clouds (PSCs) as simulated by the model.

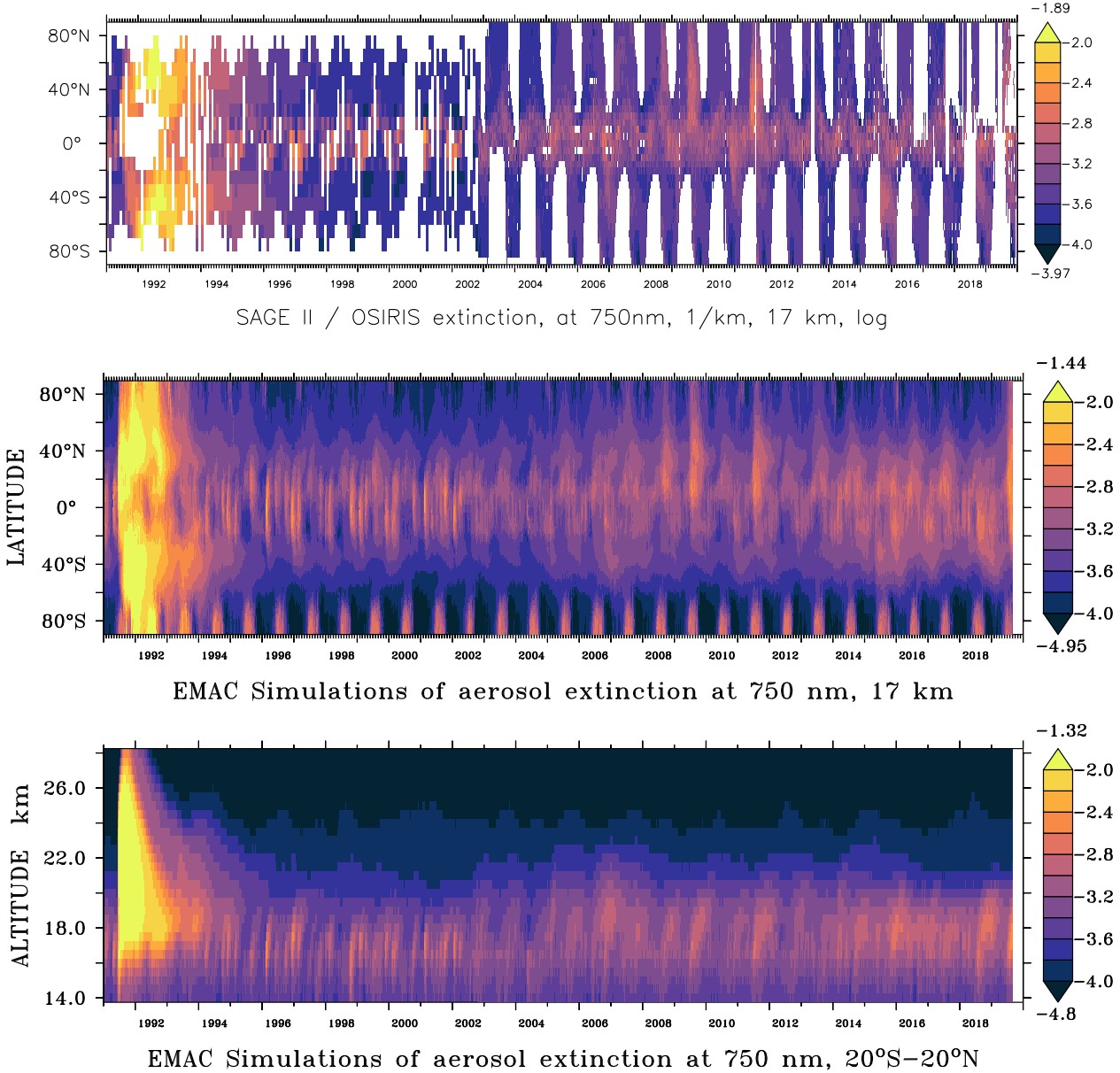

**Figure 7.** Comparison of stratospheric aerosol extinction for 750 nm wavelength at 17 km altitude between the model simulations (middle panel) and SAGE II and OSIRIS satellite data (upper panel), all values in logarithmic scale log(1/km). Vertical distribution of EMAC results for tropical regions 20° S–20° N in bottom panel. Maximum and minimum values appear above (yellow) and below (dark blue) the colour keys, respectively. 5-day averages, except for monthly SAGE data.

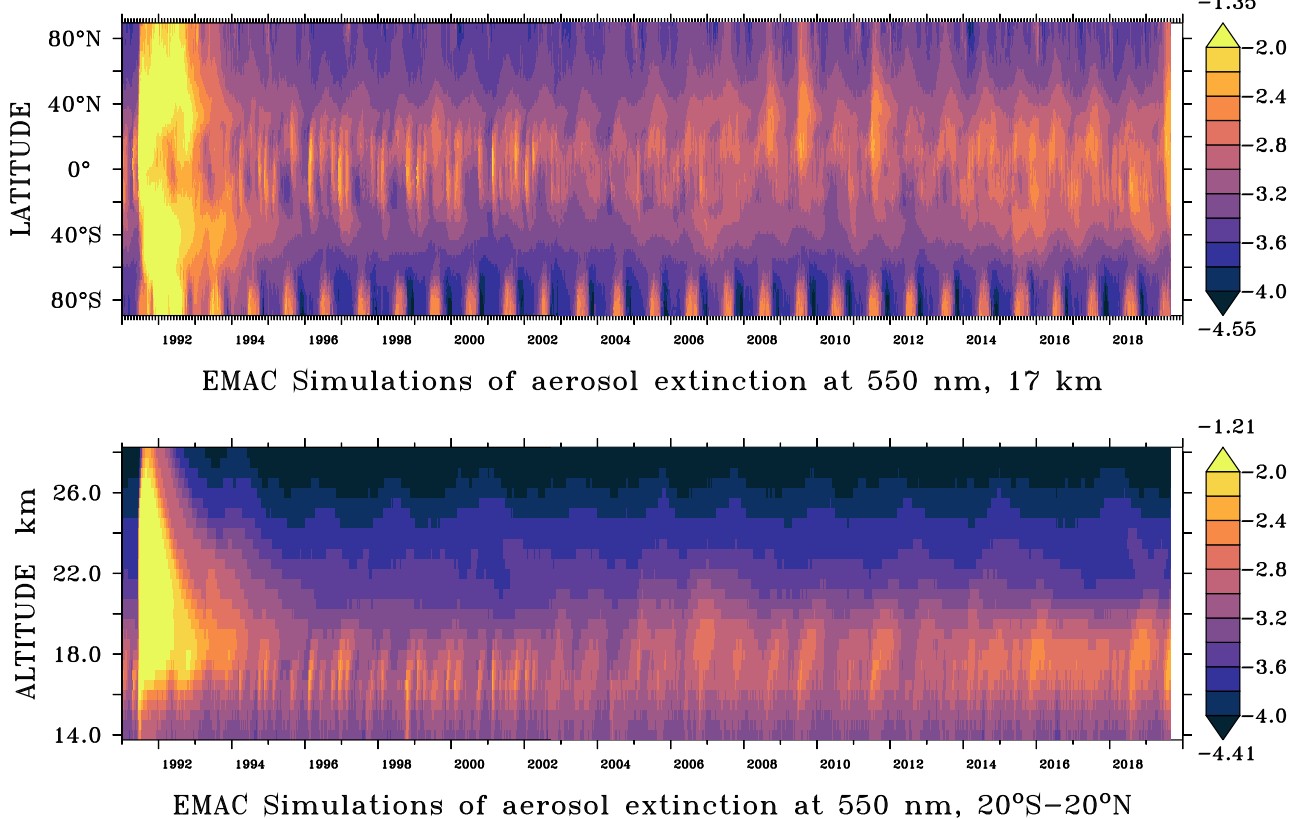

**Figure 8.** EMAC simulation of the stratospheric aerosol extinction on a logarithmic scale log(1/km) for 550 nm wavelength from January 1991–August 2019 (5-day averages), zonal mean at 17 km altitude (top) and vertical distribution for tropical regions $20^\circ$ S–$20^\circ$ N (bottom). Maximum and minimum values appear above (yellow) and below (dark blue) the colour keys, respectively.

## 6.2 EMAC simulations of Aerosol Optical Depth (AOD)

For practical reasons, the total stratospheric AOD is obtained by the vertical integral of the aerosol extinction above an altitude of about 16 km in the tropics and above about 13 km for mid-latitudes and high latitudes, to allow for a direct comparison with existing literature (Santer et al., 2014; Glantz et al., 2014) and satellite data. The stratospheric AOD is shown on a logarithmic scale in Figure 9 and Figure 10 with the new model simulations (red line) compared to satellite observations (light blue, gray, and blue lines). Using the wavelengths of the satellites in the calculations (Section 3) avoids introducing additional errors through the use of conversion factors to adjust the values between the different wavelengths.

From 1991 to 2012, SAGE II (light blue line), GOMOS (gray line) and SAGE+CALIPSO and SAGE+OSIRIS (blue line) provide satellite data at a wavelength of 550 nm (OSIRIS data were converted from 750 nm by Glantz et al. (2014)), shown in Figure 9. For comparison also GloSSAC (Kovilakam et al., 2020) is included as black line in the upper panels using the vertical integral over extinction.

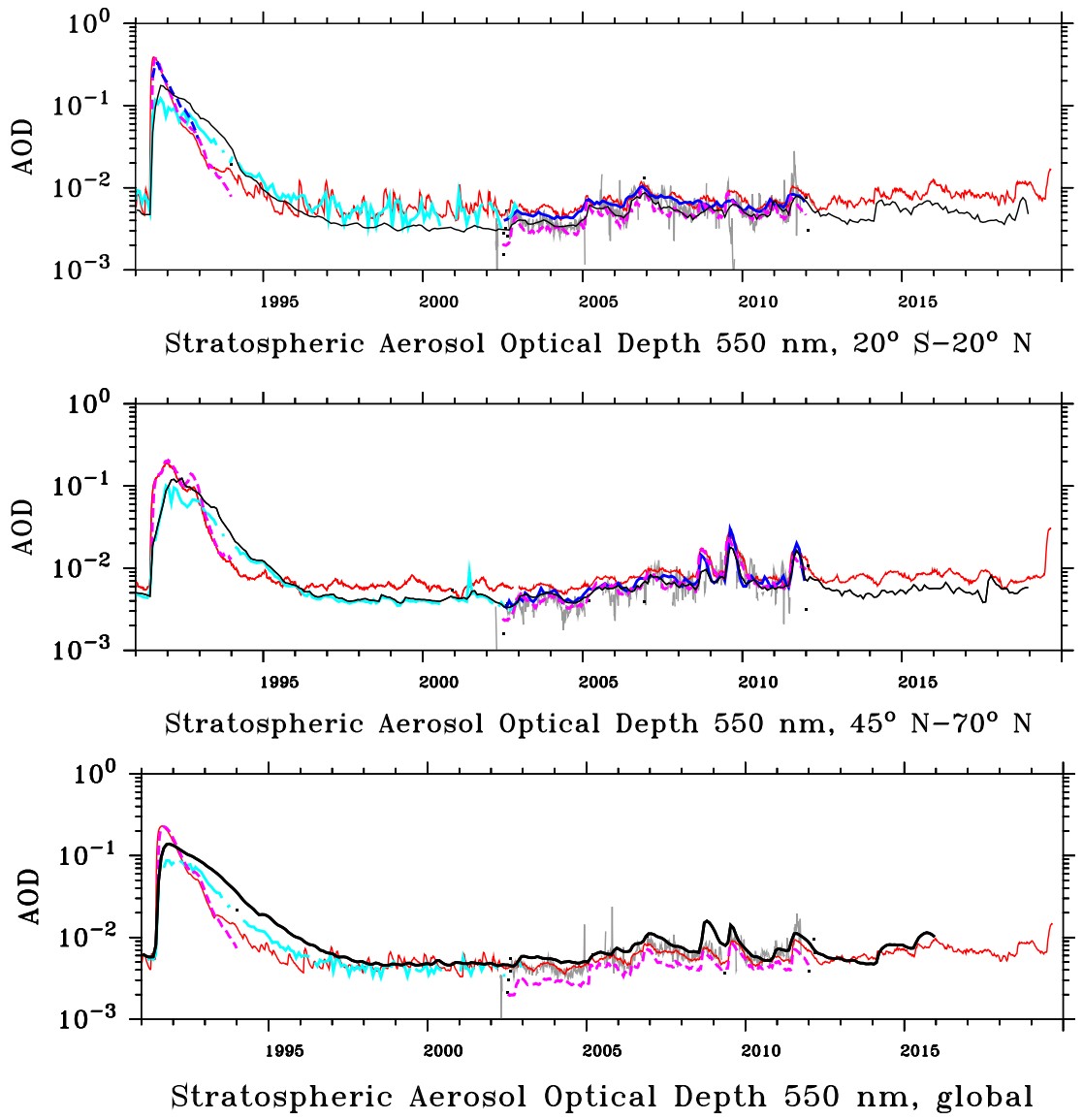

**Figure 9.** Stratospheric AOD at 550 nm wavelength: Tropical regions 20° S–20° N above 16 km are shown on the top, the northern hemisphere 45° N–70° N above about 13 km is shown in the middle and global means at the bottom. Satellite observations from SAGE II (Thomason et al., 2008) are indicated by the light blue line, GOMOS (Bingen et al., 2017) by the gray line and values derived from SAGE+CALIPSO (upper figure) (Santer et al., 2014) and SAGE+OSIRIS (middle figure) (Glantz et al., 2014) by the blue line. The red line shows the EMAC model simulations using the 3D-SO$_2$ injections of Table 2, compared to the simulations of Brühl et al. (2015) (pink dashed line) and the global stratospheric AOD from Schmidt et al. (2018) (black line in lower panel). The black line in the upper panels represents GloSSAC, derived from extinction at 525 nm. EMAC and GOMOS as 5-day averages, other data monthly.

The maximum is reached after the Pinatubo eruption with a stratospheric AOD of 0.4 in the tropics (Figure 9 upper panel, EMAC), being an order of magnitude larger than the following medium eruptions with a stratospheric AOD of about 0.01 (e.g. Manam in early 1997, Rabaul in 2006 and Nabro in 2011). The differences after the large Pinatubo eruption in 1991 between the model simulations and the SAGE II observations are related to the "saturation" effects of the satellite instrument (i.e. data

gaps due to an opaque path through the atmosphere at the tangent point) and can be observed for more than one year, also shown above in Figure 5. In GloSSAC gap filling (with lidar and CLAES data) was applied for this case. In this study about 17 Tg $SO_2$ were injected for the Pinatubo eruption (Guo et al., 2004). Model comparisons by Timmreck et al. (2018) show that the span of used injections varies between 10 Tg $SO_2$ (e. g. Dhomse et al. (2014); Mills et al. (2016); Schmidt et al. (2018)) and 20 Tg $SO_2$ (e.g. English et al. (2013)). Thus, this study is within the range of the injected sulfur masses. On the other hand,

filling the gaps in the SAGE data just by horizontal linear interpolation increases the peak AOD by about a factor of 2, which is close to the GloSSAC compilation. In Figure 10 the AOD from AVHRR (Advanced Very High Resolution Radiometer) by Long and Stowe (1994) at 630 nm is included, which is close to our simulations. Consistent with the typical wavelength dependence, these values lie between the red curves for 550 nm (Figure 9) and 750 nm (Figure 10) at the peak after the Pinatubo eruption.

When comparing the EMAC simulations (red line in Figure 9) with the simulation of Schmidt et al. (2018) (black line in Figure 9, lower panel) it can be recognized that a smaller value for the peak of the Pinatubo eruption occurs. Here it needs to be considered that Schmidt et al. (2018) inject less $SO_2$. For Pinatubo monthly averaging reduces the peak in EMAC by about 5%, the smaller events cause fluctuations up to +/-0.007 (see supplement, Fig. S2).

Between 1993 and 1996 the reduction of the stratospheric AOD in the model simulations is faster than indicated by the

satellite observations and in Schmidt et al. (2018). This indicates that the removal of stratospheric aerosol is still too rapid applying our modal model. Schmidt et al. (2018) show a slower decrease in AOD after the Pinatubo eruption. This could indicate that EMAC still needs better fine-tuning of the size distribution modes, or adding modes in the aerosol submodel to improve the aerosol removal in the stratosphere. Additionally, smaller volcanic eruptions might be missing, in view of the low number of identified events in the years after the Pinatubo eruption.

In Figure 10, the coverage of GOMOS (gray line) is often too low at a wavelength of 750 nm for the years from 2002 to 2012, so the inclusion of OSIRIS data (blue line) is important (Brühl et al., 2018). For the years after 2012 the timeline only contains data from OSIRIS at 750 nm wavelength.

Nevertheless, there remain small differences between the model simulation and the observations, for instance in 2010, which indicates missing volcanic eruptions (or an underestimation of the Merapi eruption by MIPAS compared to OSIRIS, Appendix

B, or to other data, Mills et al. (2016)).

The different distributions of the peaks in the upper and the lower panels is related to the latitude of the volcanic eruptions. Emissions reaching the stratosphere from strong eruptions in the tropics are distributed by the Brewer-Dobson circulation over the northern and southern hemisphere even to high latitudes, as in the cases of Soufriere Hills and Rabaul in 2006. However, if an eruption takes place at high latitudes (such as for Redoubt 2009) or at mid-latitudes like Kasatochi (2008), Sarychev (2009)

or Raikoke (2019), most of the emissions stay in the northern hemisphere and the signal in the tropics is weaker. Our northern

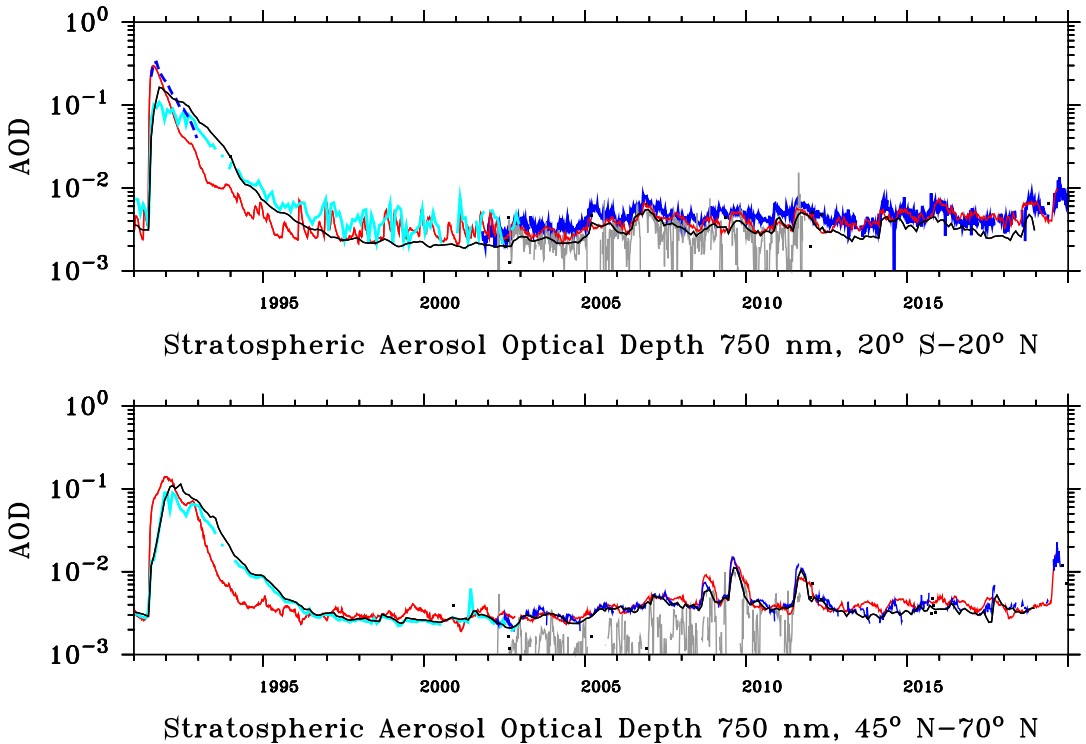

**Figure 10.** Stratospheric AOD at 750 nm wavelength: Tropical regions $20^\circ$ S–$20^\circ$ N above 16 km are shown on the top and for the northern hemisphere $45^\circ$ N–$70^\circ$ N above 13 km at the bottom. Satellite observations from OSIRIS (Rieger et al., 2019) are indicated by the blue line and GOMOS (Bingen et al., 2017) by the gray line. For the Pinatubo period the dashed blue line shows the AVHRR observations by Long and Stowe (1994) at 630 nm (upper panel). The light blue line shows the interpolation of SAGE data at 525 nm and 1020 nm wavelengths. The EMAC model simulations, using the 3D-$SO_2$ injections of Table 2, are shown by the red line. The black line represents GloSSAC, derived from extinction at 525 and 1020 nm. EMAC, GOMOS and OSIRIS as 5-day averages, other data monthly.

hemisphere results for AOD (at 550 nm) of about 0.025 after the Raikoke eruption agree within uncertainties with Kloss et al. (2021) who use different satellites and different modelling approaches.

## 6.3 EMAC simulations comparing the radiative forcing at the top of the atmosphere

The instantaneous radiative forcing of the stratospheric aerosol is calculated by multiple calls of the RAD submodel (section 3).
The simulated global instantaneous radiative forcing in $Wm^{-2}$ of stratospheric aerosol at the top of the atmosphere (TOA) is illustrated in Figure 11.

As the Pinatubo eruption caused a negative radiative forcing of more than an order of magnitude greater than all other eruptions since then, the figure is subdivided into two panels with different scaling. The lower panel shows the relatively small values. The new model simulations for the instantaneous radiative forcing at TOA with the additional volcanic eruptions (red
line) are closer to the estimates from satellite extinction measurements of SAGE, GOMOS and CALIOP by Solomon et al.

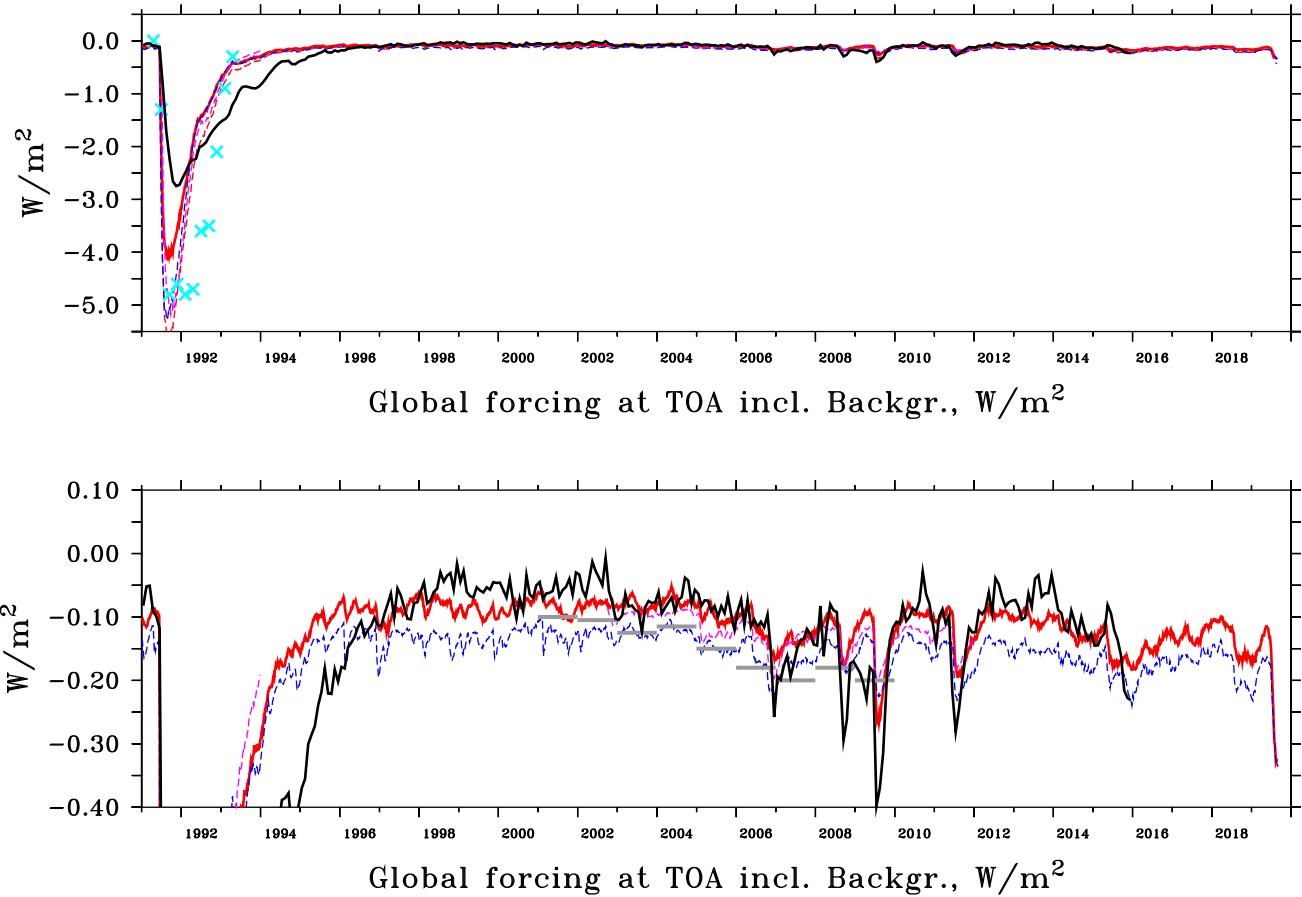

**Figure 11.** EMAC **instantaneous** radiative forcing by stratospheric aerosol (red, pink and blue lines, 5-day averages). Solar forcing at the top of the atmosphere (TOA) (dashed red line, upper panel) is compared to solar forcing at the TOA from satellite observations of the Earth Radiation Budget Experiment (ERBE), 72-day means, light blue crosses (Wong et al., 2006; Toohey et al., 2011). The full red line displays the total (solar+IR) forcing at TOA including contributions from aerosol down to the calculated tropopause. The blue dashed line shows the total forcing using the previous scheme at the tropical tropopause (Brühl et al., 2018), the dashed pink line the same with fewer volcanoes of Brühl et al. (2015). Gray bars show annual averages derived from observations by Solomon et al. (2011). The black line shows results from Schmidt et al. (2018) with volcanic effective radiative forcing (aerosol-radiation interactions) at TOA including a background aerosol forcing of -0.05 $Wm^{-2}$. The lower panel is an enhanced representation of the upper panel.

(2011) (gray bars) than in previous studies (e.g., Brühl et al. (2015), pink dashed line and Brühl et al. (2018), blue dashed line, for TOA see supplement, Fig. S1). In Figure 11 the published forcing values at the tropical tropopause of the previous studies are shown which are systematically more negative than the values at TOA. For Sarychev it is clear that this cannot compensate for the effect of the neglect of aerosol in the extratropical lowermost stratosphere. A comparison with volcanic effective radiative forcing (aerosol-radiation interactions) from Schmidt et al. (2018) is shown by the black line, including a forcing

of -0.05 Wm$^{-2}$ for stratospheric background aerosol (derived from numbers provided). Especially for high latitude eruptions their effective forcing (ari) is larger than instantaneous forcing in EMAC and the annual averages of Solomon et al. (2011), also because of a higher aerosol load in the lowermost stratosphere than in EMAC (see sensitivity studies for Sarychev in Appendix C3). In the period considered here, the volcanoes are the dominant factor in (instantaneous) global radiative forcing.

Background stratospheric aerosol like sulfate from other sources, dust and organics contributes about -0.04 Wm$^{-2}$ to the value in the order of -0.1 Wm$^{-2}$ at TOA in volcanically quiescent periods (e.g. in 2000, 2002 or 2004). At TOA absolute values up to -0.2 Wm$^{-2}$ (-0.14 Wm$^{-2}$ old approach where only aerosol above 100 hPa was considered) are reached after Rabaul (2006), Kasatochi (2008), Nabro (2011) and Calbuco/Sinabung (2015) and stronger than -0.32 Wm$^{-2}$ (old approach -0.2 Wm$^{-2}$) after Raikoke/Ulawun (2019) eruptions. The value for Raikoke/Ulawun is within the range discussed in Kloss et al. (2021).

The strongest instantaneous global radiative forcing in the model simulations is caused by the Pinatubo eruption with a maximum of about -5 Wm$^{-2}$ at TOA for the solar part (-4 Wm$^{-2}$ for Solar + IR forcing at TOA); this is in good agreement with the results of Minnis et al. (1993) and the observations of the ERBE satellite (light blue crosses in Figure 11). Schmidt et al. (2018) estimates the effective forcing from the difference of a simulation with and without volcanoes injecting into the stratosphere, i.e. including dynamical and chemical adjustments (effect for Pinatubo up to about 0.4 Wm$^{-2}$). For Pinatubo

monthly averaging instead of 5-day averaging reduces the peak by 0.2 Wm$^{-2}$, the smaller events cause fluctuations up to +/-0.02 Wm$^{-2}$ (i.e. 50% of background, see supplement, Fig. S3) in the fine temporal resolution.

## 6.4   EMAC simulations of the stratospheric aerosol radiative heating

The simulated instantaneous aerosol radiative heating in the model is derived from multiple radiation calls with and without aerosol in the radiation submodel RAD. Figure 12 shows the calculated local heating effects in the stratospheric aerosol

layer. Small and medium volcanic eruptions have the largest effects between altitudes of 17 km and 18 km and generate an atmospheric heating of up to 0.03 K/day. The eruption of Pinatubo, on the other hand, had significantly stronger effects at altitudes of 20 km to 25 km and caused atmospheric heating of more than 0.7 K/day, which corresponds quite well with the results of Rieger et al. (2020) showing a maximum of instantaneous solar heating rate of 0.5 K/day in the tropics near 24 km plus instantaneous thermal heating rates of about 0.2–0.3 K/day. This is about 23 times greater than all other eruptions in the

model simulation, including the Raikoke eruption in 2019.

Further, a seasonal signal contributes significantly to the radiative heating in the northern subtropics. This is caused by transport of desert dust to the UTLS mostly via the Asian summer monsoon convection, which generates additional heating during the time of the Asian summer monsoon (Brühl et al., 2018).

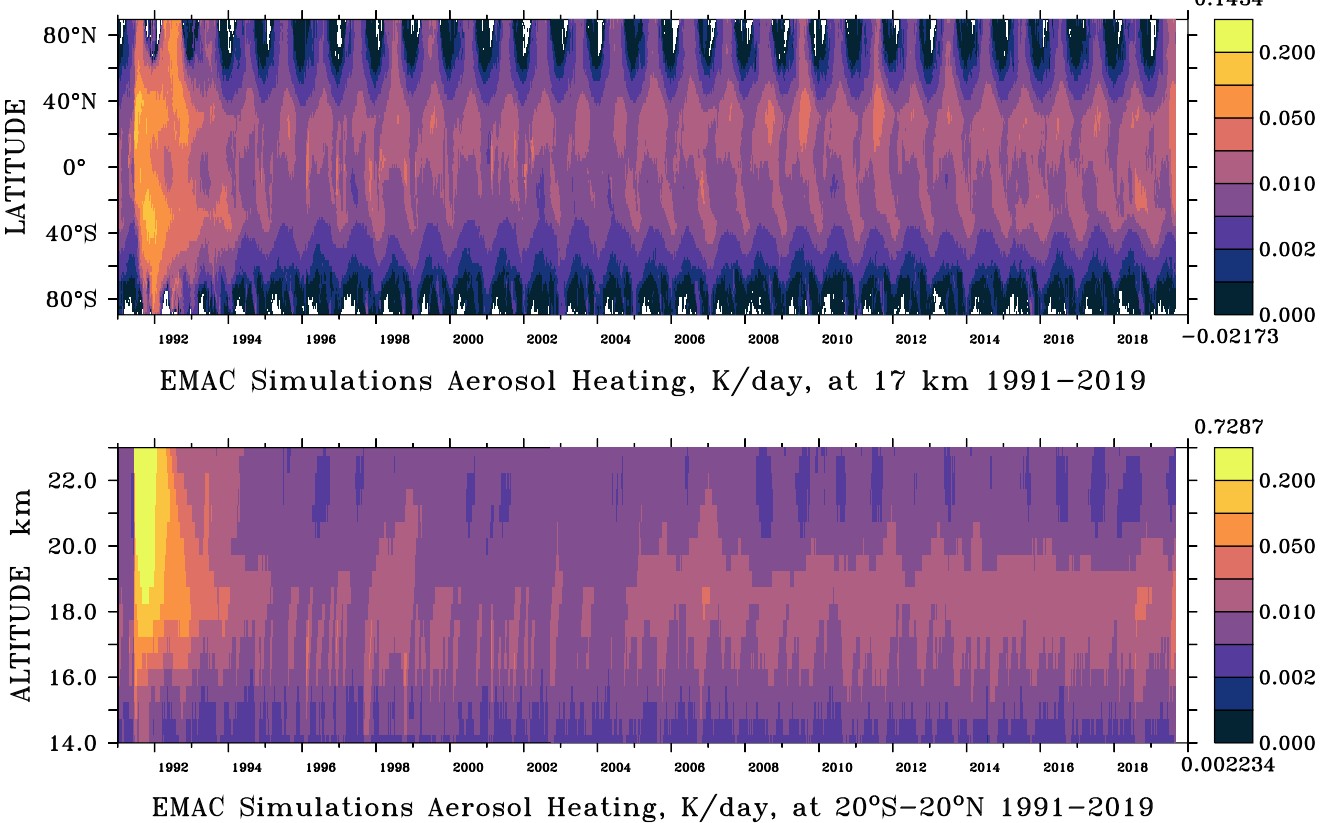

**Figure 12.** EMAC simulation of the aerosol radiative heating in K/day for solar and infrared radiation from January 1991–August 2019 (5-day averages) based on the volcanic sulfur emission inventory (Table 2), in horizontal distribution at 17 km altitude (top) and in vertical distribution for tropical regions 20° S–20° N (bottom).

## 7 Discussion and conclusions

The objective of this study was to generate a detailed volcanic sulfur emission inventory and to improve the EMAC model simulations of the global stratospheric aerosol and sulfate burden, and compute the long-term volcano-induced instantaneous radiative forcing using computed extinctions validated on the basis of satellite data.

The presented approach, based on observed 3-dimensional perturbations of SO$_2$ or extinction due to volcanic eruptions, avoids uncertainties related to the point source approach due to required additional assumptions and possible numerical artifacts
related to the extreme perturbations by several orders of magnitude in small regions and nonlinear feedback processes. Our case studies on this show large uncertainty in computed instantaneous radiative forcing and the possible development of spurious long-lived mesoscale vortices containing very high sulfate. In our approach the largest uncertainties are due to the handling of gaps in the satellite data.

Updated OSIRIS data allowed us to extend the comparisons of Brühl et al. (2018) to the year 2019; a detailed analysis of Level 2 SAGE II data (individual profiles of extinction at 2 wavelengths and SAD) together with the Smithsonian volcano database was used to extend the comparison back to 1990. As a result, the simulations in this paper encompass a total of 29 years and cover the period 1991 to 2019, compared with 2002 to 2012 in our previous work. The temporal resolution is five days for the three instruments MIPAS, GOMOS and OSIRIS and it is possible to identify multiple volcanic eruptions within a short period of time. With the three-dimensional data sets, the vertical distribution of $SO_2$ can be distinguished and the amount of sulfur reaching the stratosphere can be calculated much more accurately than by estimation from a total column. Tropospheric sulfur emissions are treated separately.

Our volcanic sulfur emission inventory is an extension and improvement of the versions published by Bingen et al. (2017) and Brühl et al. (2018). While the previous version included 230 explosive volcanic eruptions, the new version includes more than 500 eruptions. An overview of these eruptions is given in the volcanic sulfur emission inventory in Table 2, which indicates the estimated stratospheric $SO_2$ emissions as well as the plume altitudes. These consist of about 80 eruptions in the first time period between 1990–2002 measured by the SAGE II instrument, 240 eruptions in 2002–2012 measured with multiple instruments and 230 eruptions in the last time period 2012–2019 measured by OSIRIS. The consequence of the inclusion of many more small size eruptions reaching the UTLS is that stratospheric aerosol optical depth and radiative forcing do not decrease to the nonvolcanic background between medium size eruptions, in agreement with observations.

Strong volcanic eruptions can inject several teragrams of $SO_2$ directly into the stratosphere. For this reason, the maxima of the global stratospheric $SO_2$ concentrations correlate well with the eruption events of the volcanic sulfur emission inventory in Table 2. The $SO_2$ emissions of smaller volcanic eruptions can reach the lower stratosphere by convective transport through the tropical tropopause, which results in accumulation of sulfate aerosol in the lower stratosphere. This was demonstrated to be important for correctly modelling the AOD in volcanically quiescent periods.

Our analysis shows the importance of using multi-instrument satellite data sets to fill data gaps and to detect as many volcanic eruptions as possible. The optimal data coverage time period is from 2002 to 2012, for which simultaneous measurements from the MIPAS, GOMOS and OSIRIS instruments are available. For 2002 to 2005 this includes also SAGE II. The periods with simultaneous observations by MIPAS and the other instruments were used to develop and validate a method for estimating injected $SO_2$ and its distribution from extinction observations. The evaluation by the satellite data sets shows that GOMOS is important for detecting volcanic eruptions during MIPAS data gaps and for a better attribution of individual eruptions. Consequently, the combination of MIPAS, GOMOS and OSIRIS data leads to an improved $SO_2$ emission database for calculating the instantaneous radiative forcing in the chemistry climate model EMAC.

Large volcanic ash plumes can interfere with the $SO_2$ signal in satellite measurements, and satellites could be "blind" during the first few days or months after an eruption (Höpfner et al., 2015). However, most volcanic ash particles are relatively large and sediment after some hours or days, hence they have only minor climatic significance (Boucher, 2015) and are not discussed in detail here. There are, however, volcanoes (e.g. eruption of Kelut in 2014 (Zhu et al., 2020)) which emitted small ash particles which can stay in the lower stratosphere for several months (Vernier et al., 2016). Moreover, we found that the model setup might have to be improved by including an additional aerosol mode to reduce the removal by sedimentation of stratospheric

aerosol after large volcanic eruptions. Some satellite data sets are affected by gaps and noise. Comparing the model results with OSIRIS data in the tropics (Figure 10) indicates that some volcanic events are underestimated or missing in the volcanic sulfur emission inventory in the year 2010.

Frequent volcanic eruptions of moderate and small intensities, injecting sulfur gases to the upper troposphere and lower stratosphere, contribute significantly to the stratospheric aerosol layer in particular through their accumulation in time. These cause a global negative instantaneous radiative forcing of the order of more than $0.1\,\mathrm{Wm}^{-2}$ at the TOA, including a background aerosol forcing of about $0.04\,\mathrm{Wm}^{-2}$. For example, after the eruptions of Soufriere Hills/Rabaul (2006), Nabro (2011) and the combination of the Sinabung, Wolf and Calbuco eruptions (2015) a negative instantaneous radiative forcing of $0.2\,\mathrm{Wm}^{-2}$ ($0.14\,\mathrm{Wm}^{-2}$ old approach), and $0.32\,\mathrm{Wm}^{-2}$ ($0.2\,\mathrm{Wm}^{-2}$ old approach) was reached after Raikoke/Ulawun (2019) at TOA.

Note that for the calculation of the instantaneous forcing by these medium size eruptions it is essential to consider the radiative effect of volcanic aerosol down to the tropopause (Ridley et al., 2014). Including only the aerosol above the $100\,\mathrm{hPa}$ level as done in Brühl et al. (2018, 2015) can lead to significant underestimates which were partially hidden in these studies by showing the instantaneous forcing at the tropopause which is about $0.08\,\mathrm{Wm}^{-2}$ stronger than at TOA in EMAC.

The model also includes mineral dust and organics transported from the lower troposphere to the UTLS. The EMAC simulations show a seasonal signal in the stratospheric AOD and an enhanced radiative heating in the northern hemisphere, induced by the convective transport of mineral dust to the UTLS in the Asian monsoon region. This is confirmed by satellite observations and by Klingmüller et al. (2018). The influence of wildfires and other biomass burning plumes on stratospheric AOD has increased in recent years (Fromm et al., 2019), and this effect should be included in the model to account for perturbations in organic and black carbon.

*Code and data availability.* The Modular Earth Submodel System (MESSy) is continuously developed and used by a consortium of institutions. The use of MESSy and access to the source code is licensed to all affiliates of institutions which are members of the MESSy Consortium. Institutions can become a member of the MESSy Consortium by signing the MESSy Memorandum of Understanding. More information can be found on the MESSy Consortium website (http://www.messy-interface.org, MessyConsortium, 2017). The input data files and model output of EMAC used here are stored at DKRZ, Hamburg, the volcanic inventory and the output for instantaneous radiative forcing also at WDCC (https://doi.org/10.26050/WDCC/SSIRC_3). For the MIPAS data we refer to http://www.imk-asf.kit.edu/english/308.php. A detailed documentation on the used data for MIPAS, SAGE, OSIRIS and GOMOS for the individual events is available from the authors on request.

*Author contributions.* JS, CBr and JL defined the scientific questions and scope of this work. JS and CBr conceived the idea and methodology used in this paper and carried out the model simulations and the analysis. MH provided the MIPAS data, CBi provided the GOMOS data and LR provided the OSIRIS data. All authors participated in the scientific discussion in regard to satellite data in particular. JS wrote the manuscript, and all authors reviewed the manuscript and provided advice on the manuscript and figures.

*Competing interests.*   The authors declare that they have no conflicts of interest.

*Acknowledgements.*   The research leading to these results has received funding from the European Research Council under the European Union's Seventh Framework Programme (FP7/2007-–2013)/ERC grant agreement 226144 as part of the StratoClim project and funding from ESA Aerosol CCI (European Space Agency). There is a close collaboration within the SPARC/SSIRC (Stratosphere-troposphere Processes And their Role in Climate / Stratospheric Sulfur and its Role in Climate) stratospheric aerosol model intercomparison project

(http://www.sparc-ssirc.org) concerning $SO_2$ injections derived from satellite data and model intercomparisons. Many thanks to Matthias Kohl for helping to set up the additional point-source emission simulations. The computations have been performed at the Mistral and Levante supercomputers at DKRZ, Hamburg, Germany. We thank Adam Bourassa, Larry Thomason and the GOMOS, MIPAS, OSIRIS, SAGE, the Copernicus Climate Change Service (projects C3S_312a_Lot5 and C3S_312b_Lot2), and EOSDIS (NASA) teams for their productive cooperation and providing their satellite data sets. GloSSAC (V2) data were obtained from the NASA Langley Research Center

- Atmospheric Sciences Data Center. For the $SO_2$ inventory we also utilized the NASA $SO_2$ database at GSFC (http://so2.gsfc.nasa.gov) for OMI and TOMS and the Smithsonian volcano database (http://www.volcano.si.edu). We are grateful to our colleagues from the EMAC community and all other MESSy developers and users for their support. We thank Anja Schmidt for editing our paper and providing the data of her paper for comparisons with our results. We would also like to thank the referees for their comments to help us improving our paper.

## Appendix A:  List of MESSy submodels used in this study

The computations for this study were performed on the Mistral supercomputer at the Deutsches Klimarechenzentrum (DKRZ), Hamburg, Germany. For this purpose, EMAC (ECHAM5/MESSy Atmospheric Chemistry), a coupled atmospheric circulation model consisting of the $5^{th}$ generation of European Centre Hamburg general circulation model (ECHAM5) and the Modular Earth Submodel System (MESSy) was used.

ECHAM5 ($5^{th}$ generation of European Centre Hamburg general circulation model) is an atmospheric general circulation

model (Roeckner et al., 2003, 2006), which runs with self-consistent quasi-biennial oscillation (QBO). It is nudged to the meteorological ERA-Interim reanalysis data of the European Centre for Medium-Range Weather Forecasts (ECMWF) up to about 100 hPa. To avoid a phase drift, we used the QBO submodel for weak nudging to the QBO zonal wind observations (Giorgetta et al., 2002).

MESSy (Modular Earth Submodel System) is an earth system model, which consists of several submodels (Jöckel et al.,

2005, 2006, 2010). An overview of the submodels used in this study is given in Table A1.

Mineral dust emissions are calculated online using the emission scheme of Astitha et al. (2012) as part of the ONEMIS submodel. The submodel TREXP (Jöckel et al., 2010) is needed to inject $SO_2$ emissions in point source emission simulations. The convection was calculated with the CONVECT submodel (Tost et al., 2006b), with the convection scheme from Tiedtke (1989) and the Nordeng (1994) closure. The convection parametrization is sensitive to the model resolution, which results

in differences between different model resolutions in the vertical transport of tracers, like dust, water vapor, ozone and $SO_2$, especially near the low latitude tropopause (Brühl et al., 2018).

| Submodel | Function | Reference |
|----------|----------|-----------|
| AEROPT | Aerosol optical depth | Dietmüller et al. (2016) |
| AIRSEA | Air-sea exchange of trace gases | Pozzer et al. (2006) |
| CAABA/MECCA | Atmospheric chemistry | Sander et al. (2011) |
| CONVECT | Convection processes | Tost et al. (2006b) |
| CVTRANS | Convection transport of tracers | Tost et al. (2010) |
| DDEP | Dry deposition | Kerkweg et al. (2006a) |
| GMXe | Global Modal Aerosol eXtension | Pringle et al. (2010) |
| IMPORT | Import of external data files | Jöckel et al. (2006) |
| JVAL | Photolysis rate coefficients | Jöckel et al. (2006) |
| LNOX | NOx lighting production | Tost et al. (2007) |
| MSBM | Multiphase Stratospheric Box Model | Jöckel et al. (2010) |
| OFFEMIS | Off-line emissions | Kerkweg et al. (2006b) |
| ONEMIS | On-line emissions | Kerkweg et al. (2006b) |
| QBO | QBO nudging | Giorgetta et al. (2002) |
| RAD | RADiation | Dietmüller et al. (2016) |
| SCAV | Scavenging (wet removal) | Tost et al. (2006a) |
| SEDI | Aerosol sedimentation | Kerkweg et al. (2006a) |
| TNUDGE | Tracer nudging | Kerkweg et al. (2006b) |
| TREXP | Tracer release experiments from point sources | Jöckel et al. (2010) |
| TROPOP | Tropopause calculation | Jöckel et al. (2006) |

**Table A1.** List of MESSy submodels used. References and short description from http://www.messy-interface.org. Parts of the base model copied into MESSy which must always be active are not listed here.

The loss of gas phase species to the aerosol is parametrized in the 3rd EQuilibrium Simplified Aerosol Model (EQSAM3) (Metzger and Lelieveld, 2007). The uptake of gases on wet particles and on acid aerosol particles is included in the model calculation. Concerning removal mechanisms, the SCAVenging submodel calculates the loss of atmospheric tracers and aerosols

by wet deposition, as well as the liquid phase chemistry in clouds and precipitation (Tost et al., 2006a). The chemistry of the CAABA/MECCA submodel contains photolysis reactions, which need photolysis rate coefficients (J-values) for tropospheric and stratospheric species computed by the JVAL submodel. The RAD_FUBRAD sub-submodel is used to calculate the short-wave heating rates from the absorption of UV by $O_2$ and $O_3$ in the upper stratosphere and mesosphere. The lowermost level of the RAD_FUBRAD sub-submodel for the upper atmosphere is shifted from above 70 hPa in the original version of Dietmüller

et al. (2016) to 30 hPa–14 hPa to allow for scattering by the aerosol in the simulations with volcanic emissions.

## Appendix B: Comparison of volcanic injections derived from simultaneous MIPAS, SAGE, GOMOS and OSIRIS observations

The eruption of Reventador in the tropics in November 2002 has shown to be an ideal case where simultaneous observations of all satellite sensors were available so that the direct $SO_2$ observation could be used for development and validation of a

605 conversion formula for the 750 nm extinction seen by GOMOS and OSIRIS, which works also approximately for SAGE if its observations at 525 and 1020 nm are interpolated to 750 nm. Here we first use the ratio between model calculated sulfate volume mixing ratio and its share on extinction in low latitudes of the lower stratosphere which is typically $1.2 \times 10^{12}$ / air density (in molecules/$cm^3$) for the period during which MIPAS was available. This works for medium and small size eruptions and data available over about four weeks following the eruptions, and if no other events occur less than about four weeks before

610 which is the case for the Reventador eruption. If the time lag of data is several weeks a correction factor >1 has to be applied in Equation 1 to account for removal processes, if another event is relatively close in time, the factor has to be <1 to remove the influence of the previous event (see Table S1 and S2 in the supplement, which indicates also additional uncertainties for a few cases in 2018 and 2019 due to too sparse data).

For Reventador the factor is 1 for all instruments (for OSIRIS 0.8 is slightly better because of remnants from the Ruang

eruption about 4 weeks before). For all instruments the derived injected $SO_2$ mass is very close to 77 kt as shown in Table 2. The spatial patterns are similar, except that the zonal wind causes a shift in longitude due to the time lag from conversion of $SO_2$ to aerosol, see Figure B1. In the case of SAGE, the alternate method of Grainger et al. (1995) involving aerosol surface area density (SAD) and aerosol volume density is more suitable to remove cloud perturbation and used in the simulation. It is assumed that sulfate mixing ratios correspond to the $SO_2$ injected. Some uncertainty remains from removing the background

which we have done by subtracting a fraction of the derived $SO_2$ at the longitude where it has a minimum, i.e. the longitude where the effect of the volcano is smallest for all altitudes. If the extinctions at 525 and 1020 nm are used directly, the patterns are similar except for the lowermost part which contains more data gaps. This has been checked for every event prior to MIPAS also. Integrated injected $SO_2$ masses for all examples are provided in Table B1.

| Eruption | MIPAS | GOMOS | filled | OSIRIS | SAGE II Gr. | ext. |
|---|---|---|---|---|---|---|
| Reventador 2002 | 77 | 75 | - | 89 | 80 | 50 |
| Merapi 2010 | 97 | 18 | 77 | 170 | – | – |
| Sarychev 2009 | 446 | 141 | - | 353 | – | – |
| Manda Hararo 2009 | 82 | 81 | - | 101 | – | – |

**Table B1.** Integrated mass of injected $SO_2$ in kt for the different methods in Figure B1 to Figure B3.

For the eruption of Merapi in November 2010 the satellite instruments do not agree. From OSIRIS about 70% more injected

$SO_2$ is derived than from MIPAS, i.e. 170 kt instead of 97 kt used in the transient simulation (see Table 2 and differences in Figure 10). GOMOS has too sparse data here to obtain a proper integral directly but patterns are similar (Figure B2). If other

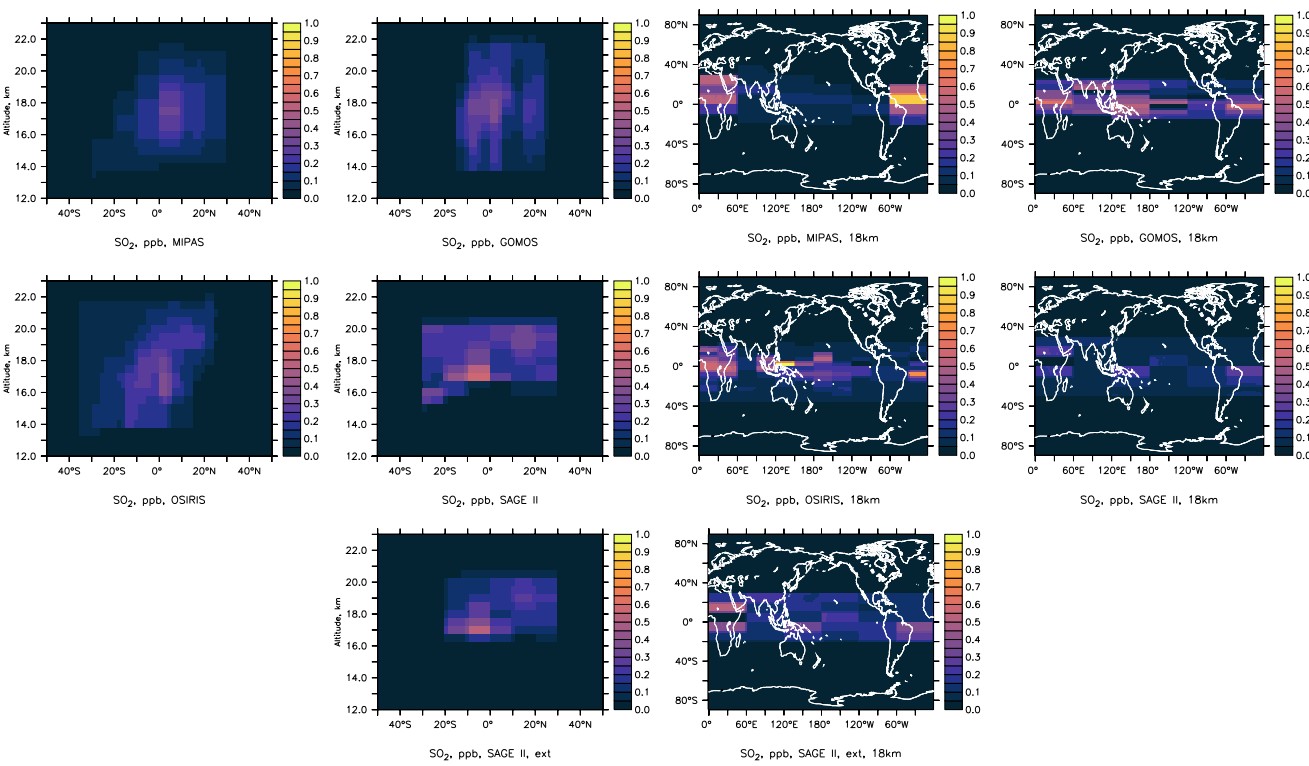

**Figure B1.** 2002 Reventador eruption: SO$_2$ mixing ratio perturbation (ppb) derived from MIPAS and the 3 extinction instruments. MIPAS data of 7 to 17 November 2002, OSIRIS and GOMOS data 11 to 22 November, SAGE II orbits of November and early December 2002. Zonal average vertical distribution (left) and plumes at 18 km altitude (right). OSIRIS with correction factor 0.8 due to remnants from the Ruang eruption, SAGE II with the Grainger-based method. Lower row: SAGE II SO$_2$ derived from extinction (interpolated to 750 nm from 525 and 1020 nm). Factor 1 is default.

information is available, the gaps can be filled with likely values in the region where the plume was seen, a method which had to be applied also to some events seen by OSIRIS in 2018 and 2019 for which the data were sparse.

For high latitude eruptions the longer conversion time of SO$_2$ to sulfate compared to the tropics has to be considered which, together with aerosol removal processes, leads to a weaker extinction signal. To account for this a correction factor of about two in the conversion formula for OSIRIS for example for Sarychev in June 2009 leads to values consistent to the ones derived by MIPAS (Figure B3). For the low latitude eruption of Mando Hararo in the same entry of Table 2 (separated at 24° N for the integration) the factor 1 is still appropriate.

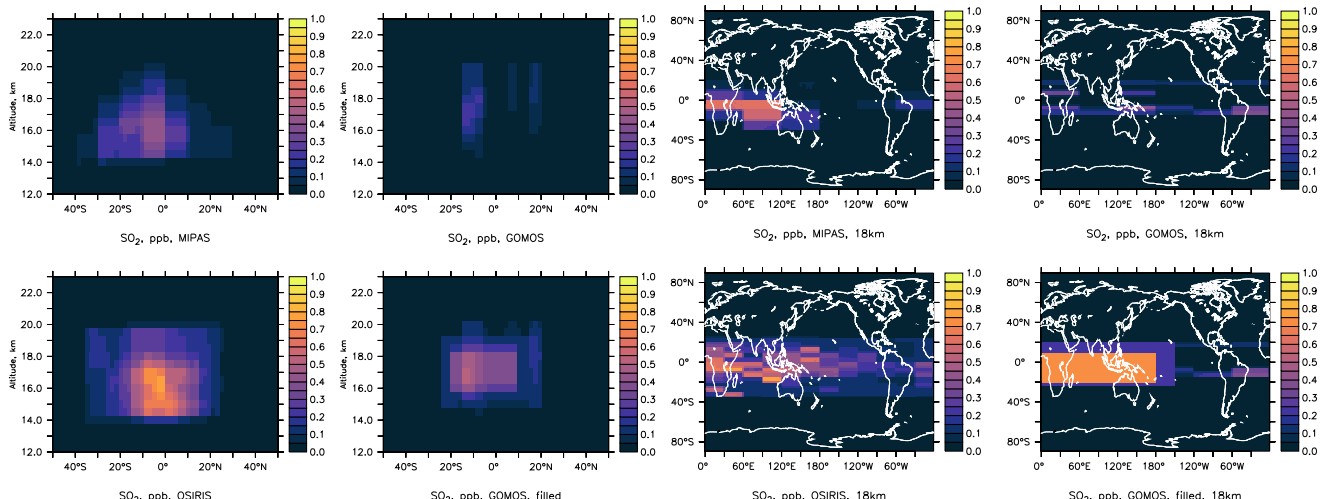

**Figure B2.** 2010 Merapi eruption: $SO_2$ mixing ratio perturbation (ppb) derived from MIPAS, GOMOS OSIRIS and GOMOS with gap filling. MIPAS data 5 to 20 November 2010, OSIRIS 14 to 25 November, GOMOS 25 November to 9 December (sparse data). Zonal average vertical distribution (left) and plumes at 18 km altitude (right).

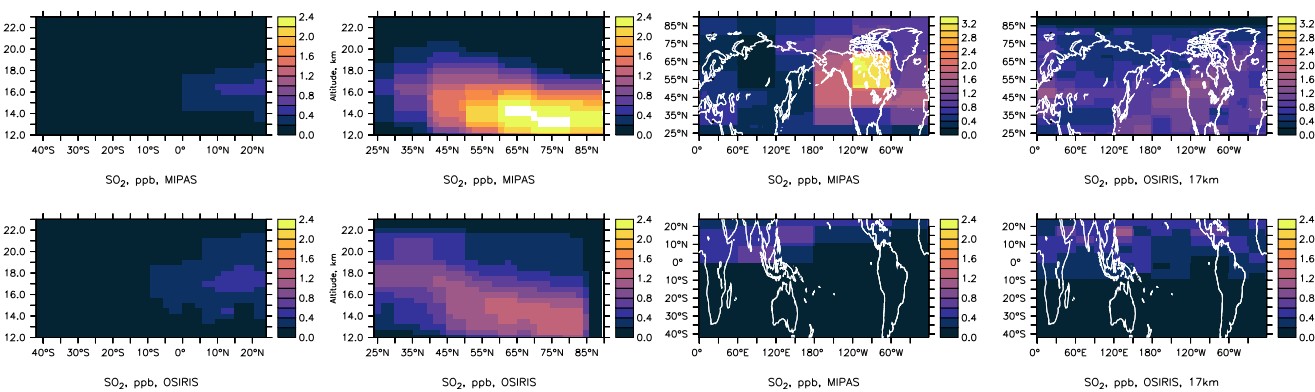

**Figure B3.** 2009 Sarychev (second column, upper right row) and Mando Hararo (first column, lower right row) eruptions: $SO_2$ mixing ratio perturbation (ppb) derived from MIPAS and OSIRIS, arrangement of panels because event is split at $24°$ N as for integrated values in Table 2. MIPAS data 18 June to 18 July 2009, OSIRIS data 17 July to 3 August. Zonal average vertical distribution (left) and plumes at 17 km altitude (right).

## C1    Comparison with GloSSAC

Figure C1 shows the aerosol extinction of GloSSAC in the same logarithmic scale log(1/km) as the used satellite data sets and the EMAC simulations in Figure 2–Figure 8.

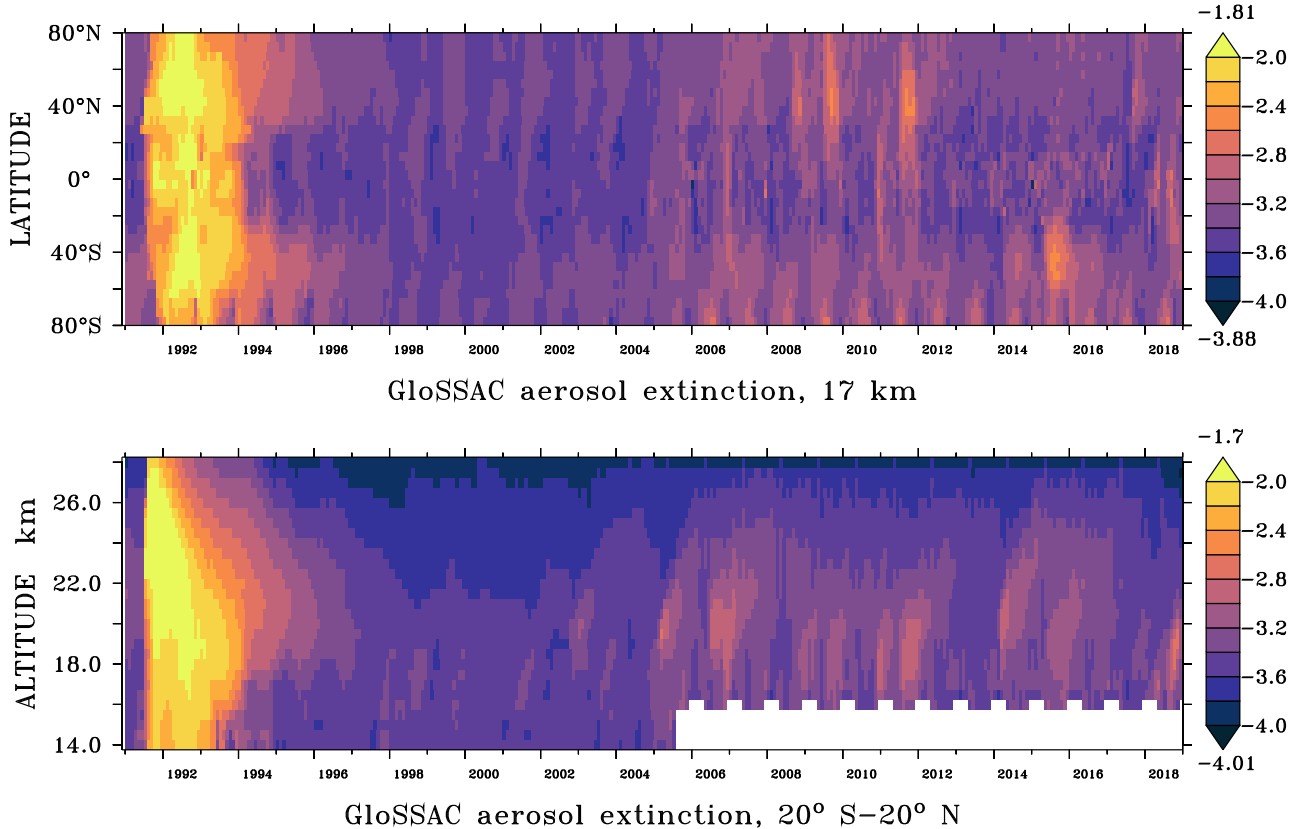

**Figure C1.** GloSSAC aerosol extinction on a logarithmic scale log(1/km) for 525 nm wavelength from January 1991–December 2018 zonal mean at 17 km altitude (top) and in vertical distribution for tropical regions 20° S–20° N (bottom). Maximum and minimum values appear above (dark red) and below (violet) the colour keys, respectively. White: no data.

## C2    Comparison of different volcanic injection inventories

Table C1 shows a comparison of annual volcanic SO$_2$ emissions inventories from Carn et al. (2016) and Mills et al. (2016)

with this study.

| Year | This study: stratospheric | Carn et al. (2016): total | Carn et al. (2016): explosive | Mills et al. (2016): total |
|------|---------------------------|---------------------------|-------------------------------|----------------------------|
| 1990 | 744 | 186 | 186 | 50 |
| 1991 | 18424 | 26082 | 24214 | 19500 |
| 1992 | 794 | 810 | 810 | 600 |
| 1993 | 426 | 450 | 450 | 400 |
| 1994 | 337 | 1874 | 360 | 300 |
| 1995 | 156 | TOMS data gap | - | - |
| 1996 | 436 | 987 | 100 | 860 |
| 1997 | 254 | 41 | 41 | - |
| 1998 | 255 | 3265 | 38 | - |
| 1999 | 398 | 130 | 85 | 70 |
| 2000 | 185 | 653 | 336 | 520 |
| 2001 | 400 | 1783 | 122 | 34 |
| 2002 | 368 | 2626 | 271 | 160 |
| 2003 | 207 | 679 | 679 | 210 |
| 2004 | 256 | 2997 | 410 | - |
| 2005 | 445 | 4634 | 2501 | 610 |
| 2006 | 611 | 1347 | 661 | 3210 |
| 2007 | 450 | 712 | 122 | 130 |
| 2008 | 688 | 2625 | 2318 | 2044 |
| 2009 | 839 | 1934 | 1379 | 1570 |
| 2010 | 424 | 1470 | 867 | 903 |
| 2011 | 689 | 6030 | 4310 | 6930 |
| 2012 | 355 | 763 | 563 | 210 |
| 2013 | 448 | 185 | 180 | 30 |
| 2014 | 716 | 5296 | 608 | 480 |
| 2015 | 993 | - | - | - |
| 2016 | 748 | - | - | - |
| 2017 | 600 | - | - | - |
| 2018 | 881 | - | - | - |
| 2019 | 1149 | - | - | - |

**Table C1.** Comparison of different volcanic emission inventories (annual $SO_2$ in kt): Volcanic $SO_2$ emissions reaching the stratosphere and the uppermost tropical troposphere from the Volcanic Sulfur Emission Inventory (Table 2) in this study, ending in August 2019 (+ about 19 MtSO$_2$ per year from degassing into the troposphere, Diehl et al. (2012)); total annual amount and explosive annual amount of global volcanic $SO_2$ emissions, calculated from satellite observations in 1979 to 2014 by Carn et al. (2016, Table 3), and total $SO_2$ emissions from Mills et al. (2016, Table S4).

## C3  Comparison with different case studies for volcanic "point source" injections

To compare the simulations from this study using volcanic $SO_2$ as a spatially resolved $SO_2$ cloud instead of the traditional point source approach some case studies are shown in this chapter. For this purpose, simulations are performed with the same model setup and with point source injections using the TREXP (Jöckel et al., 2010) submodel. As case studies we compare different point source injection methods of volcanic $SO_2$ emissions with the EMAC model simulations using the 3-D perturbations, and MIPAS Level 2 observations for the Nabro eruption in 2011 and the Sarychev eruption in 2009.

To ensure identical boundary conditions, the point source simulations were started using the identical model setup, only the volcanic $SO_2$ injections differ from those performed in this study. In contrast to the method used in this study, the point source methods always require, additionally to injected total $SO_2$ mass derived from nadir instruments, assumptions on the altitude range and the duration of the eruption, and also the area of the initial plume. In the examples we use the settings of Mills et al. (2016) and the corresponding entries in Table 2. For the latter we assume as one option that the $SO_2$ mass is injected equally distributed between the latitude dependent lower minimum altitude given in the caption and approximately the listed altitude of maximum perturbation of mixing ratio, the injection time and duration is assumed to be the same as in Mills et al. (2016) or, if missing, in the Smithsonian database. Alternatively we assume that emissions begin about 1.2 km (2 to 3 model layers) above the minimum altitude.

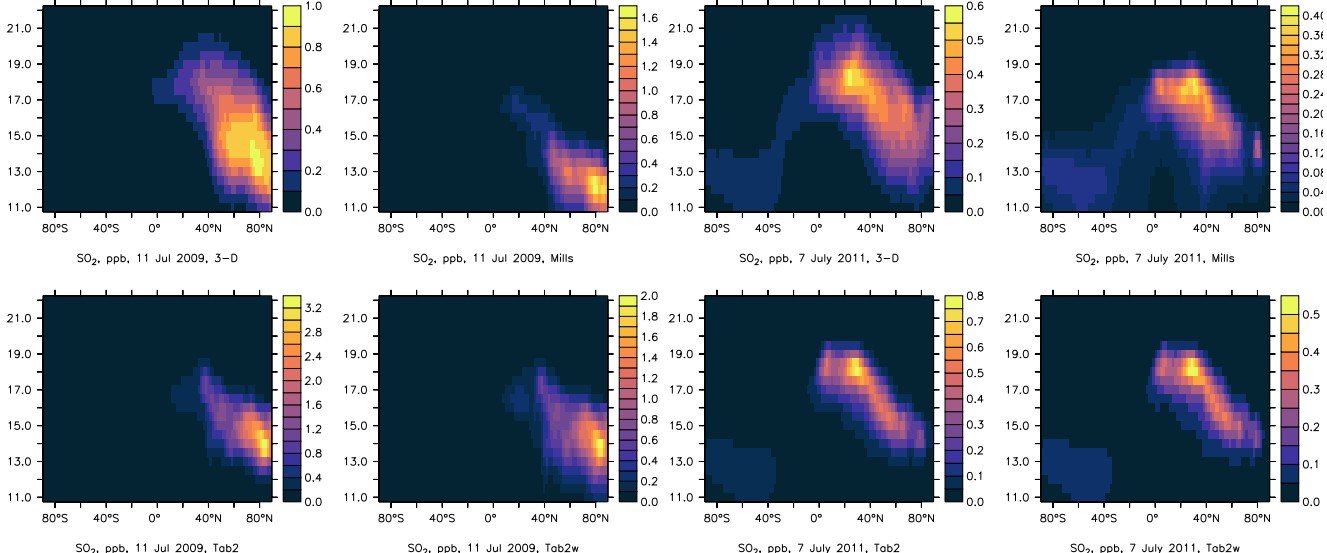

**Figure C2.** Zonal average $SO_2$ mixing ratio (ppb) distribution with latitude and altitude (km), 11 July 2009 (left 4 panels) and 7 July 2011 (right 4 panels). "3-D" this study, "Mills" emission settings of Mills et al. (2016), "Tab2w" emission in full columns of Table 2, "Tab2" emission starting about 1.2 km above minimum altitude of Table 2.

We compare zonal average vertical distributions of $SO_2$ (Figure C2) and sulfate (Figure C3) some weeks after the eruptions for the Sarychev eruption in 2009 (left) and the Nabro eruption in 2011 (right). In case of using the 3-D $SO_2$ perturbations the

resulting SO$_2$ distribution is smoother than with direct use of Table 2 for "point sources". Using Mills-"point sources" shifts the SO$_2$ plume to lower altitudes.

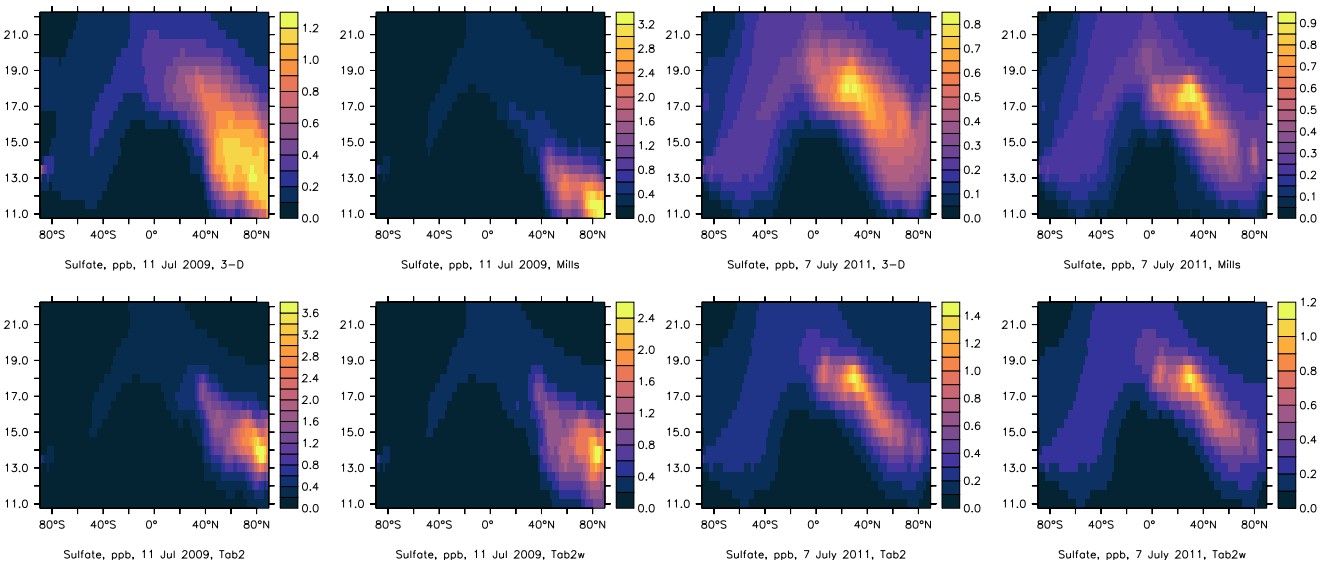

**Figure C3.** As Figure C2 but for sulfate in the accumulation mode.

Five-day-average radiative forcing at the TOA is shown in Figure C4. The black curves correspond to the red curves in Figure 11. Using the assumptions of Mills et al. (2016) on the vertical distribution of the SO$_2$ injection leads to less forcing in case of Nabro and after about 5 weeks after the Sarychev eruption (Figure C4 red curves). For the latter the peak is stronger since more mass is injected than in the other approaches. It is dominated by contributions from aerosol in the mid and high latitude lowermost stratosphere. Using Table 2 leads to approximate agreement if the provided altitude is about the top of the column into which is injected, using as a bottom about 1.2 km above the latitude dependent minimum altitude (blue curves). Using the full range leads to an underestimate compared to the simulation using the MIPAS observations directly as 3-D SO$_2$ perturbation (light blue curves). This is due to more efficient removal processes in the upper troposphere and the lowermost stratosphere than in the layers above. The lower panel for 2009 contains also a sensitivity study similar to the blue curve but with the eruption of Mando Hararo neglected as done by Mills et al. (2016) (dashed), and one where the top of the injection column for Sarychev was one layer lower than in the case with the light blue curve (purple). These examples show that the altitude of the injection has a large impact on the radiative forcing, but also that not only medium size eruptions matter. For stratospheric AOD shown in the upper panels of Figure C4 the difference between the approaches is less than for the forcing in the first weeks after the eruption. Later the SAOD decreases faster with the point source approach, especially the one of Mills et al. (2016), than with our method.

Note that considering only aerosol above the fixed pressure level of 100 hPa for the forcing calculations as we did in earlier studies causes misleading results here.

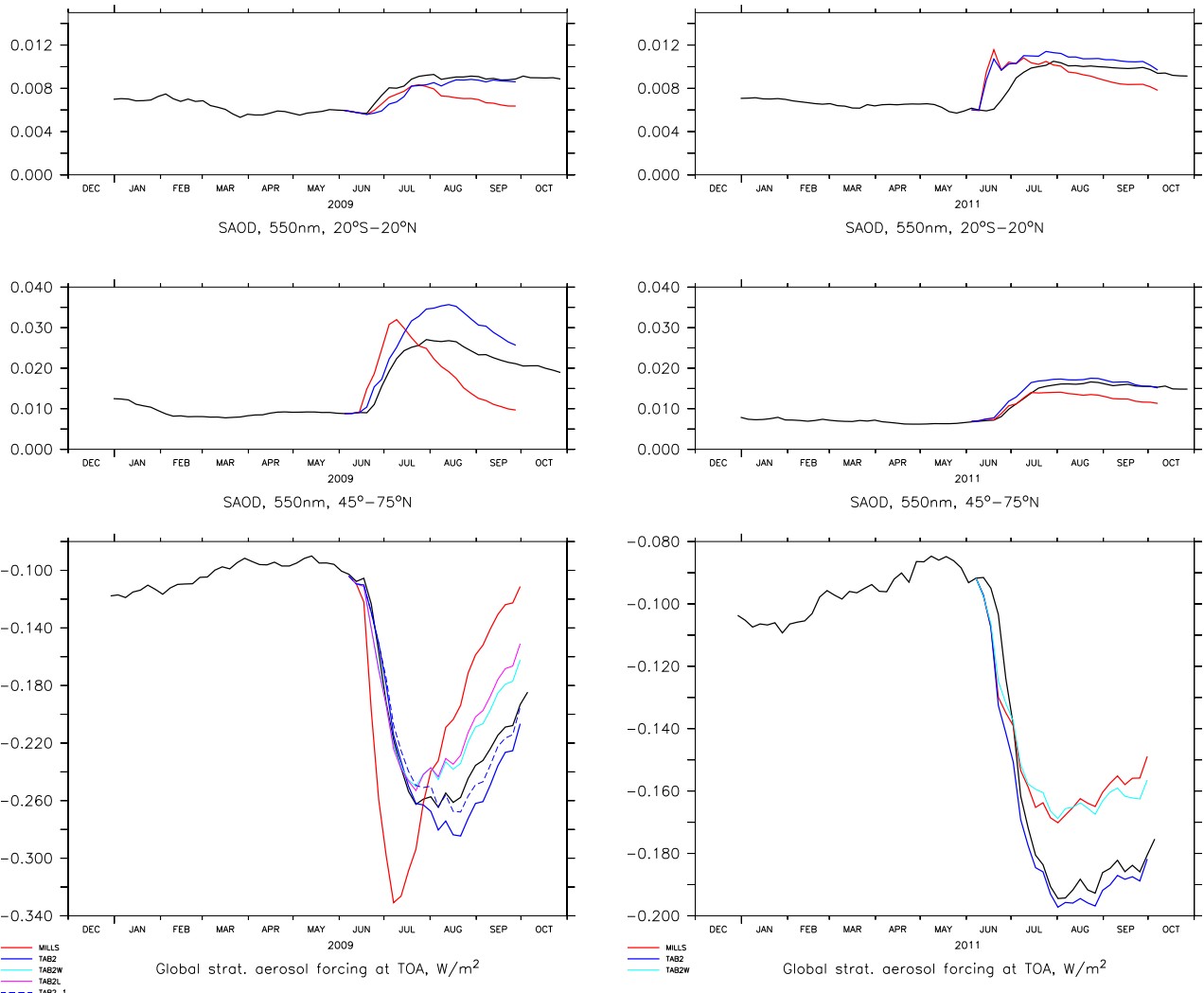

**Figure C4.** Stratospheric AOD (upper) and global radiative forcing at TOA (lower), Sarychev (left) and Nabro (right), black line as the red line in Figure 9 or Figure 11, blue, light blue and purple lines "point sources" based on Table 2 with different thickness and vertical position of the column into which is injected (see text), red lines with assumptions of Mills et al. (2016). Dashed blue, light blue and purple curves see text.

We further show maps at 16 and 18 km altitude and the corresponding Level 2 MIPAS observations (see Figure C5 to C6). In case of Nabro using column emissions up to about 18 km interaction of chemistry, radiative heating and dynamics leads to the formation of a lofting anticyclonic vortex with elevated concentrations of $SO_2$ and sulfate above 18 km, propagating westward for several weeks. In Figure C6 this is visible over the subtropical East Pacific but not supported by observations in this case. The phenomenon appears to be similar to the one observed after the 2019/2020 wildfires over the South Pacific (Khaykin et al., 2020).

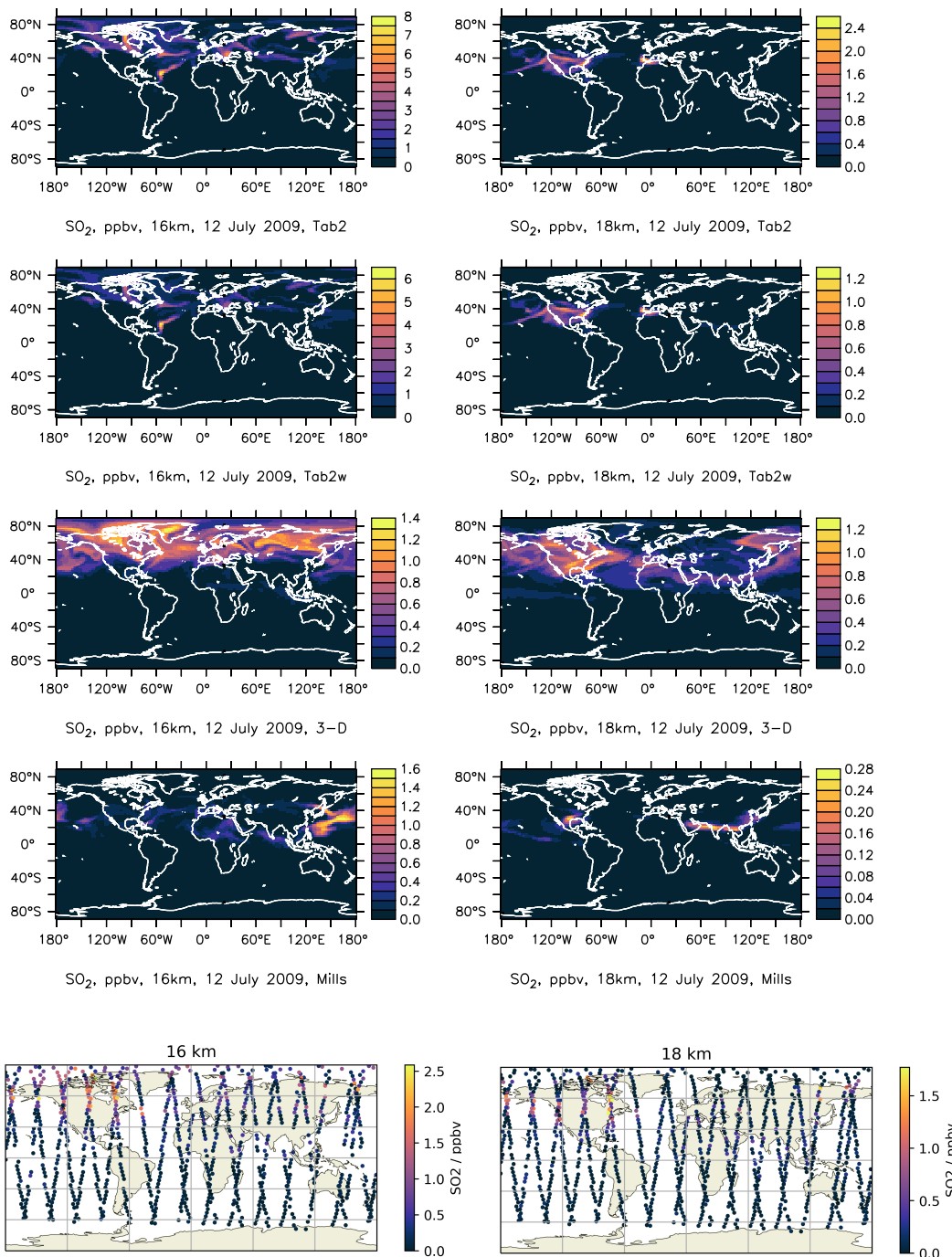

**Figure C5.** SO$_2$ mixing ratio (ppb) maps after the Sarychev eruption at 16 km (left column) and 18 km (right column) for point sources of Table 2 in column between 15 and 17 km (1st row), column between 13 and 17 km (2nd row), our 3-D-approach (3rd row) and the point source assumption of Mills et al. (2016) with a column below 15 km (Mando Hararo neglected) (4th row). Bottom row from MIPAS observations.

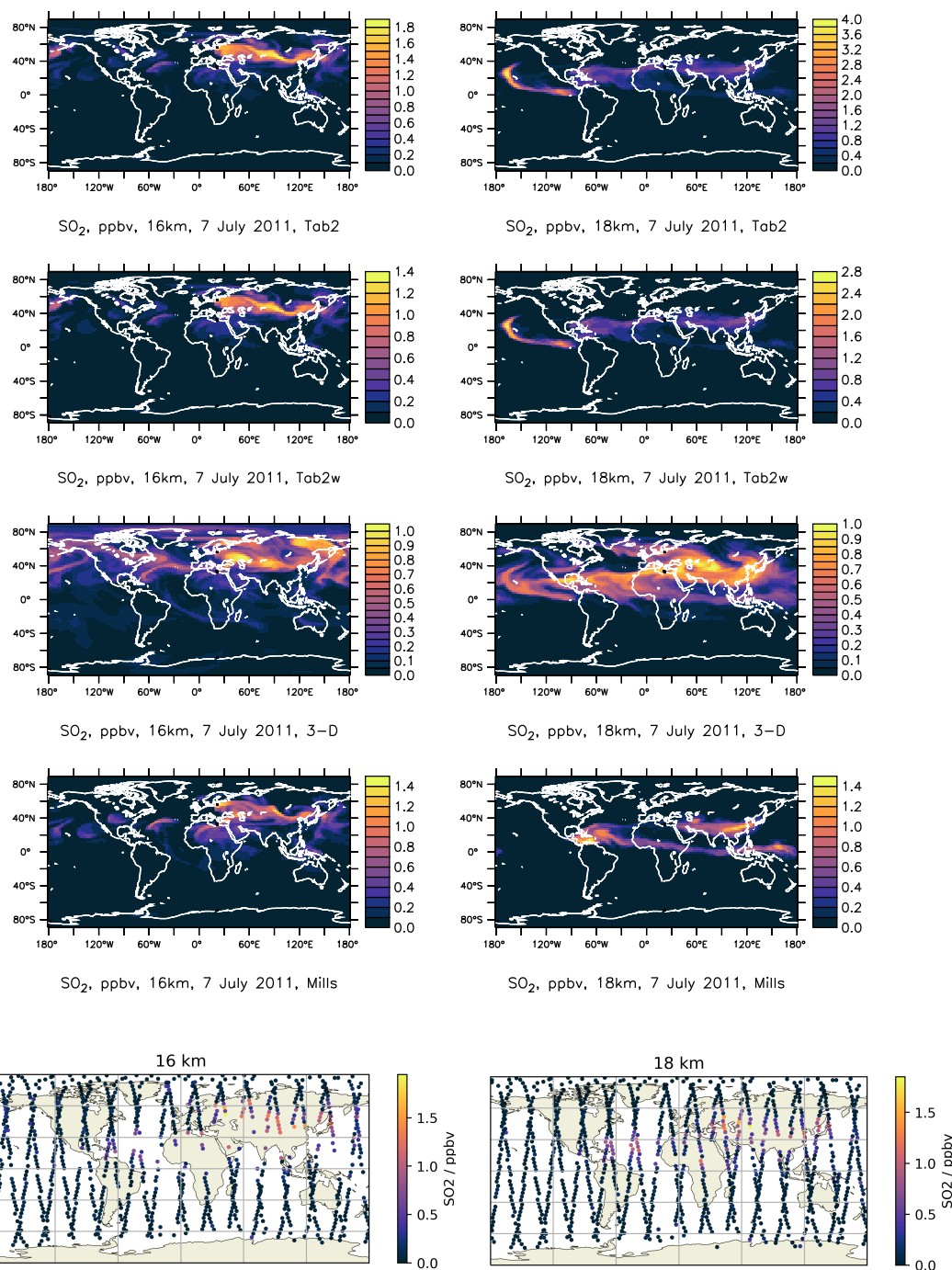

**Figure C6.** SO$_2$ mixing ratio (ppb) maps after the Nabro eruption at 16 km (left column) and 18 km (right column) for point source of Table 2 in column between 15.5 and 18 km (1st row), column between 14 and 18 km (2nd row), our 3-D-approach (3rd row) and the point source assumption of Mills et al. (2016) with a column below 16 km (4th row). Bottom row from MIPAS observations.

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
