# Peer review of "Reconstructing volcanic radiative forcing since 1990, using a comprehensive emission inventory and spatially resolved sulfur injections from satellite data in a chemistry-climate model"

_Atmospheric Chemistry and Physics, 2021_

## Author Comment (AC2)

We thank the reviewer for making very useful suggestions to improve the paper. Our point-by-point responses to the reviewers' comments and corresponding changes are detailed below in blue text, and the changes are shown in the version of the manuscript with track changes.

**Comment on acp-2021-654**

Anonymous Referee #1

Referee comment on "Radiative forcing by volcanic eruptions since 1990, calculated with a chemistry-climate model and a new emission inventory based on vertically resolved satellite measurements" by Jennifer Schallock et al., Atmos. Chem. Phys. Discuss., https://doi.org/10.5194/acp-2021-654-RC1, 2021

This paper focuses on the injection of SO2 to the stratosphere by volcanic eruptions, and the resulting variability of the stratospheric aerosol layer. It presents a new volcanic SO2 emission database, derived from a collection of satellite instruments, covering the period 1990-2019. It also presents results from a chemistry climate model which uses the updated injection database, and compares the results of the model to various satellite data sets, focusing on the multi-wavelength aerosol optical depth and instantaneous radiative forcing produced by the aerosols.

The construction of such detailed SO2 injection estimates covering the 1990-2019 period is an impressive accomplishment. It is also to my knowledge quite novel, as I believe it is the first attempt to produce SO2 injection values from sulfate aerosol extinction measurements. Unfortunately, the description of the methods used to produce these estimates is lacking. Furthermore, assumptions and choices made in the methods are not given justification. More detailed comments are included in "Major Comments" below.

Chemistry climate model simulations using the new SO2 injection data set are performed and some results shown. Good agreement with observations is achieved, but there is insufficient analysis to provide any improved understanding of the physical or chemical processes that control stratospheric aerosol evolution.

**Major comments:**

The description of how SO2 amounts were calculated lacks sufficient detail. I am not aware of any other study that has estimated SO2 injection amounts based on aerosol extinction measurements. This is thus a novel technique, but the method used is not described beyond a few statements along the lines of "The SO2 mixing ratio perturbation is derived from the extinction perturbation observed in a 10-day period beginning about a week after the eruption by dividing by air density, multiplying by a constant and subtracting a typical background." This explains extremely little: what constant is used, and why? How is the typical background determined? How well can the volume of the aerosol cloud be estimated a week after eruption from the satellite measurements? SAGE in particular has a very sparse sampling density, how does this impact the estimates? Can the method be validated? It would seem that the method could be applied to SAGE and OSIRIS during periods of overlap with MIPAS and the values from the new method compared to the "direct" MIPAS measurements. This would help increase confidence in the method, and provide some idea of the uncertainties in the estimates.

-> Response: We added a detailed explanation with case studies in Appendix B:

"The eruption of Reventador in the tropics in November 2002 has shown to be an ideal case where simultaneous observations of all satellite sensors were available so that the direct SO$_2$ observation could be used for development and validation of a conversion formula for the 750 nm extinction seen by GOMOS and OSIRIS, which works also approximately for SAGE if its observations at 530 and 1025 nm are interpolated to 750 nm. Here we first use the ratio between model calculated sulfate volume mixing ratio and its share on extinction in low latitudes of the lower stratosphere which is typically

1.2×$10^{12}$×air density (in molecules/$cm^3$). This works for medium size eruptions and data available over about four weeks following the eruptions, and if no other events occur less than about four weeks before which is the case for the Reventador eruption. If the time lag of data is several weeks a correction factor >1 has to be applied to account for removal processes, if another event is relatively close in time, the factor has to be <1 to remove the influence of the previous event. For Reventador the factor is 1 (for OSIRIS 0.8 is slightly better). From all instruments the derived injected $SO_2$ mass is very close to 77 kt as shown in Table 2. The spatial patterns are similar, except when the zonal wind causes a shift in longitude due to the time lag from conversion of $SO_2$ to aerosol, see Figure B1. In the case of SAGE, the alternate method of Grainger et al. (1995) involving aerosol surface area density (SAD) and aerosol volume density is more suitable to remove cloud perturbation. It is assumed that sulfate mixing ratios correspond to the $SO_2$ injected. Some uncertainty remains from removing the background which we have done by subtracting a fraction of the derived $SO_2$ at the longitude where it has a minimum, i.e. the longitude where the effect of the volcano is smallest for all altitudes. Integrated injected $SO_2$ masses for all examples are provided in Table B1.

For the eruption of Merapi in November 2010 the satellite instruments do not agree. From OSIRIS about 70% more injected $SO_2$ is derived than from MIPAS, i.e. 170 kt instead of 97 kt used in the transient simulation (see Table 2 and differences in Figure 10). GOMOS has too sparse data here to obtain a proper integral directly but patterns are similar (Figure B2). If other information is available, the gaps can be filled with likely values in the region where the plume was seen, a method which had to be applied also to some events seen by OSIRIS in 2018 and 2019 for which the data were sparse.

For high latitude eruptions the longer conversion time of $SO_2$ to sulfate compared to the tropics has to be considered which, together with aerosol removal processes, lead to a weaker extinction signal. To account for this a correction factor of about two in the conversion formula for OSIRIS for example for Sarychev in June 2009 leads to values consistent to the ones derived by MIPAS (Figure B3). For the low latitude eruption of Mando Hararo in the same entry of Table 2 (separated at 24° N for the integration) the factor 1 is still appropriate."

I highly recommend that the emission database be provided as an electronic supplement (e.g., csv or xls), to allow it to be readily used by other researchers.
-> Response: The input data files and model output of EMAC used here are stored at DKRZ, Hamburg, the volcanic inventory and the output for radiative forcing also at WDCC https://doi.org/10.26050/WDCC/SSIRC_3

The table, as text, presently takes up almost 8 pages of the manuscript: it would be more efficient to visualize the data somehow and include the values as supplemental information. Also, I strongly suggest that the format of the table be modified so that each individual eruption be listed per row, even if there are multiple eruptions on a given date. This will greatly improve the ease in which the data can be read within a computer program and thus used in other studies.
-> Response: $SO_2$ mixing ratios from the volcanic emission inventory are shown in Fig. 6. As an essential part of the novelty of this paper, the table should remain in the text because it is a comprehensive reference that cannot be represented by a single visualization. Additionally, the table is available for data processing as formatted ascii at this link: https://doi.org/10.26050/WDCC/SSIRC_3

The model results show good agreement with observations, but it's impossible to know whether the improved agreement (compared to prior works from the same group) is a result of the updated SO2 injection data, or to model improvements or changes in model resolution. Given the theme of the ACP journal, the reader expects that this work should improve our understanding of the chemical and/or physical processes that control stratospheric aerosol evolution, but it remains unclear if there is any improvement in understanding being extracted from the study. Nor is there any real motivation or objectives stated in the introduction for the model simulations.
-> Response: The improvements are mostly due to use of more satellite data for volcanic SO2 using a novel method. This is mentioned now at several places including the abstract. Model improvements include consideration of aerosol effects in the photolysis rate calculation (minor effects here) and compared to Bingen et al. (2017) a finer horizontal resolution (see section 3).

**Specific comments:**

L11: "Reproduce" is too strong
 -> Corrected: are consistent with

L12: Here it is said that "slight deviations … were found only for the large volcanic eruption of Pinatubo in 1991", but later in the document deviations in other time periods, e.g., 2010 are discussed, so this is inconsistent.
-> Removed:

L19: precise language is needed here, is this the peak radiative forcing produced by a typical "small" eruption, or the time average forcing from these eruptions? And what is a small or medium eruption? Also, it's not clear how this number is estimated, a value of 0.10 W/m^2 is not mentioned in the results or conclusions, and if it comes from Fig 11, how is the effect of small eruptions separated from that of "background" sulfur (e.g., DMS, OCS) transported into the stratosphere via atmospheric circulation?
-> Response: This number can be taken from Fig. 11 for volcanically quiescent periods or periods between medium size eruptions. Background is about 0.04 W/m$^2$ (not shown explicitly, taken from a simulation with much less volcanoes). This includes organics and dust.

L22-24: references needed for these statements.
-> Cited: Kloss et al., 2019 and Vernier et al., 2011

L25: I believe Bruehl et al., 2015 were not making the actual measurements of the size distribution of stratospheric aerosols. Better reference needed.
-> Cited: Wilson et al., 2008

L31: part of the aims stated here is apparently related to the interaction of aerosols with ozone, but this is not shown in the manuscript.
-> Response: text modified but this is not quantified in the text except for a number related to forcing.

L34ff: Reference(s) needed.
-> Cited: Vernier et al., 2011

L37: I am skeptical of a 3-year upper limit on the impact from volcanic eruptions: if ocean temperatures are a part of "climate", then there is good evidence that volcanic impacts on climate can last much longer than 3 years (e.g., McGregor et al., 2015). Obviously the period of impact depends on many factors, but we should be careful to not overly simplify statements which might be misleading to some readers.
-> Corrected: " These changes influence in turn the radiative forcing at tropopause altitudes (or at the top of the atmosphere) for several years after the eruptions (Timmreck et al. 2012) and can even have a more prolonged impact on the global climate (McGregor et al. 2015)."

L44: Reference(s)?
-> Added citation: Solomon et al. 2011

L65: Some information should be given on how the SO2 column data was used, especially in regards to how a stratospheric component was estimated from the full column.
-> Response: In the case of data gaps for the four main satellite instruments used, the data from additional satellites e.g. TOMS, OMI or OMPS are applied to double check, if available. Especially in 2018 and 2019 the OSIRIS data are so sparse that constraints from instruments like OMPS or analogues events of previous years have to be superimposed for some eruptions.

L130: The gaps in spatial coverage of the OSIRIS data at 17 km extend significantly beyond the polar night: they seem to extend even in best cases to 20-30deg. Some rephrasing needed.

-> Response: OSIRIS provides a surface coverage from 82° S--82° N, except in polar winter when there is no sunlight and except in the Southern Hemisphere winter when tangent point is not illuminated by the sun

L136: It's not apparent how the sensitivity to clouds can be seen in Figure 4.
-> Response: At altitudes near and below the tropopause, the OSIRIS measurements are sensitive to clouds that may be interpreted as elevated aerosols. This is likely contributing to larger background extinction values measured below approximately 17 km in the tropics, as can be seen in Figure 4.

L140: How is the correction factor determined? This sounds suspiciously like numbers have been chosen only to produce best agreement.
-> Response: To estimate the factor, we iterated calculated extinctions to agree with OSIRIS and also used observations and assumptions by Vernier et al. (2016). A detailed description with 3 case studies is added to the Appendix: "If the time lag of data is several weeks a correction factor >1 has to be applied to account for removal processes, if another event is relatively close in time, the factor has to be <1 to remove the influence of the previous event."

L157: The study of Grainger et al. (1995) does not seem to provide a relationship between SAD and SO2 mixing ratio. More explanation needed.
-> Response: This is skipped here. Now we write in Section 4: "For SAGE II in most cases the SO2 mixing ratio is derived using the parameterisation of Grainger et al. (1995) which converts SAD to volume density as a first step. We use the pressure and temperature provided to convert from mass density to a volume mixing ratio, assuming that observed sulfate is produced from injected SO2 some weeks ago. With this method it is easier to correct for cloud contamination than by using the extinction directly as above for the other instruments. Case studies for three events, comparing SO2 results from the different satellites and the different conversion methods are presented in Appendix B".

L190: It is not clear how differences in the "vertical transport of tracers, like dust and water vapor or ozone" between model resolutions has any importance to the present study.
-> Response: This is part of the general setting of the model simulations and has been moved to the appendix.

L216: What parameters?
-> Response: We removed "parameters" by: "aerosol optical properties like wavelength- dependent particle extinction cross section, single scattering albedo, and asymmetry parameter for each aerosol mode from AEROPT (Dietmüller et al. 2016) "

L218ff: The double radiation call most likely calculates the "instantaneous radiative forcing". It is important to be clear about this and consistent with the terminology.
-> Added: "instantaneous" radiative forcing

L219: There is a double radiation call, but how exactly is the radiative forcing calculated?
-> Correction: "instantaneous" radiative forcing, "..by taking the difference of the net total fluxes at 100 hPa or TOA."

L220: Not understanding this, are you diagnosing the impact of volcanic aerosol on upper stratospheric UV absorption? Nothing like this is shown in the results.
-> Response: The description of the RAD\FUBRAD sub-submodel is part of the general setting of the model simulations and has been moved to the appendix.

L241: What is the justification for the lower limits to the vertical integration given? You use 12 km as the lower limit in high latitudes, but the climatological tropopause height in high latitudes is 9-10 km. Conversely, you use 14 km in low latitudes, but the tropopause there is around 17 km. A thorough explanation for these counterintuitive thresholds will need to be given.
-> Response: The lower limit of 12 km altitude at high latitudes was chosen based on the signal-to-noise ratio, uncertainties for low altitudes, and clouds in the volume mixing ratio profiles obtained by

MIPAS and the other used satellite instruments (e.g. effects of frontal systems). In the tropics, we set the lower limit at 14 km to account for transport processes in the UTLS layer, especially during the Asian summer monsoon. Here in our extraction scheme we exclude cloud contaminated regions (see section 4 and 2).

L251: An "integration time" has not been introduced, it is not clear what this means in terms of the method.
-> Response: The temporal resolution of the satellite data is 5 days (for MIPAS, GOMOS and OSIRIS). The integration time is the case dependent time period used for the single eruptions. "The integration time (i.e the used time period)"

Table 2: There are a number of cases where the number of values do not match between the different columns in a particular row, e.g., 11 Feb 1990, 19 Aug 1992, 18 Sep 1996. Expanding the table so each eruption is listed in a single row would help this issue, as well as improve the machine readability of the table more generally. There is also a case (14 Jan 2002) where values are listed within brackets, and I did not find an explanation for what this means.
-> Response: These issues have been corrected, the table is available now in a Fortran formatted form and it has to be consistent to the 3D netcdf SO2 pertubation files provided in https://doi.org/10.26050/WDCC/SSIRC_3, which often contains multiple events. A single event list with more than 500 entries would not be acceptable for the ACP readership.

Table 2: The methods used produce an estimate of about 17 Tg for Pinatubo, which is in line with direct measurements of SO2 (e.g., Guo et al., 2004), but in contrast to recent model studies which suggest the effective injection for Pinatubo was much less (e.g., Mills et al., 2016; Dhomse et al., 2014). Some discussion of this issue would fit well into the paper.
-> Added in sec. 6.2:
"In this study about 17 Tg SO2 are injected for the Pinatubo eruption (Guo et al. 2004). Model comparisons by Timmreck et al. (2018) show that the span of used injections varies between 10 Tg SO2 (e. g. Dhomse et al. (2014); Mills et al. (2016); Schmidt et al. (2018)) and 20 Tg SO2 (e.g. English et al. (2013)). Thus, this study is in the middle range of the injected sulfur mass.
On the other hand, filling the gaps in the SAGE data just by horizontal linear interpolation increases the peak AOD by about a factor of 2, which is close to the GloSSAC compilation. In Figure 10 the AVHRR (Advanced Very High Resolution Radiometer) by Long and Stowe (1994) at 630 nm are included, which are close to our simulations (if converted to 750 nm their AOD would be slightly less or to 550 nm slightly larger).
When comparing the EMAC simulations (red line in Figure 9) with the simulation of Schmidt et al. (2018, fig.1) (black line in Figure 9, lower panel) it can be recognized that a smaller value for the peak of the Pinatubo eruption occurs, but here it needs to be considered that Schmidt et al. (2018) are using monthly global-means. This has the consequence that the signal of single eruptions is blurred and smaller sized eruptions cannot be easily identified."

L269: Mixing ratios appear quite variable, what is meant here by "typical"?
-> Response: altitude range added in text: ("25 to 29 km"), this refers also MIPAS observations in that altitude (Höpfner et al. 2013).

L271: What upper limit is referred to here?
-> Added in text: "consistent to the Smithsonian reports, SAGE and TOMS".

L281: References should be included to support this statement on the transport of aerosols from Nabro.
 -> Done: Clarisse et al. 2014

L290: "The comparison of the simulated and observed SO2 values" is really hard to do since Figures 1 and 6 use different units and color schemes. It would be helpful to extract the MIPAS years from the simulations and show them with the same units and color scheme in comparison to the observations.
 -> Response: figure 1 replotted using the same unit and color scheme as in figure 6.

L293: Is the statement on SO2 lifetimes made here a result of this study, or are the lifetimes equivalent to those given by Hoepfner et al. (2015)? If the result is the same as Hoepfner et al., (2015), that should be explicitly stated. If estimated lifetime are different from Hopefner et al. (2015), how and why?

-> Response: "Generally, the conversion of SO2 to sulfate aerosol particles depends on several factors, such as the altitude, latitude, or season of the eruption and takes according to Höpfner et al. (2015) about 13, 23 and 32 days in 10–14, 14–18 and 18–22 km altitude, respectively, in midlatitudes. Carn et al. (2016) report an e-folding time varying between 2–40 days. The range agrees with our simulations (and assumptions in section 4)."

L300: This sentence seems to say that stratospheric aerosol optical properties were calculated using a range of different aerosol types (sulfate, dust etc.). Is this correct, or is the sentence just misleading?

-> Response: Yes, this is correct. See Dietmüller et al. (2016):

"Aerosol species explicitly considered are water soluble inorganic ions (WASO), black carbon (BC), organic carbon (OC), sea salt (SS), mineral dust (DU), and aerosol water (H2O). The refractive indices for those aerosol species are extracted from various data sources (most of the data are compiled in the HITRAN2004 database) and include wavelength dependencies. … The refractive indices for each aerosol mode required as input for the lookup tables are calculated assuming an internal mixture of the aerosol components for the hydrophilic modes. A mean refractive index is calculated for each mode wavelength combination by averaging the refractive indices of the individual components weighted with their volume contributions. The corresponding Mie size parameters are derived from the median radii of the log-normally distributed modes and the respective wavelengths. The wavelength- dependent particle extinction cross section, single scattering albedo, and asymmetry parameter for each mode are then obtained from the lookup table for the appropriate modal width (σ)."

L330: The OSIRIS data is converted from 750 nm to 550 nm, which is fine, but this contradicts the statement just a couple sentences earlier that "Unlike most other studies, the stratospheric AOD is compared at the original wavelengths derived from different optical channels of the satellite instrument measurements."

-> Response: The model calculates the AOD and other optical properties directly, derived from the original wavelengths of the satellite data. For additional comparisons only, the satellite data of OSIRIS were converted to 550 nm by the cited authors.

L333: The statement that "differences after the large Pinatubo eruption in 1991 between the model simulations and the SAGE II observations are related to the "saturation" effects of the satellite instrument" seems much too confidently worded. It seems quite possible that "saturation" effects explain some of the difference, but how certain can you be sure that it is the only, or even the primary reason? In the tropics, the simulated AOD appears to be ~3 times larger than the SAGE II measurements—is it likely that the SAGE II measurement is so strong an underestimate of the true total AOD?

-> Response: See AVHRR data now included. Filling the SAGE data gaps in the lower stratosphere just by linear interpolation in the first year after the eruption increases AOD by about a factor of 2.

Fig. 11: The ERBE measurements are not described at all in the text. Are they anomalies? What is the global coverage of the measurements? Likewise, the data from Solomon is only mentioned in passing in the text, and a little more detail should be included on how those radiative forcing estimates were calculated.

-> Response: The ERBE data shown in Fig. 11 are estimated averages for solar forcing at the top of the atmosphere, with 72-day means in the near-global (60◦ S–60◦ N) data set. Details are in Wong et al., (2006) and Toohey et al., (2011). (in caption of Fig.11)

L352: "The new model simulations with the additional volcanic eruptions (red line) are closer to the calculated estimates from satellite extinction measurements of SAGE, GOMOS and CALIOP (Cloud-Aerosol Lidar with Orthogonal Polarization) by Solomon et al. (2011) (green crosses) than in previous studies (e.g., Brühl et al. (2015))." This statement, a concrete conclusion of the study, is impossible for

the reader to verify without accessing the prior study, finding the relevant figure, and trying to visually compare the two. This is asking too much of the reader. Please include the result of Bruehl et al. (2015) directly on Fig 11 here so we can directly assess the validity of this statement.

-> Response: For direct comparison, the results of Brühl et al. (2015, 2018) (pink lines) are now included in Fig. 9-11.

L361: Are the results of Minnis et al. (1993) equivalent to the ERBE data shown in Fig 11? Please clarify.
-> Response: Minnis et al. (1993) derived their results from the ERBE data.

L362: clarify that the *simulated* AOD drops too quickly compared to the observations.
-> Corrected: "the simulated instantaneous global negative radiative forcing drops again too quickly for the EMAC simulations (red line) after the Pinatubo eruption compared to the observations."

L374: "2019" is not an eruption.
-> Corrected: including the Raikoke eruption in 2019

L375ff: This paragraph is quoting results from other papers, not showing work from this study. If these statements are important, they should be moved out of the Results section or linked directly with results of the study.
-> Corrected: moved to introduction

L385: The fact that this study uses a higher resolution model than previous studies should have been mentioned earlier, in the model description and/or introduction.
-> Added in Sec. 3.1:
"For these model simulations, a higher horizontal resolution T63 (1.87°×1.87°), instead of T42 (2.81°×2.81°) in Bingen et al (2017), was chosen."

L386: This appears to be a result of the study by Bruehl et al. (2018), which would be important in describing the experiment earlier in the manuscript but not here in the conclusions.
-> Response: Skipped here (see above).

L388: The SAGE II and OSIRIS extinction measurements are not really "newly available", some version of this data has been available for many years. The estimation of SO2 from these data sets is quite new—it's what this paper is presenting!
-> Response: The resolution of the updated versions is improved, so OSIRIS data has allowed comparisons up to be extended to 2019; additional SAGE II data was also used extend the comparison back to 1990 (together with the Smithsonian database).

L402ff: This conclusion is not supported by the results: there is no quantification of the impact the increased number of eruptions included in the database has on the radiative forcing, or its level of agreement with observations.
-> Removed.

L408ff: This is an interesting conclusion, but it is not supported by the results. There is no demonstration that including the injections below the tropical tropopause improves the agreement. Even a comparison with prior studies will not prove necessarily support the statement since those prior studies used a different resolution model.
-> Response: Remarks on that are added at several places in other sections.

L418: This is not a new result, as it has been shown by prior studies.
-> Response: Paragraph skipped. Instead we included now in the paragraph beginning with:
"Our volcanic sulfur emission inventory…." "The inclusion of plenty of small size eruptions reaching the UTLS has the consequence that stratospheric aerosol optical depth and radiative forcing does not decrease to almost zero between medium size eruptions in agreement with observations, in contrast to a lot of other studies."

L422: The impact of volcanic aerosol on tropical upwelling is not diagnosed in this study. Prior studies have explored this, but statements like this can not be included in the conclusions of this work if there are no new results shown to support it and build upon prior work.
-> Moved to introduction.

L437ff: This paragraph talks about meteoritic dust, which was not investigated in the study. Perhaps simply adding a sentence or two on the agreement between the model and observed aerosol extinction in the upper stratosphere to motivate the discussion of meteoritic dust would help the reader follow the logic here.
-> Removed: Remark added to sec.6.1: "Above about 24 km altitude, EMAC underestimates the observations because in the model meteoric dust particles were not considered."

448: Confirming the findings of the IPCC report is, firstly, incorrectly phrased, since the IPCC report only summarizes and reports findings gathered from the published literature. It would be more important to compare the results here with the primary sources, including studies that have been published since the IPCC AR5 (e.g., Schmidt et al., 2018). Second, confirming some general results from prior studies does not make an overwhelming case for publication. What does this study add to the understanding of volcanic radiative forcing that wasn't known before?
-> Response: Paragraph shortened, restricted on new results.

L450: Radiative forcing is stated to be that at the surface here, where Fig 11 is said to be RF at the tropopause. Also the numbers quoted here don't seem to agree with Fig 11. It would be best to only refer to calculations for which the results are shown in the paper.
-> Response: Typo corrected, numbers refer to difference to quiescent periods which is now mentioned.

**Editorial comments:**

Line 9: Volcanic SO2 is not "pollution" in the usual sense of the word, suggest it be cut here.
-> Corrected.

L49: "Distribution"?
 -> Corrected.

L53: "constitute a source of background…"
 ->Corrected.

L55: Awkwardly phrased: the processes aren't structured, the paper is, and not strictly according to processes.
-> Corrected: "This paper is structured as follows:"

L80: I've never seen pptv written with v as a subscript, is this a new standard?
 -> Corrected.

L111: confusingly phrased.
-> Corrected concerning cited Figures. We have changed the sentence in:
"Figures 2 and 3 show the aerosol extinction from the GOMOS instrument at wavelengths of 550 nm (Figure 2) and 750 nm (Figure 3), respectively. In both cases, a gridded aerosol extinction dataset is used (CCI-GOMOS dataset in version 3.00, see Bingen et al., 2017)."

Added References:

Dhomse, S. S., Emmerson, K. M., Mann, G. W., Bellouin, N., Carslaw, K. S., Chipperfield, M. P., Hommel, R., Abraham, N. L., Telford, P., Braesicke, P., Dalvi, M., Johnson, C. E., O'Connor, F., Morgenstern, O.,

Pyle, J. A., Deshler, T., Zawodny, J. M. and Thomason, L. W.: Aerosol microphysics simulations of the Mt.~Pinatubo eruption with the UM-UKCA composition-climate model, Atmos. Chem. Phys., 14(20), 11221–11246, doi:10.5194/acp-14-11221-2014, 2014.

Guo, S., Bluth, G. J. S., Rose, W. I., Watson, I. M. and Prata, A. J.: Re-evaluation of SO 2 release of the 15 June 1991 Pinatubo eruption using ultraviolet and infrared satellite sensors, Geochemistry Geophys. Geosystems, 5(4), Q04001, doi:10.1029/2003GC000654, 2004.

McGregor, H. V., Evans, M. N., Goosse, H., Leduc, G., Martrat, B., Addison, J. A., Mortyn, P. G., Oppo, D. W., Seidenkrantz, M.-S., Sicre, M.-A., Phipps, S. J., Selvaraj, K., Thirumalai, K., Filipsson, H. L. and Ersek, V.: Robust global ocean cooling trend for the preindustrial Common Era, Nat. Geosci., 8(9), 671–677, doi:10.1038/ngeo2510, 2015.

Mills, M. J., Schmidt, A., Easter, R., Solomon, S., Kinnison, D. E., Ghan, S. J., Neely, R. R., Marsh, D. R., Conley, A., Bardeen, C. G. and Gettelman, A.: Global volcanic aerosol properties derived from emissions, 1990-2014, using CESM1(WACCM), J. Geophys. Res. Atmos., 121(5), 2332–2348, doi:10.1002/2015JD024290, 2016.

Schmidt, A., Mills, M. J., Ghan, S., Gregory, J. M., Allan, R. P., Andrews, T., Bardeen, C. G., Conley, A., Forster, P. M., Gettelman, A., Portmann, R. W., Solomon, S. and Toon, O. B.: Volcanic Radiative Forcing From 1979 to 2015, J. Geophys. Res. Atmos., 123(22), 12,491-12,508, doi:10.1029/2018JD028776, 2018.

(more in the acpd-preprint)

---

## Author Comment (AC3)

We thank Thomas Aubry for making very useful suggestions to improve the paper. Our point-by-point responses to the reviewers' comments and corresponding changes are detailed below in blue text, and the changes are shown in the version of the manuscript with track changes.

**Comment on acp-2021-654**

Thomas Aubry (Referee)

Referee comment on "Radiative forcing by volcanic eruptions since 1990, calculated with a chemistry-climate model and a new emission inventory based on vertically resolved satellite measurements" by Jennifer Schallock et al., Atmos. Chem. Phys. Discuss., https://doi.org/10.5194/acp-2021-654-RC2, 2021

This study investigates the impact of explosive volcanic eruptions on the stratospheric aerosol burden, optical depth and Earth's radiative balance using a chemistry-climatemodel and a new inventory of volcanic SO2 emissions. Comparison with satellite observations are presented and with the exception of the Pinatubo period, the simulations are shown to be in good agreement.

Although the overall methodology of this paper is not new, there exists very few studies of the historical stratospheric aerosol forcing that use chemistry-climate models and SO2 inventories and this paper is thus a useful contribution. Furthermore, the study brings two novel elements:
  i)       the use of a new volcanic SO2 emission inventory, argued to be more comprehensive and better compared to other inventories;
  ii)      ii) the strategy used to inject volcanic SO2 into the model, consisting in injecting 3D SO2 plumes instead of the standard "point source" injection. Unfortunately, I find these two points to be not sufficently motivated and explained (in the case of the second one), and analyses conducted do not enable to assess whether the new inventory and injection strategy result in improved volcanic forcing, which undermines the key contributions of this study. Furthermore, there is little to no comparison with previous work (e.g. different emission inventories, or different emission strategies). Many important references are lacking. To sum-up, I think this manuscript has the potential to become a really valuable paper for the community, but that further analyses as well as an improvement of the discussion section are still required.

**Major comments:**

1) The first novel aspect of the paper is the way in which volcanic SO2 is injected in the model. Previous studies have used a "point-source" approach with SO2 injected in one model column over a range of altitudes, with a few studies also injecting over a range of latitude for Pinatubo. However, in this study, the authors instead inject a "plume" consistent with spatially-resolved satellite observations. First, I think that this novel aspect is not highlighted enough in the introduction section and throughout the text, and it could be one of the key point of the manuscript.
-> Response: These points are now more highlighted in the introduction:
"For the ENVISAT (European Environmental Satellite) period 2002-2012 a first version of a new volcanic SO2 inventory with improved temporal and spatial resolution was developed within the framework of ISA-MIP (Timmreck et al. 2018, Brühl et al. 2018). The corresponding data base (link) contains 3D-SO2-perturbations derived from satellite data as well as integrated injected SO2 masses. In this work the data base is expanded to the period 1990-2019 and considerably improved for the period 1998-2001. The simultaneous measurements from up to four instruments from 2002 to 2011 enabled us to develop a novel procedure for conversion of aerosol extinction to SO2 needed for the period before and after ENVISAT.
Our method circumvents problems and uncertainties related to the classical point source approach like dependence on the box size and location, the time interval during which the mass is injected, and

effects of microphysical and chemical interactions of SO2 and sulfate with injected volcanic ash and water in the early phase (Zhu et al., 2020)."

I also find your new method to be poorly explained and justified, in particular in section 5. On line 264, you say that the total amount of SO2 is calculated by integrating the SO2 profile but then mentioned that you add a 3-dimensionnal perturbation to the model which confused me. In section 5, you also don't clearly state how these 3D plumes are obtained. My understanding from sections 3/4/5 is that:
-> Response: Parts of section 4 are rewritten to clarify the method. For illustrative purposes, a detailed description of case studies is added in Appendix B (see reply to referee 1).

For each eruption, 3D SO2 plumes are obtained from time-averaged SO2 observations between the 8th and 17th day following each eruption?
-> Response: Yes, 3D SO2 plumes are obtained from the satellite observations. The temporal resolution of MIPAS, GOMOS and OSIRIS data is 5-days. The chosen time interval for the integration of the emitted SO2 amount is case dependent for every single eruption, depending on data availability (data gaps, volcanic ash plumes, duration of the eruptions, etc.).
MIPAS SO2 data are used immediately after the eruption, extinction data with a time lag of about a week. This is now explained in section 4 in more detail for each data source.

The 3D plumes, obtained from measurement 8-17 days after the eruption, are injected at the time of the eruption. The 3D plumes are injected at latitude consistent with measurement taken but centered on the longitude of the volcano Did I get this right? It all need to be crystal-clear and more detailed in the text as this is key to your method and a very unusual approach?
-> Response: As mentioned in the text, it should be noted as well that the date of the volcanic eruption can differ by a few days from the date of injection in the model simulation, because the temporal resolution of the data sets is about five days at least (or weeks in the SAGE period).

You need to justify these choices better and show sensitivity tests for a large and small eruption (or ideally a full 1990-2019 simulation) showing how this differ from a standard "point" injection at the volcano location/plume height with a mass of SO2 corresponding to the initial total SO2 (not the SO2 after 8-17 days). Such tests seems really critical to demonstrate that your proposed method is better than standard methods, otherwise any related claim is unfounded.

One of the main justification you provide to justify your injection strategy is that it removes any tropospheric SO2 that is not climatically relevant but:

i) you already only consider SO2 above a threshold height (which is not justified; e.g. why 14km at the tropics instead of the tropopause height? If it's because of radiative heating and lofting where does the threshold come from?) so why do you need further processing to remove potential "shortlived" SO2?;
-> Response: The EMAC simulations include comprehensive tropospheric chemistry and various inventories of tropospheric sulfur emissions (SO2 from outgassing volcanoes, anthropogenic emissions, etc., as well as other sulfur-containing species such as DMS from ocean fluxes). Therefore, double counting of tropospheric volcanic emissions should be avoided.The lower limit of 12 km altitude at high latitudes was chosen based on the signal-to-noise ratio, uncertainties for low altitudes, and clouds in the volume mixing ratio profiles obtained by MIPAS and the other used satellite instruments. In the tropics, we set the lower limit at 14 km to account for transport processes in the UTLS layer, especially during the Asian summer monsoon and over the Andes. Here, we also have to consider high altitude clouds.

ii) The SO2 e-fold time is on the order of days-weeks (Carn et al. 2016, Fig 14); Even for stratospheric SO2 one would expect a significant amount of SO2 to be already converted to aerosol by the end of your 8-17 day time window, in particular for lower stratospheric injections. So would your method not result in large underestimation of SO2 amounts injected? I can see reasons why your method could make sense, e.g. fast SO2 scavenging

by ash during the first days-weeks, but I think it is still not justified enough in the paper. More importantly, you need to show comparison between your approach vs standard point injection with the full SO2 mass to be able to really discuss the strengths and weaknesses of your strategy.

-> Response: MIPAS directly provides SO2 data, while for the other 3 instruments we have to convert the aerosol extinction. Here we consider a time lag of some days to few weeks from the conversion of SO2 to aerosol. During the MIPAS period direct SO2 observation could be used for development and validation of a conversion formula for the 750 nm extinction seen by GOMOS and OSIRIS, which works also approximately for SAGE (see for detailed description the case studies in Appendix B).

2) Overall, your paper really lacks comparison with existing work – including that from Bruhl et al 2015 – and a lot of key references are missing. As an example, on line 245-247, you suggest that your SO2 mass estimates will be very different from those in the dataset by Carn et al. (2016). Why not show a figure, at least in SI, comparing SO2 masses and heights for all events in common? This would be really informative.

Regarding your simulations, you do not mention at all the work by Schmidt et al. (2018) which conducted exactly the same type of simulations, albeit with a different SO2 inventory and model. Citing it seems critical, and some of their time series (SAOD, radiative forcing) are likely available and could be compared to your model which would really improve the discussion.

Also, it would have been nice to see a comparison of your new simulations with the previous model version/inventory used by some of the co-authors (Bruhl et al 2015) to get a sense of whether there is improved agreement with observations.

-> Response: Pink lines with the results from Brühl et al. (2015) are added to Fig. 9-11. Schmidt et al. (2018) is available for global AOD at 550 nm and added as black line in Fig. 9, AOD at 750 nm (Fig. 10) is not shown by Schmidt et al. (2018). In Fig. 11 the data for volcanic effective radiative forcing from Schmidt et al. (2018) is added by a black line.

A comparison with Carn et al. (2016) for annual sums is added in Appendix C.

Last, you compare your simulations with observations from multiple satellite instruments which is welcome, but I was under the impression that the GloSSAC dataset – built using some of the data you use – is now the reference for the community (at least for CMIP6 forcing). Could you add a comparison to GloSSAC?

-> Response: GLOSSAC (Thomason et al. 2018), a time dependent aerosol climatology sometimes used for climate simulations, has a coarse temporal resolution, and many discontinuities in space and time due to change of instruments or gaps and excludes important satellite data (e.g. MIPAS). It does not provide SO2 (sulfur dioxide) needed as input for chemistry climate models directly but only extinction and highly derived quantities like estimates for aerosol surface area density and mode radius assuming unrealistic monomodal size distributions. (Added in introduction)

A line for GLOSSAC is added in Fig.9 for convenience.

**Minor comments**

Title: I think the title does not convey clearly enough the novelty of the new injection method; consider replacing "vertically-resolved satellite measurements" by something else? Maybe "Reconstructing volcanic forcing since 1990 using a comprehensive volcanic emission inventory and spatially resolved sulfur injection in a chemistry-climate model"?

-> New title: "Reconstructing volcanic radiative forcing since 1990, using a comprehensive emission inventory and spatially resolved sulfur injections from satellite data in a chemistry-climate model"

Your 3D plume are not just vertically resolved?

-> Response: No, we are working with 3-dimensional plumes.

Abstract: the long list of satellite instruments and their acronym is not needed in an abstract?
-> Response: Names and acronyms of satellites are removed.

I find that the abstract does not highlight enough the novel and extensive character of the SO2 emission inventory nor the 3D plume injection method.
-> Response: Parts of the abstract are rewritten to highlight the novel character of the paper:
"This paper presents model simulations of stratospheric aerosols with a focus on explosive volcanic eruptions. Using various (occultation and limb-based) satellite instruments, with vertical profiles of sulfur dioxide (SO2) and vertical profiles of aerosol extinction, we characterized the influence of volcanic aerosols for the period between 1990 and 2019.
We established an improved and extended volcanic sulfur emission inventory that includes more than 500 explosive volcanic eruptions reaching the upper troposphere and the stratosphere. Each perturbation identified was derived directly from the satellite data and incorporated as a three-dimensional SO2 plume into a chemistry-climate model. The simultaneous measurements of SO2 and aerosol extinction by up to four instruments enabled us to develop a reliable method to convert extinction measurements into injected SO2. In the chemistry climate model, the SO2 from each individual plume is converted into aerosol particles and their optical properties are determined. Furthermore, the Aerosol Optical Depth (AOD) and the instantaneous climate radiative forcing are calculated online. Combined with model improvements, the simulations are consistent with the observations of the various satellites.
Slight deviations between the observations and model simulations were found for the large volcanic eruption of Pinatubo in 1991 and cases where simultaneous satellite observations were not unique or too sparse. Weak- and medium-strength volcanic eruptions captured in satellite data and the Smithsonian database typically inject about 10 kt to 50 kt SO2 directly into the upper troposphere/lower stratosphere (UTLS) region or transport it indirectly via convection and advection. Our results confirm that these relatively smaller eruptions, which occur quite frequently, can nevertheless contribute to the stratospheric aerosol layer and are relevant for the Earth's radiation budget. These eruptions cause a global radiative forcing of the order of −0.1 Wm$^{-2}$ at the tropopause (compared to a background aerosol forcing of about −0.04 Wm$^{-2}$)."

Abstract, lines 17-20: you say that your results "show" and that eruption "are found to"; I would instead say that your "confirm" these results as this has been shown by Schmidt et al. (2018)?
-> Corrected: "confirm"

Introduction: Also see major comment 1: the two main novelties of your study are overall not motivated in your intro (i.e. new injection strategy and improved SO2 inventory).
-> Response: These points are now more strongly highlighted in the introduction:
"For the ENVISAT (European Environmental Satellite) period 2002-2012 a first version of a new volcanic SO2 inventory with improved temporal and spatial resolution was developed within the framework of ISA-MIP (Timmreck et al., 2018; Brühl et al., 2018). The corresponding data base (https://doi.org/10.1594/WDCC/SSIRC_1) contains 3D-SO2-perturbations derived from satellite data as well as integrated injected SO2 masses. In this work the data base is expanded to the period 1990-2019 and considerably improved for the period 1998-2001. The simultaneous measurements from up to four instruments from 2002 to 2011 enabled us to develop a novel procedure for conversion of aerosol extinction to SO2 needed for the period before and after ENVISAT. Our method circumvents problems and uncertainties related to the classical point source approach like dependence on the box size and location, the time interval during which the mass is injected, and effects of microphysical and chemical interactions of SO2 and sulfate with injected volcanic ash and water in the early phase (Zhu et al., 2020)."

Introduction: I think the work of Mills et al (2016) and Schmidt et al 2018 (not cited) need to be discussed more given strong similarities with your study. Also you don't mention ISAMIP at all (Timmreck et al 2018) whereas your simulations are obviously relevant to this MIP?

-> Added citations: Timmreck et al. 2018: "For the ENVISAT (European Environmental Satellite) period 2002-2012 a first version of a new volcanic SO2 inventory with improved temporal and spatial resolution was developed within the framework of ISA-MIP (Timmreck et al., 2018; Brühl et al., 2018)."

Line 36: is it important to specify at which level it affects Earth radiative balance? If so also mention surface level in addition to TOA and tropopause.
-> Response: The forcing at that the surface cannot be retrieved by satellites, i.e. this would show model only.

Line 38: Multiple papers discuss how climate-volcano feedback could modulate future volcanic forcing though, and it may be a good place to mention it? See e.g. Swindles et al. (2017) (deglaciation effect on eruption frequency), Fasullo et al. (2018) (modulation of volcanic influence on surface temperature by changes in ocean stratification), Aubry et al. (2021) (impact of climate change on the volcanic stratospheric sulfate aerosol cycle).
-> Suggested references added.

Line 40: unless I misunderstand I guess you are talking about (mostly CMIP5) simulations that did not account for this forcing? Many model studies have accounted for this forcing since then, including CMIP6 historical simulations that use GloSSAC or e.g. Mills et al. (2016) and Schmidt et al. (2018)?
->These references and an additional one and more text added, see also reply to major comments.

Line 43/44: please add references
-> Response: See above.

Line 46: do you mean "overlooked" instead of "underestimated"? If not what was underestimated? Their radiative forcing? But does it not contradict the previous sentence?
-> Response: Text rewritten here, now related to GloSSAC and ISA-MIP.

Line 50: The SO2 emission and time-averaged volcanic forcing of degassing volcanoes and small eruptions is one order of magnitude larger than that of eruptions associated with stratospheric SO2 injections (e.g. Schmidt et al. 2012, Carn et al 2016). So clarify what you mean by "smaller natural source of aerosols" as this seems wrong as written.
-> Corrected: "a smaller" natural source -> "another" natural source

Section 2: could you group satellite instruments in terms of those used to constrain SO2 inputs in your model vs those used to evaluate the output of the model simulations? This would add a lot of clarity to this section. Also why not using GloSSAC (Thomason et al. 2018, 2020)?
-> Response: Data from all four satellite instruments described in this section are used as input to the simulations. MIPAS directly provides SO2 mixing rations, while for GOMOS, OSIRIS and SAGE II the SO2 input data are derived from aerosol extinction (for details see Sec. 4 and Appendix B). As explained in the introduction GloSSAC does not provide what we need.

Line 119-120: as said in my major comment I think you need to discuss the strength and limitations of choosing such a time window, and in particular how it compares to the SO2 e-folding time and the fact that choosing this time window may result in neglecting a large portion of SO2 converted to aerosols (even though I understand the argument that an earlier time window could account for SO2 estimates accounting for SO2 that will be rapidly scavenged by co-injected ash or hydrometeors; but this all needs to be discussed carefully). Sensitivity tests for this time window and understanding its impact on your SO2 estimates would be welcome.
-> Response: The satellite data (for MIPAS; GOMOS and OSIRIS) are provided in 5-day time intervals. These processes are important if point sources are used since the occur in the very first days. Use of a period of about at least 10 days of satellite date circumvents this, see introduction and Appendix B. In our institute studies on the early phase processes are in progress, also we included 2 references on this (Zhu et al. 2020 and Clyne et al. 2021).

Line 137: again this time window needs to be justified better. Also I'm not at all a remote sensing expert but I think it's the first time I see SO2 estimated from extinction coefficients in visible wavelength? Is that a standard method? How is the effect of SO2 on radiation properties isolated from other species, in particular sulfate aerosols? It may be standard techniques that I'm not aware of about but it would be good to clarify.

-> Response: The time lag is needed to allow for production of particles from oxidation of the injected SO2. Our method is a novel alternative also addressed in the introduction now. The part you refer to has been moved to section 4 and includes more details.

Line 139: My understanding here is that you are saying that if there is a data gap during the peak perturbation, you scale up by an arbitrary factor to recover a reasonable peak value? How is that factor chosen? There is absolutely no explanation nor reference and it may deserve dedicated SI plots?

-> Response: To estimate the factor, we iterated calculated extinctions to agree with OSIRIS and also used observations and assumptions by Vernier et al. (2016). A detailed description with 3 case studies is added to the Appendix:

"If the time lag of data is several weeks a correction factor >1 has to be applied to account for removal processes, if another event occurs relatively close in time, the factor has to be <1 to remove the influence of the previous event." See section 4 now.

Line 139 and 170: about data gaps and how to treat them, I'm just wondering why not using GloSSAC where the same problem had to be addressed and which is the reference dataset for the community? I understand you can't use it for SO2 but surely for aerosol properties it would make sense? The fact that major initiatives such as GloSSAC or ISAMIP are not mentioned is a bit surprising.

-> Response: GloSSAC does not solve problems with data gaps in some cases. If there is no other data set available it just smears out the gap (see also remark in introduction). However, for Pinatubo, SAGE data could be improved by CLAES data. In ISAMIP also an earlier version of our 5-day-dataset (link in Brühl et al. 2018 or Timmreck et al. 2018) is included.

Line 236: no apostrophe needed for Global Volcanism Program
-> Corrected.

Line 241: The tropopause altitude varies between ca. 8-9 and 16-17km depending on latitude and season, why not using the model diagnostic tropopause instead of the three thresholds used? Justify rigorously why you consider a threshold way below the tropopause height in the tropics but potentially way above at high latitudes. Also why do you need to mask tropospheric SO2? Would your model not account for the fact that tropospheric aerosol would have minimal impact on climate? I get that you don't want an overlap between the tropospheric and stratospheric volcanic SO2 inventory, but does the tropospheric SO2 inventory really account for emissions as high as 12-14 km or is it only passively degassing volcanoes?

-> Response: The EMAC simulations include comprehensive tropospheric chemistry and various inventories of tropospheric sulfur emissions (SO2 from outgassing volcanoes, anthropogenic emissions, etc., as well as other sulfur-containing species such as DMS from ocean fluxes). Therefore, double counting of tropospheric volcanic emissions should be avoided.

The lower limit of 12 km altitude at high latitudes was chosen based on the signal-to-noise ratio, uncertainties for low altitudes, and clouds in the volume mixing ratio profiles obtained by MIPAS and the other used satellite instruments. In the tropics, we set the lower limit at 14 km to account for transport processes in the UTLS layer, especially during the Asian summer monsoon. Here, we also have to consider high reaching clouds.

Lines 243-247: see my major comment #1
-> See response to major comment #1.

Table 2: this table really must be made available as a csv file or something that researchers can download and read in scientific programming software. Remove the table from the body of text as it is way too big.

-> Response: As an essential part of the novelty of this paper, the table should remain in the text because it is a comprehensive reference that cannot be represented by a single visualization. Additionally, the table is available for data processing at this link: https://doi.org/10.26050/WDCC/SSIRC_3

Lines 257-264: see my major comment #1. While I think this is at the moment poorly explained and that you have to show analyses demonstrating the advantages and challenges with this injection method, I do think that it is one of the most novel and important aspect of the paper (combined with your inventory) and that it should be highlighted and motivated a lot more.
-> Response: We added a detailed explanation with case studies in Appendix B (see reply to referee 1).

Line 275-276: you either need a reference backing this claim or data analysis to support it (e.g. does the GVP database have a comparable number/frequency of VEI 3-5 eruptions during 1991-2002 relative to 2002-present day? Or was it really a more quiescent period?
-> Response: The VEI index was developed for the volcanic explosivity, but is not a direct indicator for the climate relevance of volcanic eruptions, e.g. the eruption of Eyjafjallajökull in 2010 VEI4 had almost no influence on the stratosphere.

Indeed, there was a relative volcanically quiescent period from 1999–2002 (Schmidt et al. 2018), there are also less entries in the Smithsonian database, but this does not explain the low number of detected volcanic eruptions in the years directly after Pinatubo (1992-1998).

Lines 293-294: a brief comparison with observations in Carn et al. 2016 would be welcome here (I think they suggest even lower UTLS e-folding time). Also you say yourself here that the conversion time is about 2 weeks, which seems to strongly undermine your chosen 8-17 day time window to constrain SO2 emission from satellites?
-> Inserted:
" The conversion of SO2 to sulfate aerosol particles depends on several factors, such as the altitude, latitude, or season of the eruption and takes about 13, 23 and 32 days in 10–14, 14–18 and 18–22 km altitude (Höpfner et al 2015, midlatitudes), while Carn et al. (2016) report an e-folding time varying between 2-40 days. The range agrees with our simulations (and assumptions in section 4). "

Line 303-304: please clarify what you mean by "feedback to atmospheric dynamics" and cite appropriate references
-> Response: This means that radiative heating implies an enhancement of upward motion (or cooling a descent).

Line 309-310: the reader has to look at three different figures and compare them to verify this statement. It would me much better if you could present equivalent observations and model plots on the same figure and different panels. This would greatly facilitate modelobservation comparisons.
-> Response: In figure 8 we added an additional panel with observations from SAGE II and OSIRIS for the direct comparisons with the model simulations for aerosol extinction at 750 nm wavelength:
"Figure 8. Comparison of aerosol extinction for 750 nm wavelength at 17 km altitude between the model simulations (lower panels) and SAGE II and OSIRIS satellite data (upper panel). EMAC simulation of the stratospheric aerosol extinction are given on a logarithmic scale log(1/km) for 750 nm wavelength from January 1991–August 2019 based on the volcanic sulfur emission inventory (Table 2), in horizontal T63 resolution of zonal mean at 17 km altitude (middle) and in vertical distribution for tropical regions20° S–20° N (bottom). Maximum and minimum values appear above (dark red) and below (violet) the color keys, respectively."

Line 326: the vast majority of studies use SAOD at 550nm like you (e.g. Schmidt et al. 2018), and also 1020nm (e.g. Aubry et al. 2021) which is another standard one for some instruments? So this statement seem really not justified and should be removed or modulated.
-> Modified.

Line 331: clarify that the AOD of 0.4 is in the tropics and isn't a global mean value -> done: "with a stratospheric AOD of 0.4 in the tropics"
-> Corrected: "with a stratospheric AOD of 0.4 in the tropics"

Line 334: There could be other factors explaining model-observation differences in the post-Pinatubo period including flaws in the model (as evident from the different decay timescales) and uncertainty in the SO2 mass, or at least the "climatically relevant" portion of it (you use 17Tg, other studies use as little as 10 which should be briefly discussed; see Zhu et al. 2021, Mills et al. 2016, Schmidt et al. 2018).
-> Response: In this study about 17 Tg SO2 are injected for the Pinatubo eruption. Model comparisons by Timmreck et al. (2018) show that the range of used injections varies between 10 Tg SO2 (e. g. Mills et al. 2016, Schmidt et al. 2018) and 20 Tg SO2 (e.g. English et al. 2013). Thus, this study is in the middle range of injected sulfur mass.

Line 337: unless major eruptions are missing, is it really likely that imperfections in your inventory explain the large SAOD differences over 1993-1996?
-> Response: Between 1993 and 1996 the reduction of the stratospheric AOD in the model simulations is faster than indicated by the satellite observations and in Schmidt et al. (2018). This indicates that the removal of stratospheric aerosol is still too rapid from applying the modal model. Schmidt et al. (2018) show a slower decrease in AOD after the Pinatubo eruption. This could indicate that EMAC still needs better fine-tuning of the size distribution modes, or adding modes in the aerosol submodel to improve the aerosol removal in the stratosphere. Here the sectional aerosol model used by Schmidt et al. (2018) might have an advantage. Additionally, smaller volcanic eruptions might be missing, in view of the low number of identified events in the years after the Pinatubo eruption.

Figure 11: it may be better to show horizontal bars (with a length of 1 year) instead of green crosses as these are time-average measurement and it would facilitate comparison with your high-resolution output?
-> Done: Green crosses are replaced by bars with a length of 1 year.

Figure 11: Here and on Figure 9 and 10, could you not show for comparison the simulations from at least Bruhl et al. (2015) and maybe Schmidt et al. 2018 assuming their data are available with the paper? Discussing the differences would really improve the discussion.
-> Response: Pink lines with the results from Brühl et al. (2015/2018) are added to Fig. 9+11. Schmidt et al. (2018) is available for global AOD at 550 nm and added as black line in Fig. 9, AOD at 750 nm (Fig. 10) is not shown by Schmidt et al. (2018). In Fig. 11 the data for volcanic effective radiative forcing from Schmidt et al. (2018) is added as a black line.

Legend of Figure 11: specify the time resolution of the ERBE data. Is there no other observational estimate of radiative forcing to complement observations shown? E.g. CERES data?
-> Response: 72-day means are used in the near-global data set of ERBE (Toohey et al. 2011 fig. 2). In the AOD-figure we included AVHRR.
-> Question: CERES?  Can you provide a reference?

Line 354: "previous studies"-> show their data and discuss comparison? On that note making sure that your key outputs (SAOD/radiative forcing time series) are easily available is important and I don't think it's the case yet? Key outputs should not be made "available upon request" but should ideally be provided as SI or in a data repository.
-> Response: The output for radiative forcing is now available at WDCC:
(https://doi.org/10.26050/WDCC/SSIRC_3).

Line 359: For reference, can you indicate the SO2 mass for Merapi used in your and other (e.g. Carn et al. 2016) inventories? Overall, it would be really useful to have a comparison of your inventories with other standard ones, in particular those used in ISA-MIP (Timmreck et al. 2018). Another potentially useful reference, showing how different inventories affect the SAOD prediction by a simple model, is Aubry et al. (2020) (see Figure 8 there).

-> Response: See also Höpfner et al. (2015), Carn does not provide the stratospheric fraction. A range from our data sources is in Appendix B.

Figure 12: could you discuss how these results compare with recent studies, e.g. Rieger et al. (2020) or Stocker et al. (2019)
-> Response: Comparing our results with Rieger et al. (2020) shows that our results:
"...corresponds quite well with the results of Rieger et al. (2020) showing a maximum of instantaneous solar heating rate of 0.5 K/day in the tropics near 24 km plus thermal heating rates of about 0.2–0.3 K/day. "

Section 7: Overall I find that some of the most natural lines of discussion (and accompanying analyses) are completely missing including:

- i)        comparison of your new inventory with other ones, including Carn et al.;
  -> Response: A table for comparison of annual global volcanic SO2 emissions between this study and Carn et al. (2016) is added in Appendix C.
- ii)      comparison of your new simulations with other equivalent ones, including Schmidt et al (2018) and Brühl et al (2015);
  -> Response: A comparison with Schmidt et al. (2018) and Brühl et al. (2015) is added in Fig. 9 +11 and discussed in the text.
- iii)     discussion of how your 3D-plume injection strategy compares to a point injection.
  -> Response: With our model this cannot be done since a proper module for point sources is not available yet. Possible problems of the point source approach are addressed in the introduction now.

Line 385: provide numbers (e.g. latitude resolution at equator) that make it easier for the reader to understand the difference between these resolutions.
-> Added: T42L90 (2.81°×2.81°) to T63L90 (1.87°×1.87°). This you find now only in section 3.

Line 410: Missing reference?
-> Response: This is novel and expanded. Sentence corrected.

Line 429-430: Zhu et al. (2020) should be cited here
-> Cited: Zhu et al. (2020)

Line 429-432: On model difference/setup and how it may affect simulated aerosol properties, Clyne et al. (2021) is an important difference and should be discussed here and elsewhere.
-> Response: Clyne et al. (2021) is cited now in section 6.2 in connection with the sectional model of Schmidt since they show similar differences between modal and sectional models concerning the behavior after a major eruption (Fig .9). The major eruptions are not the main scope of our paper.

Lines 448-455: This whole paragraph doesn't acknowledge the contribution of previous studies when most of the statements made are not really new. First maybe you should refer to the AR6 report now that it is out instead of the AR5? Second, for radiative forcing estimate, the contribution of Schmidt et al. (2018) should be acknowledged and you should compare in details your forcing estimates to theirs. Third, for temperature effects, you should cite the papers by Santer and co-authors (2014, 2015) and Schmidt et al. (2018). I personally think that the novel aspects of your paper would be highlighted better if you ended it on key points related to the new inventory and the 3D plume injection method.
-> Response: Parts added in abstract, introduction and conclusions. Surface temperature is not the focus of this study since we nudge the troposphere and prescribe a time dependent SST. Comparison with Schmidt is done in Section 6.2 and 6.3.

**Added References:**

Aubry, Thomas J., et al. "Climate change modulates the stratospheric volcanic sulfate aerosol lifecycle and radiative forcing from tropical eruptions." Nature Communications 12.1 (2021): 1-16.

Brühl, C., et al. "Stratospheric sulfur and its implications for radiative forcing simulated by the chemistry climate model EMAC." Journal of Geophysical Research: Atmospheres 120.5 (2015): 2103-2118.

Carn, S. A., Lieven Clarisse, and Alfred J. Prata. "Multi-decadal satellite measurements of global volcanic degassing." Journal of Volcanology and Geothermal Research 311 (2016): 99-134.

Clyne, Margot, et al. "Model physics and chemistry causing intermodel disagreement within the VolMIP-Tambora Interactive Stratospheric Aerosol ensemble." Atmospheric Chemistry and Physics 21.5 (2021): 3317-3343.

Fasullo, J. T., et al. "The amplifying influence of increased ocean stratification on a future year without a summer." Nature communications 8.1 (2017): 1-10.

Kovilakam, Mahesh, et al. "The Global Space-based Stratospheric Aerosol Climatology (version 2.0): 1979–2018." Earth System Science Data 12.4 (2020): 2607-2634.

Rieger, Landon A., et al. "Quantifying CanESM5 and EAMv1 sensitivities to Mt. Pinatubo volcanic forcing for the CMIP6 historical experiment." Geoscientific Model Development 13.10 (2020): 4831-4843.

Schmidt, Anja, et al. "Volcanic radiative forcing from 1979 to 2015." Journal of Geophysical Research: Atmospheres 123.22 (2018): 12491-12508.

Swindles, Graeme T., et al. "Climatic control on Icelandic volcanic activity during the mid-Holocene." Geology 46.1 (2018): 47-50.

Timmreck, Claudia, et al. "The interactive stratospheric aerosol model intercomparison project (ISA-MIP): Motivation and experimental design." Geoscientific Model Development 11.7 (2018): 2581-2608.

Zhu, Yunqian, et al. "Persisting volcanic ash particles impact stratospheric SO 2 lifetime and aerosol optical properties." Nature communications 11.1 (2020): 1-11.

---

## Author Comment (AC4)

We thank the reviewer for making very useful suggestions to improve the paper. Our point-by-point responses to the reviewers' comments and corresponding changes are detailed below in blue text, and the changes are shown in the version of the manuscript with track changes.

**Comment on acp-2021-654**

Anonymous Referee #3

Referee comment on "Radiative forcing by volcanic eruptions since 1990, calculated with a chemistry-climate model and a new emission inventory based on vertically resolved satellite measurements" by Jennifer Schallock et al., Atmos. Chem. Phys. Discuss., https://doi.org/10.5194/acp-2021-654-RC3, 2021

Review of "Radiative forcing by volcanic eruptions since 1990, calculated with a chemistry-climate model and a new emission inventory based on vertically resolved satellite measurements by Jennifer Schallock et al.

Using various (occultation and limb based) satellite instruments, with vertical SO2 profiles from different satellite instruments and chemistry climate simulations, this study characterizes the influence of stratospheric volcanic aerosols for the period between 1990 and 2019. The results show that small but relatively frequently eruptions contribute to the stratospheric aerosol layer and could cause a global radiative forcing in the order of −0.1 Wm−2 at the tropopause. In specific, the objective of this study was to generate a detailed volcanic sulfur emission inventory, to improve the EMAC model simulations of the global stratospheric aerosol and sulfate burden, and to compute the volcano-induced radiative forcing through validation with satellite data.

Honestly, the paper keeps me a bit loss, as I am not sure if it is a more scientifically or more technically oriented paper. The scientific objective is not clear to me in particular the added value to the recent literature. I am wondering if the paper would not better fit in Earth System Science data (ESSD, https://www.earth-system-science-data.net/) or in Geoscientific Model Development (GMD, https://www.geoscientific-model-development.net/). The topic of the paper is in general very suitable for ACP but the paper needs major substantial revisions before publishing in ACP, see my major comments below.

**Major comments:**

The introduction needs a complete rewriting, less text book more scientific background with respect to the questions to be addressed. The paper is a successor of Brühl et al. (2015; 2018) and Bingen et al. (2017) but I miss a clear separation and explanation about the added values of this paper compared to its predecessors. The better horizontal resolution has already been discussed in Brühl et al. (2018), so the new aspect, as far as I understood it, is the increased amount of volcanic eruptions and the extend time period by using new satellite data.

-> Response: Abstract and introduction are expended to address this. The resolution is now only mentioned in the model description section

I completely miss references to recent literature in the introduction with respect to radiative forcing estimates of recent eruptions. There are several publications e.g. Andersson et al., (2015); Friberg et al., (2018), Schmidt et al., (2018); Kloss et al;(2021) just to name a few which have addressed the radiative forcing of small to moderate volcanic eruptions in the recent years. These papers have to be cited and differences/added values to their work have to be addressed in the introduction.

-> Response: Most included in introduction now, see replies to other referees and "Friberg et al. (2018) included the whole time series of CALIOP data from 2006 to 2015 and derived stratospheric AOD using reanalysis data for the tropopause, but mentions only medium size eruptions explicitly. Radiative forcing is estimated there from multiplying AOD with -25, an approach which is valid only for purely scattering aerosol". Kloss et al. (2021) cited in sections 6.2 ("Our northern hemisphere results for AOD of about 0.025 for Raikoke (550 nm) agree within uncertainties with Kloss et al. (2021) who use different satellites and different modelling

approaches") and 6.3 ("The value for Raikoke/Ulawun is within the range discussed in Kloss et al. (2021)") now. This paper was not available when we wrote the first version, thank you.

The discussion needs also to be rewritten. As mentioned above the lack of references of recent literature is astonishing. The results of the study need to be discussed in the context of recent literature, e.g. what do we learn from this paper, what we didn't know before from previous studies.
-> Response: Done, see comments to other referees.

I am also wondering about the importance of the small eruptions for the global radiative forcing. It would be interesting if you neglect all small eruptions below a certain threshold values e.g. 10 kT SO2, how this would really change the global radiative forcing. What is range of uncertainties, the range of interannual variability in background periods? Estimates about the uncertainty range are completely missing in the paper.
-> Response: See comparison with results of Schmidt et al. (2018) in Fig 11. They neglect also larger eruptions than your threshold, however. See also remark to referee 1 on background.

Last but not least, differences between the model simulations and satellite measurement need not to be the only cause of missing SO2 sources. There could be several other reasons for the discrepancies (transport, microphysics), neither model simulations nor satellite measurements are perfect. This has to be discussed here as well.
-> Response: Mentioned at several places.

**Specific comments**

Abstract, line 17: "significantly" is a big word. I did not find any significance tests in the paper.
-> Removed: "significantly"

Page 3, which SSTs do you use? I suppose you run only one ensemble members did you check for the influence of internal variability at least in short sensitivity studies?
-> Response: The CCM is nudged to ERAI which includes SSTs (see section 3).

Description of the EMAC module could be reduced, to only the parts which are really relevant for the paper,. e.g. the calculation of the radiative forcing. This part could be more elaborated. More detailed model descriptions can be put in the appendix.
-> Response: Parts of the model description are moved to the appendix:
"As EMAC is a very complex chemistry climate model it contains many submodels and functions which are essential for running the simulations but are not directly related to the sulfur cycle, these are mentioned in Appendix A. In this section we focus on the sulfur cycle."

Page 12, lines 245-247 It would be nice to see a comparison with Carn et al (2017) and other recent emission data
-> Response: You compare then apples and oranges, a hint is given in Appendix C.

Table 2: It would be nice to see (e.g. with different color) which entries are new or changed with respect to the previous data set.
-> Response: Eruptions from in Bingen et al. (2017) and Brühl et al. (2018) are marked in italics in the table:
"Based on a previous study from Brühl et al. (2018) with scaling factors for T63 and already published in an earlier version in Bingen et al. (2017) (in italics)."

Will the data set be published?
-> Response: Yes, the data set is published on WDCC: https://doi.org/10.26050/WDCC/SSIRC_3.

Page 21, line 279 "strong" I wouldn't call Kasatochi or Raikoke a strong eruption
-> Corrected: "medium strong"

Figures 9, 10, 11: A comparison with Brühl et al. (2015) for the Pinatubo period and with Brühl et al (2018) for 2002 to 2012 would be nice, to better asses the improvements of this study. Also a validation with GloSSAC (Thomason et al., 2018; Kovilakam et al., 2020) would more than beneficial.
-> Response: Pink lines for comparison with Brühl et al. (2015) are added in fig. 9+11. GloSSAC gives no additional information here since it is derived from data shown in the figure but we can include it in the upper panel since it covers a longer time period than the blue line. Nevertheless, we include a black line in Fig 9 since it is interesting for Pinatubo and the period 2012 to 2018 (based on V2 of Kovilakam).

Section 6.3: Any reason why you look at the tropopause? What is the uncertainty range in your forcing estimates?
-> Response: This is because of the comparison with Solomon et al. (2011). For Pinatubo it differs not much from the value at TOA.

Figure 11 I recommend a comparison with Schmidt et al (2018) here
-> Response: Schmidt et al. (2018) is available for global AOD at 550 nm and added as black line in Fig. 9, AOD at 750 nm (Fig. 10) is not shown by Schmidt et al. (2018). In Fig. 11 the data for volcanic effective radiative forcing from Schmidt et al. (2018) is added as black line.

Page 409, 410: "This was demonstrated to be essential for correctly assessing the extinction coefficient in volcanically quiescent periods." By whom? Maybe I have overseen it but I didn't find it in the paper.
-> Reformulated: "This was demonstrated to be important for correctly modelling the AOD in volcanically quiescent periods". Convection was mentioned in earlier sections.

Page 445, Which studies?
-> Check given reference and Brühl et al. (2018).

**Added References:**

Andersson, S. M., Martinsson, B. G., Vernier, J. P., Friberg, J., Brenninkmeijer, C. A. M., Hermann, M., Van Velthoven, P. F. J., and Zahn, A.: Significant radiative impact of volcanic aerosol in the lowermost stratosphere, Nat. Commun., 6, 1–8,https://doi.org/10.1038/ncomms8692, 2015.

Bingen, C., Robert, C. E., Stebel, K., Brühl, C., Schallock, J., Vanhellemont, F., Mateshvili, N., Höpfner, M., Trickl, T., Barnes, J. E., Jumelet, J., Vernier, J.-P., Popp, T., de Leeuw,G., and Pinnock, S.: Stratospheric aerosol data records for the climate change initiative: Development, validation and application to chemistry-climate modelling, Remote Sensingof Environment, 203, 296–321,https://doi.org/10.1016/j.rse.2017.06.002, 2017.

Brühl, C., Lelieveld, J., Tost, H., Höpfner, M., and Glatthor, N.: Stratospheric sulphur and its implications for radiative forcing simulated by the chemistry climate model EMAC, J. Geophys. Res.-Atmos. 120, 2103–2118, https://doi.org/10.1002/2014JD022430, 2015.

Brühl, C., Schallock, J., Klingmüller, K., Robert, C., Bingen, C., Clarisse, L., Heckel, A., and North, P.: Stratospheric aerosol radiative forcing simulated by the chemistry climate model EMAC using aerosol CCI satellite data, Atmospheric Chemistry and Physics, 18,1–15, https://doi.org/https://doi.org/10.5194/acp-18-12845-2018, 2018.

Friberg, J., Martinsson, B. G., Andersson, S. M., and Sandvik, O. S.: Volcanic impact on the climate – the stratospheric aerosol load in the period 2006–2015, Atmos. Chem. Phys., 18, 11149–11169, https://doi.org/10.5194/acp-18-11149-2018, 2018

Kloss, C., Berthet, G., Sellitto, P., Ploeger, F., Taha, G., Tidiga, M., Eremenko, M., Bossolasco, A., Jégou, F., Renard, J.-B., and Legras, B.: Stratospheric aerosol layer perturbation caused by the 2019 Raikoke and Ulawun eruptions and their radiative forcing, Atmos. Chem. Phys., 21, 535–560, https://doi.org/10.5194/acp-21-535-2021, 2021.

Kovilakam, M., Thomason, L. W., Ernest, N., Rieger, L., Bourassa, A., and Millán, L.: The Global Space-based Stratospheric Aerosol Climatology (version 2.0): 1979–2018, Earth Syst. Sci. Data, 12, 2607–2634, https://doi.org/10.5194/essd-12-2607-2020, 2020.

Schmidt, A., Mills, M. J., Ghan, S., Gregory, J. M., Allan, R. P., Andrews, T., Bardeen, C.G., Conley, A., Forster, P. M., Gettelman, A., Portmann, R. W., Solomon, S., and Toon, O.B.: Volcanic Radiative Forcing From 1979 to 2015, J. Geophys. Res.-Atmos.,123,12491–12508, https://doi.org/10.1029/2018jd028776, 2018.

Thomason, L. W., Ernest, N., Millán, L., Rieger, L., Bourassa, A., Vernier, J.-P., Manney, G., Luo, B., Arfeuille, F., and Peter, T.: A global space-based stratospheric aerosol climatology: 1979–2016, Earth Syst. Sci. Data, 10, 469–492,https://doi.org/10.5194/essd-10-469-2018, 2018.

---

## Author Response (AR2)

We thank the reviewers again for making very useful suggestions to further improve the paper. Our point-by-point responses to the reviewers' comments and corresponding changes are detailed below in blue text, and the changes are shown in the version of the manuscript with track changes.

**Report #1**
Anonymous Referee #1

I appreciate the work the authors have put into improving the manuscript. The method of obtaining SO2 values from extinction observations is now less opaque than it originally was, and many smaller details have been improved. I do however still have some concerns with the manuscript. I will here revisit the major comments from my first review, then move into specific comments on the revised manuscript.

**General comments:**
1. On the method of estimating SO2 injection from satellite aerosol extinction observations, I do appreciate the new material added, which help to understand the method. But still many important details are missing. I point out that according to the ACP review criteria, my job is to ask "Is the description of experiments and calculations sufficiently complete and precise to allow their reproduction by fellow scientists (traceability of results)?" Presently, I do not think this is the case. Here are some unresolved questions I have about the method:

• The method uses data typically from "a ten-day period beginning about a week after the eruption". However, the given e-folding lifetime of SO2 is between 13 and 32 days according to Hoepfner et al. This suggests that at the period from which the extinction measurements are used, not all SO2 has been converted to sulfate aerosol, by an amount that depends on altitude. Is it assumed that all SO2 has been converted, and if so, can you quantify the impact this assumption would have on your results?
-> Response: In section 4 is added "... in the tropics. For higher latitudes the selected period is later and longer, taking into account the longer conversion time (due to less OH)." More on that in Appendix B and the electronic supplement.
• The effect described above could be exacerbated by the fact that satellite instruments may be insensitive to aerosol particles of very small size. Some related discussion would be useful.
-> Response: We always check the total amount of injected SO2 derived by our method against estimates from nadir instruments for consistency as stated in the text. Volcanic aerosol and its extinction is always dominated by the accumulation mode with a small fraction from the Aitken mode (and bigger particles for Pinatubo).
Satellite measurements are not insensitive to aerosol extinction by very small particles, but it is true that the retrieval of aerosol extinction by very small particles is challenging because such particles result in Rayleigh scattering that has to be distinguished from Rayleigh scattering by neutral air density. For this reason, the estimation of this contribution from neutral air has to be as accurate as possible. We added as short discussion on this issue, and more detail about the GOMOS retrieval in section 2.2.
• A key part of the method is the construction of a scaling factor from model results between extinction and sulfate mixing ratio. It is not obvious that a single number will be universally applicable to this purpose, rather one would expect the scaling factor would depend (perhaps strongly) on the aerosol size distribution. I think a detailed illustration of the extraction of this scaling factor from model results and a discussion of the uncertainty in this scaling factor is essential to the description of this novel method.
-> Response: This is explained in more detail now in the main text and Appendix B. The uncertainty depends on the individual eruptions and the data coverage but it should be less than the one from the idealized assumptions on the vertical distributions in the point source methods.
• At line 313 of the tracked changes document, can you explain why one needs to divide by air density? If the scaling factor allows one to covert from extinction to sulfate mixing ratio, its not clear to me why air density is needed to get to SO2 mixing ratio.
-> Response: We have modified the text concerning that to be more precise. The extinction is proportional to the concentration so to get a mixing ratio you have to divide by density as also shown in the formula in Appendix B.
• More detail on how the spatial structure of the extinction plume is quantified is needed. There are many references to the 3D structure of the plume, however, references to the zonal mean are also present (e.g., line 317), and so it's not quite clear if the quantities are based on full zonal averages or an integration over a limited spatial range. And if the latter, how that range is defined. It's also not clear to me if the method assumes that the aerosol plume is stationary in space. If the plume is moving, then over the 5 day period of measurements, similar parts of the plume may be measured in different locations. This would seem to be a significant source of uncertainty in the method.

-> Response: It is not our objective to model the details of the plume or local effects but the impact of the volcanoes on climate forcing. There is some smearing out of details due to the use of data averaged over a period of several days. The examples with point sources in Appendix C show that the forcing can be similar but that the uncertainties from the assumptions on the vertical distribution are larger. The text on spatial integration at the beginning of section 5 has been corrected (see also details).

• There is a final correction factor which takes values between 0.5 and 3 (so, close to an order of magnitude of impact on the final result) to correct for time gaps between SO2 injection and extinction measurement, or the influence of prior eruptions. The description of how this correction factor is derived and applied is inadequate. What data sets are used to construct this correction factor, and how exactly does it depend on time since eruption and any other factors (latitude, injection height, etc). I assume that the correction factor is constructed from cases where both extinction and MIPAS SO2 measurements are available, and then it is applied in the cases when MIPAS measurements are not available, but this is not explicitly stated anywhere. Please elaborate.

-> Response: We hope we have clarified the concept of the correction factor now in the text. As described in more detail in Appendix B the correction factor is always case dependent. If the time lag of data is several weeks a correction factor >1 has to be applied to account for removal processes, if another event is relatively close in time, the factor has to be <1 to remove the influence of the previous event. Factor 1 is default. For convenience, we supply now a table with the correction factors and the selected times for OSIRIS in the electronic supplement (referred to in section 4).

• I do not understand the advantage of using SAGE II SAD values rather than extinction observations for the analysis. SAD is a derived product, which comes from the extinction but incorporates a number of assumptions, and is therefore more uncertain than the extinction measurements. Plus, since the study uses extinction measurements from OSIRIS and GOMOS, it would be more consistent to use the extinction measurements from SAGE II. The justification given for using SAGE II SAD is that it "it is easier to correct for cloud contamination", but why exactly is this?

-> Response: We used both approaches as shown in Appendix B.
In the lower part SAD has a slightly better data coverage than the extinction at 525 nm. Another reason for using SAD here is for comparison of results with our earlier studies.

2. Concerning the table of results, the formatting is improved compared to the first version. My opinion is that a format with one event per line would be much easier for users to use (it could be read into a spreadsheet program for instance), the FORTRAN formatted table would be difficult for some users to parse. But I leave this to the discretion of the authors and editor.

-> Response: Comprehensive models normally use netcdf or ascii data as input as provided in the link. Preferably the 3D-data should be used, the table is more for comparison purposes as indicated at the beginning of section 5. A table with about 550 lines would be unacceptable for the readership. The caption of Table 2 is slightly modified to be more clear what is available in the link.

3. Finally, concerning the agreement between the model output and observations, I appreciate the work the authors have done to adjust the colorbars of the various figure to make them consistent. This makes quantitative comparison possible for the reader, if not especially easy. Comparing the tropical extinction profile timeseries between the model (Fig 7, 8) with the SAGE II (Fig 5), OSIRIS (Fig 4) and GOMOS (Fig 3) extinctions, there seems to be a noticeable difference in the vertical profile, with the model results showing larger extinctions generally between 18-20 km and lower extinction below 16 km. This difference seems to cancel out in the SAOD, but should not be ignored.

-> Response: When comparing the vertical distribution of aerosol extinction at 750 nm between the satellite observations below 16 km altitude of GOMOS, OSIRIS and SAGEII one recognizes the different sensitivities of the instruments on cloud perturbations and measurement uncertainties. As written in chapter 2:

OSIRIS: "At altitudes near and below the tropopause, the OSIRIS measurements are sensitive to clouds that may be interpreted as elevated aerosols. This is likely contributing to larger background extinction values measured below approximately 17 km in the tropics, as can be seen in Figure 4 (bottom), and the uncertainty is higher."

SAGE: "Red pixels around 14–16 km correspond to measurements contaminated by clouds, increasing the optical depth in the upper troposphere/lower stratosphere (UTLS) region on the lower panel of Figure 5. The perturbations by convective clouds occur mostly over the West Pacific and were excluded in the procedure for estimating the SO2 injections. "

This is also one reason, why we can't set the integration boundary easily to the tropopause level.

It is helpful to have the SAOD results from Bruehl et al (2015) on Fig. 9. However, for a large portion of the 2002-2012 period of overlap, the difference between the results of Bruehl et al. and the present study seem very small, and it is not clear if there is any significant difference between them. They might both be within the uncertainty spread of the observations. There needs to be more clear evidence to support the conclusion that the new data set is an improvement upon prior emission estimates.

-> Response: Figures 9, 10 and 11b are presented with a logarithmic scale on the y-axis, so differences could seem very small. The differences are up to about 40%. See attached a linear version of the figure without Pinatubo.

[Figure]

Zoom in of Figure 9a. Stratospheric AOD at 550 nm wavelength: Tropical regions 20° S–20° N above 110 hPa are shown. Satellite observations from GOMOS (Bingen et al., 2017) are indicated by the green line and values derived from SAGE+CALIPSO (Santer et al., 2014) by the blue line. The red line shows the EMAC model simulations using the SO2 injections of Table 2, compared to the simulations of Brühl et al. (2015) (pink dashed line). The black line is from GloSSAC.

[Figure]

Zoom in of Figure 11a. Global radiative forcing by stratospheric aerosol. Estimated averages for solar forcing at the top of the atmosphere from satellite observations of annual averages derived from observations by Solomon et al. (2011) as green bars. The EMAC model simulations with instantaneous forcing at the tropopause (185 hPa, solar + IR) based on volcanic SO2 emissions are represented by the red line, compared to the simulations of Brühl et al. (2015) (pink line) and data from Schmidt et al. (2018) with volcanic effective radiative forcing (black line).

**Specific comments:**
All line numbers in reference to tracked changes document:

L10: "Directly" seems unfounded. If SO2 injections are derived from extinction measurements days to weeks after the injection, and derived using a model-based conversion factor and an empirical correction factor, is this "direct"?
-> Corrected: Removed "directly"

L16: the results of a simulation might be consistent with observations, not the simulations themselves.
-> Corrected: "results of the" simulations

L30: this sounds like a description of an externally mixed aerosol, which is incorrect.
-> Corrected: "internal liquid mixture"

L50-52: references needed for these statements.
-> Added citations: Aquila et al. (2012), Toohey et al. (2011)

L59: reference needed
-> Inserted: "(e.g. Kasatochi compared to Glantz et al 2014)", presentations at SSIRC and EGU meetings.

L66: This description of GloSSAC is so negatively phrased as to be potentially insulting to the authors of that work. Those authors have put years of work into filling the gaps, ensuring the best possible smooth transition between instruments, and addressing myriad other difficulties in the construction of such a long-term data set. It may not be perfect, and one can surely point this out, but I would encourage a more measured tone. Also, GloSSAC is primarily a climatology of aerosol extinction, and does not include SO2 at all. Therefore it is incorrect to say that GloSSAC excludes MIPAS data, since MIPAS does not provide measurements of extinction or the other optical properties included in GloSSAC.

-> Thank you for noticing this. Rephrased: "GLOSSAC (Thomason2018, Kovilakam2020), a time dependent aerosol climatology sometimes used for climate simulations, has a coarse temporal resolution and sometimes large uncertainties due to data gaps. It does not provide SO2 needed as input for chemistry climate models directly but only extinction."

L70: This makes no sense. It's true GloSSAC doesn't include SO2, so you don't need to explain why you don't use GloSSAC for SO2 injections. You do need to justify your work in relation to the other SO2 data sets that are available, including the work of Carn et al. (2016).

-> Response: In Appendix C a comparison of annual volcanic SO2 emissions from Mills et al. 2016 is added to the comparison between Carn et al 2016 and this study. To include that in detail in a table or a figure we lack data (since in Carn et al only a subset is provided).

L81: how is the location of the eruption a problem: the volcano location is known in almost all cases, is it not?

-> Response: Since simulations of point source emissions are very sensitive to the emission conditions, in some cases it may be more appropriate to implement the main plume of volcanic emissions in the model not directly at the volcano location, and instead choose other coordinates according to satellite observations. A case study for point source emissions is shown in Appendix C2.

L84: the logic here isn't clear: if tropospheric aerosols are removed by oxidation and rainout, how can they contribute to stratospheric aerosol layer at all, background or otherwise?

-> Added: most, "but not all" of the released SO2 is removed... and only a small fraction can reach the stratosphere by convection or large scale transport.

L98: It would be clearer for the reader to be as explicit as possible about the difference between which instruments are measuring SO2 and which extinction.

-> Response: Added "SO2 data" and "aerosol extinction data" to the single satellite instruments for clarification.

L171: the white areas in Fig 4 extend in latitude from the pole down to 0-30deg in both the NH and SH every year. The text here still appears to not explain this.

-> Sentences rephrased: "OSIRIS provides coverage from 82°S-82°N over the course of the year. Extinction is retrieved where the tangent point is illuminated, which is primarily in the summer hemisphere (see Figure 4)."

L215: SAD is a derived product included in GloSSAC, not "used in GloSSAC".

-> Corrected: "included"

L278: I still think it's worth being explicit that the IRF is calculated based on a difference of the two cases.

-> Corrected: Sentence rephrased: "Via multiple calls of the RAD submodel in one simulation, the instantaneous forcing is calculated online from the difference of fluxes for the cases with stratospheric aerosol only above 100 hPa and without any aerosol (Brühl et al 2012), additionally to the call with full aerosol used for the interaction with dynamics."

L280: The reader won't know what RAD_FUBRAD is if the sentence prior is removed.

-> Moved to Appendix A.

L319: "Observations and assumptions by Vernier (2016)" needs to be explained in much more detail.

-> Response: "like the decay of extinction by sulfate with time over 4 months."

L330: Are these limb or nadir measurement data sets" If the former, how does that affect consistency between those data and your results? If the latter, is your technique applied in full?

-> Response: nadir instruments (text modified)

L359: "integrating the vertical profiles" is fine, but to produce a single number for the SO2, you also need to integrate over the horizontal extent of the plume. This needs to be explained in detail.

-> Corrected: by integrating the 3-dimensional SO2 perturbations "over the boxes related to the volcano"

L374: I believe most use the term "Junge layer" to refer to the whole stratospheric sulfate aerosol layer, from the tropopause up, not just 25-29 km.
->Sentence rephrased: "Figure 6 shows the modeled vertical distribution of stratospheric SO2 in the Junge-aerosol layer with the local maximum of SO2 around 25 to 30 km altitude (Höpfner et al., 2013), typical mixing ratios of SO2 are about 0.03 ppbv."

L383: This might be true, but do you know for absolute sure? You mention there are less entries in the Smithsonian in the 1990s. Wouldn't this suggest it is at least possible there have been changes in eruption frequency?
-> Corrected: "is" -> "might be"

L385: It is true that at 17km, SO2 mixing ratios are generally larger in the tropics, but this may not be the case for all altitudes.
-> Added: in the "lower" stratosphere... See also next sentences.

L390: The impact of the Asian monsoon on Nabro aerosols has been addressed first by studies other than Clarisse et al., 2014.
-> Added citations: "Fairlie et al., 2014 and Bourassa et al., 2012".

L393: Is the NASA SO2 database referred to here the same as that of Carn et al. (2016) which is referred to throughout the manuscript?
-> Added source: Online NASA SO2 database at: https://so2.gsfc.nasa.gov

L436: This might well be the case, but unless an experiment has been performed to test this, the statement should be softened (e.g., "...likely because...").
-> Corrected: "...likely because..."

L437: The previous paragraph introduced 3 figures, it's not clear which 2 are referred to here.
-> Corrected: "Figure 7 and Figure 8"

L448: Typically, stratospheric aerosol optical depth is calculated by integrating from the height of the tropopause up. Here, it is said that a lower bound is used that is 16 km in the tropics and 14 km in the midlatitudes. This is quite different definition, which needs to be justified. It is also quite unclear if this procedure is applied to both the model results and the observations.
-> Response: For practical reasons, the total stratospheric Aerosol Optical Depth (AOD) is obtained by the vertical integral of the aerosol extinction above an altitude of about 16 km (for mid-latitudes above about 14 km), since there is no information on the actual/variable tropopause altitude in the satellite observations to compare with.

L461: How do you know for sure? The GloSSAC data has been gap-filled. This may not be perfect, but it attempts to correct for the saturation. The difference between the model and GloSSAC then could be for another reason.
-> Response: "GLOSSAC tried to correct for that and has larger values." In the data for July to September 1991 you still find remnants of the gaps due to saturation.

L468: What data product from AVHRR is included?
-> Response: "...the AOD from AVHRR (Advanced Very High Resolution Radiometer) by Long and Stowe 1994 at 630 nm is included." The data were taken from a figure in that paper, there is no information on the data version.

L469: The AVHRR AOD product is at a different wavelength than the products shown on Fig 10. Any conclusion to be reached from the comparison may depend critically on the degree to which it would be "slightly less" or "slightly larger" when converted to the correct wavelength. Showing AVHRR, at the wrong wavelength on Fig 10, but not Fig 9 gives an impression of cherry-picking, showing data only when it appears to agree but not when it disagrees. If AVHRR is to be included, more effort should be put into making the comparison quantitative and fair.
-> Response: Text modified: "Consistent with the typical wavelength dependence, these values lie between the red curves for 550 nm (Fig. 9) and 750 nm (Fig. 10) at the peak after the Pinatubo eruption." The curve is left out for practical reasons in Fig. 9 since it would cover other curves (see attached version).

[Figure]

Stratospheric Aerosol Optical Depth 550 nm, 20° S–20° N

Figure 9a. Stratospheric AOD at 550 nm wavelength: Tropical regions 20° S–20° N above 110 hPa are shown. Satellite observations from SAGE II (Thomasonet al., 2008) are indicated by the light blue line, GOMOS (Bingen et al., 2017) by the green line and values derived from SAGE+CALIPSO (Santer et al., 2014) by the solid blue line. The red line shows the EMAC model simulations using the SO2 injections of Table 2, compared to the simulations of Brühl et al. (2015) (pink dashed line) and the black line in the upper panel is from GloSSAC. In this version we added the dashed blue line showing the AVHRR observations by Long and Stowe (1994) at 630 nm.

[Figure]

Stratospheric Aerosol Optical Depth 750 nm, 20° S–20° N

Figure 10a. Stratospheric AOD at 750 nm wavelength: Tropical regions 20° S–20° N above 110 hPa are shown. Satellite observations from OSIRIS (Rieger et al., 2019) are indicated by the blue line and GOMOS (Bingen et al., 2017) by the green line. For the Pinatubo period the blue dashed line shows the AVHRR observations by Long and Stowe (1994) at 630 nm. The light blue line shows the interpolation of SAGE data at 550 nm and 1025 nm wavelengths. The EMAC model simulations, using the SO2 injections of Table 2, are shown by the red line.

L471: this sentence seems to focus first on the Pinatubo eruption, but later on the impact of temporal resolution on the detection of individual eruptions. I doubt that temporal resolution has any impact on the magnitude of the peak AOD for Pinatubo. The argument needs to be clearer here.
-> Response: For better understanding, the sentence is split now. Of course, on the time scale of volcanic eruptions, which in some cases span only a few days, it makes a mathematical difference whether monthly means or multiple values with finer temporal resolution are considered; the same is held for zonal means. For a peak, the plotted maximum value is lower for a monthly average than for 5-day averages.

L503: "The large difference…" It's not clear, but I think you refer here to the difference between your current simulations and the simulations of Schmidt et al. If so, then yes, the fact that the Schmidt et al. simulations used a smaller number of eruptions is *one possible reason* for the differences in RF in the periods between eruptions. But there are other possible reasons, including difference amounts of background aerosol injection, since a larger baseline in the Schmidt et al results would reduce the difference substantially. On the other hand, the model simulation from Bruehl et al 2018 also used a smaller eruption list than the present study, and the difference in RF between that simulation and the one of the current study look negligible in Fig 11. This would suggest that those smaller eruptions have a negligible impact on the RF.

-> Added: The large difference "of the simulations by Schmidt et al 2018" to our simulation…

L530: I'd suggest "infrared" rather than "thermal"
-> Response: The terminus "thermal heating rate", as well as "solar heating rate" is cited from Rieger et al 2020 (fig. 4). One may use "terrestrial" but "infrared" includes also solar infrared what is not meant here.

L540: "compute … through validation" makes no sense
-> Sentence rephrased: "… compute the volcano-induced radiative forcing using computed extinctions validated with satellite data."

L547: L2 needs to be defined
-> Corrected: "Level 2"

L571: This was not really demonstrated.
-> Response: Gaps in SAOD are not observed and background from OCs or other non-volcanic sources cannot explain the observations.

Fig B1, B2, B3: Over what time period are these values compiled? Are the plots to the left showing full zonal averages, or averages over the defined zonal extent of the plumes?
-> Response:  Captions expanded:
"Figure B1. 2002 Reventador eruption: SO2 mixing ratio perturbation derived from MIPAS and the 3 extinction instruments. MIPAS data of 7 to 17 November 2002, OSIRIS and GOMOS data 11 to 22 November, SAGE II orbits of November and early December 2002. Zonal average …"
"Figure B2. 2010 Merapi eruption: SO2 mixing ratio perturbation derived from MIPAS, GOMOS OSIRIS and GOMOS with gap filling. MIPAS data 5 to 20 November2010, OSIRIS 14 to 25 November, GOMOS 25 November to 9 December (sparse data). Zonal average…"
"Figure B3. 2009 Sarychev and Mando Hararo eruptions: SO2 mixing ratio perturbation derived from MIPAS and OSIRIS., arrangement of panels because event is split at 24°N as for integrated values in Table 2. MIPAS data 18 June to 18 July 2009, OSIRIS data 17 July to 3 August."

Fig B3: These plots seem to clearly show a continuous single plume. It may be the combined impact of two eruptions, but to split the plume at 24N seems extremely arbitrary.
-> Response: The event has been split at 24°N as for integrated values in Tab.2.  (see above)

Table C1: How is explosive defined? Column "in %" needs to be better defined/described.
-> Response: Column "in %" is the fraction of explosive emissions from total emissions, now removed to get space for comparison to Mills et al. 2016. The term "explosive" SO2 emissions is used by Carn et al. 2016 [Table 3] for "explosive eruptions only (i.e., excluding effusive eruptions)". In the caption we have added contributions from outgassing (Diehl et al, 2012).

**Report #2**

Referee #2: Thomas Aubry, ta460@cam.ac.uk

First, I thank the authors for their efforts in addressing my comments and for significant improvements to the paper. However, one of my main suggestion has not been addressed. The ultimate goal of this paper is to take advantage of a range of observational capabilities to conduct simulations in which volcanic SO2 is injected as a spatially-resolved SO2 cloud instead of the traditionnal point source approach. I think this is an interesting and valuable idea, but I find that the paper is undermined by the fact that no equivalent simulations with the same model and with point source injections were run. This makes it difficult to understand and discuss the advantages and limitations of the methods proposed by the authors. This is the last major comment I have, and I otherwise find the manuscript acceptable for publication.

-> Response: In Appendix C2 we added a chapter with sensitivity simulations for comparison of different case studies for volcanic point source injections (about 5 pages including figures). This is now also mentioned in the abstract, introduction, section 5 and the conclusions. Appendix A includes an additional module for that.

---

## Author Response (AR3)

General remarks:

We thank the reviewers again for making very useful suggestions to further improve the paper. Our point-by-point responses to the reviewers' comments and corresponding changes are detailed below in blue text, and the changes are shown in the version of the manuscript with track changes.

We thank the editor A. Schmidt and the reviewer for pointing to some misunderstandings concerning the different treatment of background stratospheric aerosol and small eruptions in EMAC and WACCM. We have corrected the text in the introduction, section 6.3 and conclusions accordingly.

Figure 11 had to be redesigned. We show now mostly the forcing at the top of the atmosphere because that is the quantity used by IPCC and because the data for calculating the forcing at the tropopause from WACCM were not available. The corresponding arrays for radiation fluxes contained only zero values.

[Figure]

Figure 11. EMAC instantaneous radiative forcing by stratospheric aerosol (red, pink and blue lines, 5-day averages). Solar forcing at the top of the atmosphere (TOA) (dashed red line) is compared to solar forcing at the TOA from satellite observations of the Earth Radiation Budget Experiment (ERBE) (72-day means) (Wong et al., 2006; Toohey et al., 2011) (light blue crosses). The full red line displays the total (solar+IR) forcing at TOA while the blue dashed line shows the same at the tropopause. The dashed pink lines show total forcing at TOA of Brühl et al. (2015). Green bars show annual averages derived from observations by Solomon et al. (2011). The black line shows results from Schmidt et al. (2018) with volcanic radiative forcing at TOA including a background aerosol forcing of -0.05 Wm$^{-2}$. The lower panel is a zoom of the upper panel.

GloSSAC is included now in most frames of Figures 9 and 10 using provided extinction. An additional figure using GloSSAC as counterpart to our Figure 8 is provided in the supplement (appendix C1).

We have added also remarks on possible underestimates in radiative forcing due our calculation method in EMAC, which neglects the radiative effects of aerosol between the

tropopause and 100 hPa, in Section 6.3, conclusions and appendix C3. The discussion during the review process will lead to model improvements.
We checked the whole manuscript and corrected or removed misleading or distracting words and sentences.

**Report #1**
Anonymous Referee #1

**Suggestions for revision or reasons for rejection (will be published if the paper is accepted for final publication)**

All line numbers refer to tracked changes version of manuscript.

L52: "Radiative forcing is…" I don't believe it is true that this approach is "valid only for purely scattering aerosol". The -25 factor comes from Hansen et al. (2005), from model simulations including prescribed stratospheric aerosol for Pinatubo, and is based on the TOA net (SW+LW) flux anomalies, thus incorporating both scattering and absorption effects.

-> Response: Text modified: "…valid only for scattering sulfate aerosol (see e.g. Sellitto et al. (2022))." There are several other references on that, nevertheless, the factor is still good for plausibility studies.

L56: The description of GloSSAC is somewhat improved compared to prior versions. But still, some evidence (via citation) should be included to support the claim that GloSSAC includes "large uncertainties due to data gaps"? I think here you are really referring to the issues with "saturation" after Pinatubo? If so, it would be helpful to be specific about it and make reference to statements in the relevant literature, e.g., from the GloSSAC papers.

-> Response: Rephrased to a more neutral formulation without the remark on data gaps: "The model simulations in this study are compared to GloSSAC V2 (Thomason et al., 2018; Kovilakam et al., 2020), a time dependent multi-satellite zonal average aerosol climatology which provides extinction data (Figure C1 in Appendix C1)". The gap-filling in the Pinatubo-period is addressed in detail in section 6.2 based on Thomason et al 2018. (see also below).

L57: It would still make sense to include a mention of SAD here since you use it in your analysis.

-> Response: SAD is taken directly from the Level2-SAGE data, in GloSSAC V2 it is not provided. Remarks on that in the MS were removed.

L202: How are SSTs treated, interactive or prescribed? This is important for the calculation of radiative forcing and the comparison with the simulations of Schmidt et al. who used prescribed SSTs.

-> Response: Section 3 is expanded. We mention now ECHAM and inserted: "The temperature and the dynamics above the boundary layer are nudged to the meteorological ERA-Interim reanalysis data of the European Centre for Medium-Range

Weather Forecasts (ECMWF) up to about 100 hPa, while the sea surface temperatures (SST) and sea ice are prescribed using ECMWF data (more details in Jöckel et al., 2006)".

L263: The correction factor is still unclear to me. I appreciate more detail being added to the appendix, but the description in the main text needs to be understandable enough for the reader to make sense of what it is. It would help in the first sentence to clearly state what quantity the correction factor is applied to and what the target is. The 2nd sentence here says "we iterated calculated extinctions to agree with OSIRIS", so this sounds like the correction factor was applied to the extinctions from other instruments (GOMOS?) to produce best agreement with OSIRIS extinction. But later it is mentioned for one case that a correction factor of 0.8 is applied to OSIRIS? Is it not rather that the correction factor is applied to the non-SO2 measurements (OSIRIS, SAGE and GOMOS) to produce best agreement with MIPAS SO2? Or is the correction factor also applied to MIPAS in some cases? Is the factor 1 for all cases when the target dataset (MIPAS or OSIRIS?) is not available?

-> Response: Inserted after "applied": "to the formula provided in Appendix B and the supplement". Also added: after "OSIRIS": "where the problem with data gaps is largest". Added in line 247: "For MIPAS sometimes corrections in the order of 30% were necessary because of gaps. Here the corrected values serve as reference for the other instruments".
Line 600 (Appendix B) is expanded for clarity.
More details see next point.

L266: In this paragraph we have statements that the correction factor takes values "as high as 3" and "up to 2" with no clear indication of a difference in scenarios between those two maximum values. Please clarify.

-> Response: We modified the text to clarify, that the correction factor '3' and the following sentence refer only to Calbuco:
"If data gaps cause a shift of the time period away from the maximum perturbation or a bias in the zonal average, a correction factor is applied to the formula in Appendix B. Correction factors up to 2 have to be applied in some cases because of data gaps, incomplete profiles (both containing zero values) or for high latitudes (examples see Appendix B). One exception is the eruption of Calbuco, with a correction factor of 3, because of a shift of three months due to a big data gap. To estimate the factor in this worst case, we iterated calculated extinctions to agree with OSIRIS and also used observations and assumptions by Vernier et al. (2016) like the decay of extinction by sulfate with time over 4 months."

L304: What does "boxes related to the volcano" mean?

-> Response: This paragraph is rephrased and moved to the previous section because this describes the quantities provided in the table, but not how we use the data in the simulation. See point L306.

L305: "split into boxes considering the mean wind in the lower stratosphere and consistency with nadir observations" does not help understand how or why this split is done.

-> Response: Rephrased, see below.

L306: "lower boundary…" this also does not make sense to me. Is this related to the different min altitudes for tropical and extratropical eruptions that are mentioned later in the manuscript? It wouldn't seem to make any sense to use a lower integration limit over the whole globe for an extratropical eruption. In any case, here you are talking about a single case, so why not just mention what it is and how the total SO2 was calculated?

-> Response: Moved to previous section and rewritten, to make it clear that it refers to latitude belts. This is corrected also in the caption of Table 2.
In section 4: "The amount of sulfur emitted by each single eruption is calculated by integration over the three-dimensional $SO_2$ perturbation plumes, excluding tropospheric emissions below 12 km at high latitudes, 13 km at mid-latitudes, and 14 km at low latitudes. The latter is selected to include possible convective transport from the upper troposphere into the stratosphere in the tropics. The limits in mid and high latitudes above the mean tropopause were selected to exclude cloud perturbations by frontal systems. The plumes don't cover the whole globe, they are always in a latitude range derived manually from the satellite data."
Table 2: "……integrated over latitude belts above 14 km …high latitudes from the 3D mixing ratio perturbations. Listed altitudes and latitudes represent the region of maximum mixing ratio perturbation, the altitudes are close to the top injection height."

L341: Please rephrase to remove "only if"—I doubt all other possible methods have been attempted.
-> Response: Sentence rephrased to remove "only if" by "by considering". Added: "… or most other data bases in ISAMIP, e.g. Mills et al., 2016."

L391: This statement is false: tropopause heights can be extracted from any meteorological reanalysis, and are often available as part of satellite data sets since temperature is typically an ingredient of the retrieval method.

-> Response: We removed that statement. We argue now that fixed heights are more convenient here for comparison with existing literature. We have seen that for example GloSSAC contains a seasonal tropopause climatology which might be useful for improving the radiative forcing calculations.

L399: "Note the odd…" There is nothing odd per se about the downward trend in GloSSAC AOD in 2012, the slope of this trend is similar to other periods including 2016-2017 and 2007-2008. The OSIRIS data mentioned that doesn't show this downward trend—do you refer to Fig 10? To my eye, both OSIRIS and GloSSAC show a mostly monotonic decrease from 2012 to 2014, reaching a minimum value which is slightly lower than any time before around 2005. So qualitatively, I don't see anything "odd" about GloSSAC compared to OSIRIS in and after 2012. What is clear is that there is a discrepancy between GloSSAC and the model beginning in 2012, which is not apparent in the comparison between OSIRIS and the model at 750 nm.

-> Response: We have skipped the sentence to avoid confusions. C.Br. contacted the responsible scientist for that at the SSIRC-Meeting in Leeds. The problem is visible also for 750nm if derived from 525 and 1020 nm GloSSAC data, but less. GloSSAC is now in the upper 2 panels of Fig. 9 and 10, integrated over the same height ranges as for the other data from extinction.

L405: GloSSAC doesn't try anything, it's a dataset.

-> Rephrased: "In GloSSAC gap filling (with lidar and CLAES data) was applied for this case."

L405: Larger than what?

-> Response: Sentence removed.

L415: What temporal resolution is the EMAC simulations shown in? This should be stated in the figure caption. Comparisons should be made at the same temporal resolution--just calculate monthly means of the EMAC data for a comparison with the Schmidt et al. simulations.

-> Response: The EMAC simulations are based on 5-daily emission data sets from MIPAS, GOMOS and OSIRIS. "5-day averages" or "monthly data" is added to the figure captions. Typical numbers of the temporal resolution effects are mentioned in the text of section 6.2 and 6.3. Text modified, figures with a comparison are provided in the supplement.

L446: There are many other differences between these simulations, they use different models! This is not enough evidence to make such a confident statement.

-> Response: Text of section 6.3 modified. There was a misunderstanding concerning the treatment of background stratospheric aerosol. The model output of the values was set to the level of TOA and a background forcing taken from their paper was added to the forcing of Schmidt et al 2018 (Eq.1, first term, which appears to be closest to our approach since we don't consider aerosol cloud interactions) to avoid to compare apples and oranges. See also general remarks.

L449: "Dominant factor" implies a comparison between different factors—what are you comparing to?

-> Response: Text modified: "In the period considered here, the volcanoes are the dominant factor in instantaneous global negative radiative forcing. Background stratospheric aerosol like sulfate from other sources, dust and organics contributes about -0.04 $Wm^{-2}$ to the value of -0.07 $Wm^{-2}$ at TOA (-0.12 $Wm^{-2}$ at tropopause) in volcanically quiescent periods (e.g. in 2000 or 2002). At TOA absolute values up to -0.15 $Wm^{-2}$ (-0.2 $Wm^{-2}$ at tropopause) are reached after Rabaul and Nabro (2011) and more than -0.2 $Wm^{-2}$ (-0.3 $Wm^{-2}$ at tropopause) after Raikoke/Ulawun (2019) eruptions."

L450: concerning the RF values given here, are these absolute values of the simulated IRF, or anomalies with respect to the 2002 quiescent period? This needs to be explained more fully. The sentence seems to imply anomalies, but the values look like absolute values. (The peak after Raikoke is around 0.3 W/m^2, and the value in 2002 is around but larger than 0.1, so the difference should be less than 0.2?). Also, what do you mean by "for" when you connect these values to certain eruptions? It seems clear that as you say, the RF depends also on the history of eruptions before, so it would be better to say the IRF reaches particular values "after" these eruptions, so as not to imply these values are fully attributable to those specific eruptions.

-> Response: Thanks for the remark. The given values are absolute values. The values have been standardized to total values throughout the text for TOA and tropopause. Corrected formulation "for" -> "after". See above.

L459: Schmidt also included nudging of meteorology. Critical is also that Schmidt used historical SSTs and sea ice.

-> Response: Here the models are similar, the main difference is that Schmidt used two simulations and we just one with diagnostic output of radiation fluxes. A remark on that is included now in section 6.3.
In EMAC as described in section 3, see point L202.

L513: Better than what?

-> Response: Wording modified. "an improved $SO_2$ emission database"

L533: "These cause a global negative radiative forcing of 0.12 (0.22 to 0.08) W m^-2": what does the value 0.12 Wm^-2 mean here, do you mean here the value of RF during the quiescent period, or an average over some other period? What does the range 0.22 to 0.08 mean? Later in the sentence, values are given which don't seem to match what was given in Sec 6.3 (0.2 or 0.3 Wm^-2 "for" Raikoke?). Coming here as the last sentence of the conclusions, statements should be repeating or summarizing results shown earlier in the paper, not bringing up newly derived values.

-> Response: Thanks for the remark. The values have been standardized to total values throughout the text for TOA and tropopause. This part is now consistent with section 6.3.
"These cause a global negative radiative forcing of the order of more 0.1 $Wm^{-2}$ at the tropopause, including a background aerosol forcing of about 0.04 $Wm^{-2}$. For example, in the case after the eruptions of Soufriere Hills/Rabaul (2006), Nabro (2011) and the combination of the Sinabung, Wolf and Calbuco eruptions (2015) a negative radiative forcing of down to 0.2 $Wm^{-2}$ (0.15 $Wm^{-2}$ at TOA), and 0.3 $Wm^{-2}$ (0.2 $Wm^{-2}$ at TOA) was reached after Raikoke/Ulawun (2019)."

Fig B1 and B2: please give units for the quantities shown on the plots.

-> Response: Added "ppb" in the captions and "Altitude, km" on y-axes in the figures.

---

## Author Response (AR4)

**General remarks:**

We thank the reviewers again for making very useful suggestions to further improve the paper. Our point-by-point responses to the reviewers' comments and corresponding changes are detailed below in blue text, and the changes are shown in the version of the manuscript with tracked changes (referring to the Manuscript from 16 Feb. 2022 and the last request for major revisions from 27 May 2022).

We thank the editor A. Schmidt and the reviewer for pointing to some misunderstandings concerning the different treatment of background stratospheric aerosol and minor eruptions in EMAC and WACCM. We have corrected the text in the introduction, section 6.3 and conclusions accordingly.

GloSSAC is included now in most frames of Figures 9 and 10 using provided extinction. An additional figure using GloSSAC as counterpart to our Figure 8 is provided in Appendix C1.

We have added also remarks on possible underestimates in radiative forcing due our calculation method in EMAC. The discussion during the review process has led to significant model improvements:

Due to a new much more powerful super computer we were able to redo our simulation with an improved code which considers the contribution of aerosol in the extratropical lowermost stratosphere to radiative forcing as indicated in the reply to the previous version. Now the results are much more consistent with other studies. Figure 11 contains now the improved method for the calculation of forcing. Section 6.3, section 3, abstract, conclusions and appendix C3 are adapted accordingly.

Figure 11 had to be redesigned. We show now mostly the forcing at the top of the atmosphere but kept for the previous versions the published values at the tropopause (see also corresponding TOA values in the supplement):

[Figure]

Global forcing at TOA incl. Backgr., W/m$^2$

[Figure]

Global forcing at TOA incl. Backgr., W/m$^2$

Figure 11. EMAC instantaneous radiative forcing by stratospheric aerosol (red, pink and blue lines, 5-day averages). Solar forcing at the top of the atmosphere (TOA) (dashed red line, upper

panel) is compared to solar forcing at the TOA from satellite observations of the Earth Radiation Budget Experiment (ERBE), 72-day means, light blue crosses (Wong et al., 2006; Toohey et al., 2011). The full red line displays the total (solar+IR) forcing at TOA including contributions from aerosol down to the calculated tropopause. The blue dashed line shows the total forcing using the old scheme at the tropical tropopause (Brühl et al., 2018), the dashed pink line the same with less volcanoes of Brühl et al. (2015). Green bars show annual averages derived from observations by Solomon et al. (2011). The black line shows results from Schmidt et al. (2018) with volcanic radiative forcing at TOA including a background aerosol forcing of -0.05 Wm$^{-2}$. The lower panel is a zoom of the upper panel.

Abstract: "These eruptions cause a total global radiative forcing of the order of -0.1 Wm$^{-2}$ at the top of the atmosphere (TOA) compared to a background stratospheric aerosol forcing of about -0.04 Wm$^{-2}$.
Medium strength eruptions injecting about 400 kt $SO_2$ into the stratosphere or accumulation of consecutive smaller eruptions can lead to a total forcing of about -0.3 Wm$^{-2}$. We show that it is critical to include the contribution of the extratropical lowermost stratospheric aerosol in the forcing calculations."

Conclusions: End rearranged. Last paragraph replaced by: "Note that for the calculation of the forcing by these medium size eruptions it is essential to consider the radiative effect of volcanic aerosol down to the tropopause (Ridley et al., 2014). Including only the aerosol above the 100 hPa level as done in Brühl et al. (2018, 2015) can lead to significant underestimates which were partially hidden in these studies by showing the forcing at the tropopause which is about 0.08 Wm$^{-2}$ stronger than that at the TOA in EMAC."

Appendix C3: "Five-day-average radiative forcing at the TOA is shown in Figure C4. The black curves correspond to the red curves in Figure 11. Using the assumptions of Mills et al. (2016) on the vertical distribution of the $SO_2$ injection leads to less forcing in case of Nabro and after about 5 weeks after the Sarychev eruption (Figure C4, red curves). For the latter the peak is stronger since more mass is injected than in the other approaches. It is dominated by contributions from aerosol in the mid and high latitude lowermost stratosphere. Using Table 2 leads to approximate agreement if the provided altitude is about the top of the column into which is injected, using as a bottom about 1.2 km above the latitude dependent minimum altitude (green curves). Using the full range leads to an underestimate compared to the simulation using the MIPAS observations directly as 3-D $SO_2$ perturbation (blue curves). This is due to more efficient removal processes in the upper troposphere and the lowermost stratosphere than in the layers above. The lower panel for 2009 contains also a sensitivity study similar to the green curve but with the eruption of Mando Hararo neglected as done by Mills et al. (2016) (purple), and one where the top of the injection column for Sarychev was one layer lower than in the case with the blue curve (light blue). These examples show that the altitude of the injection has a large impact on the radiative forcing, but also that not only medium size eruptions matter. For stratospheric AOD shown in the upper panels of Fig. C4 the difference between the approaches is less than for the forcing in the first weeks after the eruption. Later the SAOD decreases faster with the point source approach, especially the one of Mills et al. (2016), than with our method.
Note that considering only aerosol above the fixed pressure level of 100 hPa for the forcing calculations as we did in earlier studies causes misleading results here."

[Figure]

Figure C4. Stratospheric AOD (upper) and global radiative forcing at TOA (lower), Sarychev (left) and Nabro (right), black line as red line in Figure 9 or Figure 11, blue, light blue and green lines "point sources" based on Table 2 with different thickness and vertical position of the column into which is injected (see text), red lines with assumptions of Mills et al. (2016). Purple and light blue curves see text.

We checked the whole manuscript and corrected or removed misleading or distracting words and sentences, including minor corrections in the caption of Figure 7.

**Report #1**
Anonymous Referee #1

**Suggestions for revision or reasons for rejection (will be published if the paper is accepted for final publication)**

All line numbers refer to tracked changes version of manuscript.

L52: "Radiative forcing is…" I don't believe it is true that this approach is "valid only for purely scattering aerosol". The -25 factor comes from Hansen et al. (2005), from model simulations including prescribed stratospheric aerosol for Pinatubo, and is based on the TOA net (SW+LW) flux anomalies, thus incorporating both scattering and absorption effects.

-> Response: Text modified: "…valid only for scattering sulfate aerosol (see e.g. Sellitto et al. (2022))." There are several other references on that, nevertheless, the factor is still good for plausibility studies.

L56: The description of GloSSAC is somewhat improved compared to prior versions. But still, some evidence (via citation) should be included to support the claim that GloSSAC includes "large uncertainties due to data gaps"? I think here you are really referring to the issues with "saturation" after Pinatubo? If so, it would be helpful to be specific about it and make reference to statements in the relevant literature, e.g., from the GloSSAC papers.

-> Response: Rephrased to a more neutral formulation without the remark on data gaps: "The model simulations in this study are compared to GloSSAC V2 (Thomason et al., 2018; Kovilakam et al., 2020), a time dependent multi-satellite zonal average aerosol climatology which provides extinction data (Figure C1 in Appendix C1)".
The gap-filling in the Pinatubo-period is addressed in detail in section 6.2 based on Thomason et al 2018. (see also below).

L57: It would still make sense to include a mention of SAD here since you use it in your analysis.

-> Response: SAD is taken directly from the Level2-SAGE data, in GloSSAC V2 it is not provided. Remarks on that in the MS were removed.

L202: How are SSTs treated, interactive or prescribed? This is important for the calculation of radiative forcing and the comparison with the simulations of Schmidt et al. who used prescribed SSTs.

-> Response: Section 3 is expanded. We mention now ECHAM and inserted:
"The temperature and the dynamics above the boundary layer are nudged to the meteorological ERA-Interim reanalysis data of the European Centre for Medium-Range Weather Forecasts (ECMWF) up to about 100 hPa, while the sea surface temperatures (SST) and sea ice are prescribed using ECMWF data (more details in Jöckel et al., 2006)".

In section 3 (line 232) we adapted the text for the new calculation method of radiative forcing:
"… instantaneous forcing is calculated online from the difference of fluxes for the cases with stratospheric aerosol above the calculated tropopause (in earlier studies only above 100 hPa) …"

L263: The correction factor is still unclear to me. I appreciate more detail being added to the appendix, but the description in the main text needs to be understandable enough for the reader to make sense of what it is. It would help in the first sentence to clearly state what quantity the correction factor is applied to and what the target is. The 2nd sentence here says "we iterated calculated extinctions to agree with OSIRIS", so this sounds like the correction factor was applied to the extinctions from other instruments (GOMOS?) to produce best agreement with OSIRIS extinction. But later it is mentioned for one case that a correction factor of 0.8 is applied to OSIRIS? Is it not rather that the correction factor is applied to the non-SO2 measurements (OSIRIS, SAGE and GOMOS) to produce best agreement with MIPAS SO2? Or is the correction factor also applied to MIPAS in some cases? Is the factor 1 for all cases when the target dataset (MIPAS or OSIRIS?) is not available?

-> Response: We added the formula for the $SO_2$ mixing ratio perturbation to the main text in section 4:

"The $SO_2$ mixing ratio perturbation $\Delta VMR$ is derived from the extinction perturbation $\Delta\beta_{ext}$ (750nm) as in Equation 1 using a constant ratio between model calculated sulfate concentration and its share on extinction in the lower stratosphere of low latitudes:

$$\Delta VMR = 1.2 \times 10^{12}\ \Delta\beta_{ext}/\rho\ \ f \tag{1}$$

$\rho$ is the altitude dependent air density in molecules/cm$^3$ (Examples and more details see Appendix B). We assume that the spatial patterns of the perturbation of extinction and sulfate are the same as for $SO_2$. A similar technique is used for OSIRIS and can be used for SAGE II data.

If data gaps cause a shift of the time period away from the maximum perturbation or a bias in the zonal average, a correction factor ($f \neq 1$) is applied to Equation 1."

Line 605 (Appendix B) is expanded for clarity: "If the time lag of data is several weeks a correction factor >1 has to be applied in Equation 1 to account for removal processes, if another event is relatively close in time, the factor has to be <1 to remove the influence of the previous event (see Table S1 in supplement, which indicates also additional uncertainties for a few cases in 2018 and 2019 due to too sparse data)."

Added in line 247: "For MIPAS sometimes corrections in the order of 30% were necessary because of gaps. Here the corrected values serve as reference for the other instruments".

More details see next point.

L266: In this paragraph we have statements that the correction factor takes values "as high as 3" and "up to 2" with no clear indication of a difference in scenarios between those two maximum values. Please clarify.

-> Response: We modified the text to clarify, that the correction factor '3' and the following sentence refer only to Calbuco:

"… Correction factors up to 2 have to be applied in some cases because of data gaps, incomplete profiles (both containing zero values) or for high latitudes (examples see Appendix B). One exception is the eruption of Calbuco, with a correction factor of 3 for removal processes, because of a shift of three months due to a big data gap. To estimate the factor in this worst case, we iterated calculated extinctions to agree with OSIRIS and also used observations and assumptions by Vernier et al. (2016) like the decay of extinction by sulfate with time over 4 months. On the other hand, the factor can be as small as 0.5 to account for sulfate remnants of eruptions occurring 2-4 weeks before the date of the eruption to be analysed or for cloud perturbations. These factors, together with the used time periods, are provided in the electronic supplement (Table S1 for OSIRIS and Table S2 for GOMOS)".

L304: What does "boxes related to the volcano" mean?

-> Response: This paragraph is rephrased and moved to the previous section because this describes the quantities provided in the table, but not how we use the data in the simulation. See point L306.

L305: "split into boxes considering the mean wind in the lower stratosphere and consistency with nadir observations" does not help understand how or why this split is done.

-> Response: Rephrased and rearranged, see below.

L306: "lower boundary…" this also does not make sense to me. Is this related to the different min altitudes for tropical and extratropical eruptions that are mentioned later in the manuscript? It wouldn't seem to make any sense to use a lower integration limit over the whole globe for an extratropical eruption. In any case, here you are talking about a single case, so why not just mention what it is and how the total SO2 was calculated?

-> Response: Moved to previous section and rewritten, to make it clear that it refers to latitude belts. This is corrected also in the caption of Table 2.
In section 4: "The amount of sulfur emitted by each eruption is calculated by integration over the three-dimensional $SO_2$ perturbation plumes, excluding tropospheric emissions below 12 km at high latitudes, 13 km at mid-latitudes, and 14 km at low latitudes. The latter is selected to include possible convective transport from the upper troposphere into the stratosphere in the tropics. The limits in mid and high latitudes above the mean tropopause were selected to exclude cloud perturbations by frontal systems or uncertain satellite data. This can lead to an underestimate of injected mass in some cases. The plumes don't cover the whole globe, they are always in a latitude range derived manually from the satellite data."
Table 2: "… integrated over latitude belts above 14 km … high latitudes from the 3D mixing ratio perturbations. Listed altitudes and latitudes represent the region of maximum mixing ratio perturbation, the altitudes are close to the top injection height."

L341: Please rephrase to remove "only if"—I doubt all other possible methods have been attempted.
-> Response: Sentence rephrased to remove "only if" by "by considering". Added: "… or most other data bases in ISAMIP, e.g. Mills et al. (2016)."

L391: This statement is false: tropopause heights can be extracted from any meteorological reanalysis, and are often available as part of satellite data sets since temperature is typically an ingredient of the retrieval method.

-> Response: Thanks for the remark. We removed that statement. We argue now that fixed heights are more convenient here for comparison with existing literature.
"For practical reasons, the total stratospheric Aerosol Optical Depth (AOD) is obtained by the vertical integral of the aerosol extinction above an altitude of about 16 km in the tropics and above about 13 km for mid-latitudes and high latitudes, to allow for a direct comparison with existing literature (Santer et al., 2014; Glantz et al., 2014) and satellite data." See also captions of Fig. 9 and 10.

Furthermore, we have improved the radiative forcing calculations by using a model calculated tropopause height, instead of using a fixed pressure level of 100 hPa (see General remarks and sec. 6.3).

L399: "Note the odd…" There is nothing odd per se about the downward trend in GloSSAC AOD in 2012, the slope of this trend is similar to other periods including 2016-2017 and 2007-2008. The OSIRIS data mentioned that doesn't show this downward trend—do you refer to Fig 10? To my eye, both OSIRIS and GloSSAC show a mostly monotonic decrease from 2012 to 2014, reaching a minimum value which is slightly lower than any time before around 2005. So

qualitatively, I don't see anything "odd" about GloSSAC compared to OSIRIS in and after 2012. What is clear is that there is a discrepancy between GloSSAC and the model beginning in 2012, which is not apparent in the comparison between OSIRIS and the model at 750 nm.

-> Response: We have skipped the sentence to avoid confusions. C.Br. contacted the responsible scientist for that at the SSIRC-Meeting in Leeds. The problem is visible also for 750 nm if derived from 525 and 1020 nm GloSSAC data, but less. GloSSAC AOD is now in the upper 2 panels of Fig. 9 and 10, integrated over the same height ranges as for the other data from extinction.

L405: GloSSAC doesn't try anything, it's a dataset.

-> Rephrased: "In GloSSAC gap filling (with lidar and CLAES data) was applied for this case."

L405: Larger than what?

-> Response: Sentence removed.

L415: What temporal resolution is the EMAC simulations shown in? This should be stated in the figure caption. Comparisons should be made at the same temporal resolution--just calculate monthly means of the EMAC data for a comparison with the Schmidt et al. simulations.

-> Response: The EMAC simulations are based on 5-daily emission data sets from MIPAS, GOMOS and OSIRIS. "5-day averages" or "monthly data" is added to the figure captions. Typical numbers of the temporal resolution effects are mentioned in the text of section 6.2 and 6.3. Text modified, figures with a comparison are provided in the supplement (Fig. S2 and S3).

L446: There are many other differences between these simulations, they use different models! This is not enough evidence to make such a confident statement.

-> Response: Text of section 6.3 modified. There was a misunderstanding concerning the treatment of background stratospheric aerosol. The model output of the values was set to the level of TOA and a background forcing taken from their paper was added to the forcing of Schmidt et al. (2018) (Eq.1, first term, which appears to be closest to our approach since we don't consider aerosol cloud interactions) to avoid to compare apples and oranges. See also general remarks.

"The new model simulations for the forcing at TOA with the additional volcanic eruptions (red line) are closer to the estimates from satellite extinction measurements of SAGE, GOMOS and CALIOP by Solomon et al. (2011) (green bars) than in previous studies (e.g., Brühl et al. (2015), pink dashed line and Brühl et al. (2018), blue dashed line, for TOA see supplement, Fig. S1). In Fig. 11 the published forcing values at the tropical tropopause of the previous studies are shown which are systematically more negative than the values at TOA. For Sarychev it is clear that this cannot compensate for the effect of the neglect of aerosol in the extratropical lowermost stratosphere. A comparison with volcanic radiative forcing (aerosol-radiation interactions) from Schmidt et al. (2018) is shown by the black line, including an instantaneous forcing of -0.05 Wm$^{-2}$ for stratospheric background aerosol (derived from numbers provided).

Especially for high latitude eruptions their forcing is larger than EMAC and the annual averages of Solomon et al. (2011), also because of a higher aerosol load in the lowermost stratosphere than in EMAC (see sensitivity studies for Sarychev in Appendix C3)."

L449: "Dominant factor" implies a comparison between different factors—what are you comparing to?

-> Response: Text modified: "In the period considered here, the volcanoes are the dominant factor in instantaneous global radiative forcing. Background stratospheric aerosol like sulfate from other sources, dust and organics contributes about -0.04 $Wm^{-2}$ to the value in the order of -0.1 $Wm^{-2}$ at TOA in volcanically quiescent periods (e.g. in 2000, 2002 or 2004). At TOA absolute values up to -0.2 $Wm^{-2}$ (-0.14 $Wm^{-2}$ old approach where only aerosol above 100 hPa was considered) are reached after Rabaul (2006), Kasatochi (2008), Nabro (2011) and Calbuco/Sinabung (2015) and stronger than -0.32 $Wm^{-2}$ (-0.2 $Wm^{-2}$ old approach) after Raikoke/Ulawun (2019) eruptions. The value for Raikoke/Ulawun is within the range discussed in Kloss et al. (2021)."

L450: concerning the RF values given here, are these absolute values of the simulated IRF, or anomalies with respect to the 2002 quiescent period? This needs to be explained more fully. The sentence seems to imply anomalies, but the values look like absolute values. (The peak after Raikoke is around 0.3 W/m^2, and the value in 2002 is around but larger than 0.1, so the difference should be less than 0.2?). Also, what do you mean by "for" when you connect these values to certain eruptions? It seems clear that as you say, the RF depends also on the history of eruptions before, so it would be better to say the IRF reaches particular values "after" these eruptions, so as not to imply these values are fully attributable to those specific eruptions.

-> Response: Thanks for the remark. The given values are absolute (or total) values. The values have been standardized to total values throughout the text for TOA and tropopause. All values of radiative forcing have been corrected by the results of the improved simulations with aerosol above the calculated tropopause considered.
Corrected formulation "for" -> "after". See above, point L449.

L459: Schmidt also included nudging of meteorology. Critical is also that Schmidt used historical SSTs and sea ice.

-> Response: Here the models are similar, the main difference is that Schmidt et al. (2018) used two simulations and we just one with diagnostic output of radiation fluxes. A remark on that is included now in section 6.3.
In EMAC as described in section 3, see point L202.

L513: Better than what?

-> Response: Wording modified. "an improved $SO_2$ emission database"

L533: "These cause a global negative radiative forcing of 0.12 (0.22 to 0.08) W m^-2": what does the value 0.12 Wm^-2 mean here, do you mean here the value of RF during the quiescent period, or an average over some other period? What does the range 0.22 to 0.08 mean? Later in the sentence, values are given which don't seem to match what was given in Sec 6.3 (0.2 or 0.3 Wm^-2 "for" Raikoke?). Coming here as the last sentence of the conclusions, statements

should be repeating or summarizing results shown earlier in the paper, not bringing up newly derived values.

-> Response: Thanks for the remark. The values have been standardized to total values throughout the text for TOA. This part is now consistent with section 6.3.

"These cause a global negative radiative forcing of the order of more than 0.1 Wm$^{-2}$ at the TOA, including a background aerosol forcing of about 0.04 Wm$^{-2}$. For example, in the case after the eruptions of Soufriere Hills/Rabaul (2006), Nabro (2011) and the combination of the Sinabung, Wolf and Calbuco eruptions (2015) a negative radiative forcing of down to 0.2 Wm$^{-2}$ (0.14 Wm$^{-2}$ old approach), and 0.32 Wm$^{-2}$ (0.2 Wm$^{-2}$ old approach) was reached after Raikoke/Ulawun (2019) at TOA."

Fig B1 and B2: please give units for the quantities shown on the plots.

-> Response: Added "ppb" in the captions and "Altitude, km" on y-axes in the figures.

---

## Author Response (AR5)

**Response to the Editor**:

**General remarks:**

We would like to thank our editor A. Schmidt for her quick response to our inquiry and her personal efforts in revising our manuscript. We thank her for making very useful suggestions to further improve the paper. Our point-by-point responses to the Editors' comments and corresponding changes are detailed below in blue text, and the changes are shown in the version of the manuscript with tracked changes (together with other minor corrections).

• Lines 55-56: The factor of -25 is not stated in Sellitto et al. (2022). I agree a factor of -25 is appropriate but you must provide the correct reasoning and provide an appropriate reference, thus please revise the sentence in question. Note, sulfate particles also absorb outgoing terrestrial radiation and near-infrared solar radiation, causing stratospheric heating.

-> Response: The sentence was rephrased and a reference was added.

"Friberg et al. (2018) included the entire time series of CALIOP (Cloud-Aerosol Lidar with Orthogonal Polarization) data from 2006 to 2015 and derived stratospheric AOD using reanalysis data for the tropopause, but only mention medium size eruptions explicitly. Radiative forcing is estimated in this case by multiplying AOD with a factor of -25 (Hansen et al., 2005) rather than using a radiative transfer model, an approximation valid in the absence of major forest fires (see e.g. Sellitto et al. (2022))."

• Lines 264ff: The description of the correction factor that has been applied is still hard to follow even with the text you added in the Appendix. Please revise the text to fully address the reviewer's initial comments on this part of the manuscript. The justification reads okay but technical details are still missing so that readers can follow as to what has been done.

-> Response: More details are added to the text:

"…, f an empirical factor which equals 1 for sufficient data coverage (examples and more details see Appendix B)."

"If data gaps cause a shift of the time period away from the maximum perturbation or a low bias in the zonal average due to the zero values in the gaps at some longitude bin or in a period, a correction factor f > 1 based on comparison of total injected $SO_2$ mass with the one taken from other satellite data is applied in Equation 1. ... An extreme case is the eruption of Calbuco, with a correction factor of 3 for removal processes, because of a shift..."

We have also expanded the corresponding part on MIPAS in the second paragraph of the same section:

"For MIPAS sometimes corrections in the order of up to 30% were necessary because of gaps (containing zero values) to be consistent with the total injected $SO_2$ mass derived from MLS (Microwave Limb Sounder) or OMI (Ozone Monitoring Instrument). Here the corrected values serve as reference for $SO_2$ derived from the instruments measuring extinction."

Furthermore, a remark on related uncertainties is added in the conclusions in the second paragraph: "In our approach the largest uncertainties are due to the handling of gaps in the satellite data."

In "data availability" we added: "A detailed documentation on the used data for MIPAS, SAGE, OSIRIS and GOMOS for the individual events is available from the authors on request."

• Throughout the manuscript make sure to clearly state that you are calculating an instantaneous radiative forcing whereas Schmidt et al. (2018) for example calculated an effective radiative forcing. You must clearly indicate what forcings are compared with each other. This applies to relevant Figures such as Figure 11 and the main text.

-> Response: Throughout the manuscript where it refers to our simulations, we have added the term "instantaneous" radiative forcing to the text. In Section 6.3 and Figure 11, we have added "effective" radiative forcing (ari) for reference with Schmidt et al. (2018). For details see the manuscript with tracked changes.

• Please do not use red and green colours at the same time in Figures (in particular in line plots) as readers with colour vision deficiencies will not be able to correctly interpret your figures. See ACP guidance: https://www.atmospheric-chemistry-and-physics.net/submission.html#figurestables

-> Response: Thank you for drawing our attention to graphics with color vision deficiencies. For our color maps we have now chosen the cmocean perceptually-uniform colormap "cmocean_thermal".
https://ferret.pmel.noaa.gov/Ferret/faq/ferret-color-palettes
For line plots, we have substituted green colors.